

# Depolarization measurements using the CANDAC Rayleigh-Mie-Raman Lidar at Eureka, Canada

Emily M. McCullough[1,2a,*], Robert J. Sica[1], James R. Drummond[2], Graeme Nott[2,4a], Christopher Perro[2], Colin P. Thackray[2], Jason Hopper[2], Jonathan Doyle[2], Thomas J. Duck[2], and Kaley A. Walker[3]

[1]Department of Physics and Astronomy, The University of Western Ontario, 1151 Richmond St., London, ON, N6A 3K7
[2]Department of Physics and Atmospheric Science, Dalhousie University, 6310 Coburg Rd., PO Box 15000, Halifax, NS, B3H 4R2
[3]Department of Physics, University of Toronto, 60 St. George St., Toronto, Ontario, M5S 1A7
[4]Facility for Airborne Atmospheric Measurements, Building 146, Cranfield University, Cranfield, MK43 0AL, UK
[a]Indicates present affiliation

*Correspondence to:* Emily M. McCullough (e.mccullough@dal.ca)

**Abstract.** The Canadian Network for the Detection of Atmospheric Change (CANDAC) Rayleigh–Mie–Raman Lidar (CRL) at Eureka, Nunavut, has measured tropospheric clouds, aerosols, and water vapour since 2007. In remote and meteorologically significant locations, such as the Canadian High Arctic, the ability to add new measurement capability to an existing well-tested facility is extremely valuable. In 2010, linear depolarization 532 nm measurement hardware was installed in the lidar's receiver.

To reduce its impact on the existing, well-characterized lidar channels, the depolarization hardware was placed near the end of the receiver cascade. The upstream optics already in place were not optimized for preserving the polarization of received light. Calibrations and Mueller matrix calculations were used to determine and mitigate the contribution of these upstream optics on the depolarization measurements. The results show that with appropriate calibration, indications of cloud particle phase (ice vs. water) are now possible to precision within $\pm 20\%$ uncertainty at time and altitude resolutions of 5 min $\times$ 37.5 m, with higher

precision and higher resolution possible in select cases. Monitoring changes in Arctic cloud composition, including particle phase, is essential for a complete understanding of the changing climate locally and globally.

## 1 Introduction

Clouds influence Earth's radiation budget, and thus its weather and climate. Clouds reflect sunlight (cooling), and trap heat from the ground (warming). The combined effect of these competing influences is poorly understood, especially in the Arctic,

because it depends significantly on the structure and microphysical properties of the clouds, and the environment in which the clouds exist (Curry et al., 1996). Ice clouds radiate differently than water clouds (Sun and Shine, 1994). Tropospheric clouds occur frequently in the Arctic, with liquid content found at all times of year, often within mixed-phase clouds (Intieri et al., 2002; Shupe, 2011). The evaluation of cloud phase in models requires more observational datasets in order to improve (Shupe, 2011), with phase transitions being of particular interest (Kalesse et al., 2016). Therefore, measurements of cloud particle phase (ice vs. water) are necessary in order to more fully understand the radiation balance of the Arctic atmosphere. Liquid droplets can exist well below 0° C, so cloud temperature is not sufficient to determine the phase of cloud particles.





Lidar depolarization measurements, which discern cloudy regions containing spherical particles (i.e. liquid droplets) from those containing nonspheres (i.e. ice particles), are one method by which cloud particle phase may be examined (Schotland

et al., 1971; Sassen, 2005; Bourdages et al., 2009).

The Polar Environment Atmospheric Research Laboratory (PEARL) is located in Eureka, Nunavut (80° N, 86° W) Canada's High Arctic. PEARL has more than 25 instruments dedicated to the in situ and remote sensing study of atmospheric phenomena at a latitude where few measurements are typically available. With climate changes amplified at such latitudes (Serreze and Barry, 2011), PEARL's measurements give a valuable contribution to global atmospheric and environmental science.

The Canadian Network for the Detection of Atmospheric Change (CANDAC) Rayleigh–Mie–Raman Lidar (CRL), was installed at PEARL in 2007. It has since made measurements of visible and UV particulate backscatter coefficient, aerosol extinction, water vapour mixing ratio, and other quantities using its 355 nm and 532 nm lasers and comprehensive detection package (Doyle et al., 2011; Nott et al., 2012). Adding 532 nm linear depolarization capabilities to this instrument is an economical way to add additional capacity to study Arctic clouds, in concert with other instruments at PEARL such as the Mil-

limetre Cloud Radar (Moran et al., 1998), the E-AERI interferometer (Mariani et al., 2012), and the Starphotometer (Baibakov et al., 2015).

The basic quantity upon which lidar depolarization calculations are based is the ratio of photons returned with polarization perpendicular to that of the transmitted laser beam, to those returned with polarization parallel to that of the transmitted laser beam (e.g. Hohn (1969); Schotland et al. (1971); Liou and Schotland (1971)). This quantity is known as the depolarization

ratio, $\delta$. An alternate expression for depolarization is the ratio of photons returned with polarization perpendicular to that of the transmitted laser beam, to the total number of returned photons of any polarization (Flynn et al., 2008; Gimmestad, 2008). This alternative to the depolarization ratio is called the depolarization parameter, $d$.

In the traditional expression of $\delta$ and $d$ some intrinsic assumptions are made regarding the nature of the signals recorded in the parallel and perpendicular channels. It was not obvious that these assumptions would be appropriate for CRL, given the

numerous optics upstream of the depolarization PMT. With careful characterisation of the lidar system using a full Mueller matrix calibration scheme we show that the traditional equations for lidar depolarization are valid for CRL, and we then find the appropriate calibration constants. A Mueller Matrix approach has been used by Hayman and Thayer (2009), in which they use Mueller Matrix algebra to more fully explore the optical properties of the atmosphere. Here, we use such mathematics with the aim of more fully diagnosing the optical properties of CRL itself, similar to the approach taken by Di et al. (2016).

The Mueller Matrix algebra upon which this technique relies was introduced as lectures and conference proceedings by Hans Mueller in the early 1940s (e.g. Mueller (1946a, b, 1948)). These and his previous works (Mueller, 1943a, b) remain difficult to obtain, and those available (e.g. in summary report Bush (1946), which describes the design and use of the shutter described in Mueller (1943a)) do not explicitly demonstrate the matrix algebra. A better and more available source describing all of the Mueller matrix algebra in considerable detail is the thesis of Mueller's PhD student, Nathan Grier Parke III (Parke III, 1948).

The result for CRL is a new depolarization tool tied into a scientifically significant long-term measurement record, all without compromise to the continued acquisition of the original types of data. To date, linear depolarization measurements have been





made for three polar sunrise seasonal campaigns at Eureka: 2013, 2014, and 2016. No measurements were made with the lidar during 2015. Calibrated examples from the first season's measurements will be presented in Sect. 8.

## 2 Installation of depolarization hardware

To make depolarization measurements, the lidar must be able to distinguish between backscattered light which is polarized parallel to the outgoing laser light, and that which is returned polarized perpendicularly. While a polarizing beamsplitter and two additional photomultiplier tubes (PMTs) would accomplish this requirement, we opted to use a rotating polarizer which permits lidar returns in two orthogonal polarization planes to be measured by a single detector in an automated version of the measurement approach used in the very first depolarization lidars (Schotland et al., 1971). This design reduces the number

of differences between the hardware of both depolarization channels because the backscattered light traverses identical optics and uses the same photomultiplier tube. Given that the basic depolarization calculation is a ratio, having identical components means that many terms cancel out of the depolarization calculation.

The priority during installation of the polarization capability was not to impact any of the well-calibrated measurements in the other pre-existing lidar channels (Nott et al., 2012). No optics for the other channels were changed or removed during the

installation of the depolarization channel, as these changes could have affected the other measurements.

### 2.1 Polarotor

The Licel Polarotor rotating polarizer (Licel GmbH, 2006) was designed specifically for multispectral detection systems such as that of CRL . The polarotor acts as the master trigger for the lidar. Its $\alpha$-BBO Glan Thompson prism is spun steadily at high speed, and a synchronization pulse from the built-in timing disk triggers the lidar system at 10 Hz. This trigger signals the laser

to fire and the detectors to record every time the prism rotates through $450°$, which corresponds to the prism's acceptance plane being rotated by $90°$. The PMT is exposed to backscattered laser light which is polarized parallel to, and perpendicular to, the outgoing laser light, on alternate laser shots. Two recording buffers are used in the Licel Transient Recorder, one for parallel and one for perpendicular photocount profiles. The extinction ratio of the polarizer was characterized by the manufacturer to be $5 \times 10^{-5}$ or smaller (Licel GmbH, 2006), leading to high-quality separation of the polarization states. Only photons of

the appropriate polarization orientation enter the measurement profiles for each of the parallel and perpendicular photocount measurement channels.

### 2.2 Positioning of depolarization channel within CRL

During manufacture, the CRL polychromator had two spare locations for potential expansion of the lidar. The depolarization channel was installed in the spare location between the 532 nm Visible Rayleigh Elastic channel and the 607 nm Nitrogen

channel (Fig. 1). This location is on the visible light side of the polychromator, but it suffers from being "downstream" of many optics. The original Chroma 580DCLP Visible Long Wave Pass (VLWP) filter was chosen in 2007 specifically to reflect as much 532 nm light into the original Visible Rayleigh Elastic channel as possible (approximately 97 %). Part of this reflected



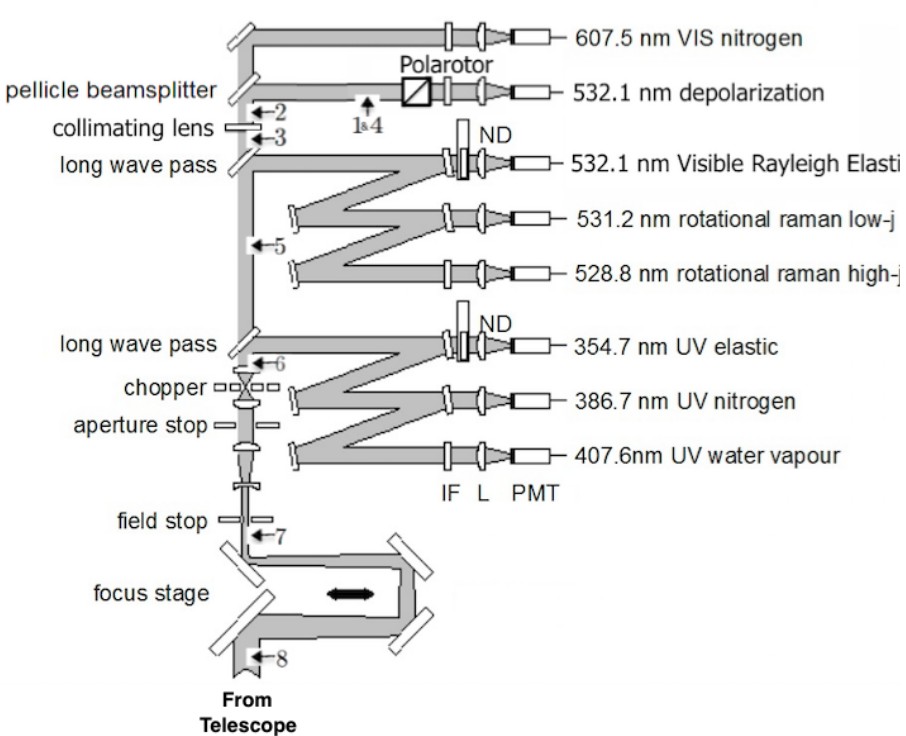

**Figure 1.** Diagram of CRL's receiver system, showing all 7 existing measurement channels plus the newly installed depolarization hardware. Numbers correspond to calibration test numbers from Table 1, and indicate the test locations of the depolarizing sheet used in those tests. "7" also marks the location of the calibration cube polarizer added to the system for the calibrations in Sect. 5.1, and "8" is also the location of the lamp and depolarization sheet during the polarized light tests in that section. During regular lidar sky measurements, neither the depolarizing sheet nor the calibration cube polarizer remain in the optical path. This figure is based on Fig. 2 of Nott et al. (2012).

light is used in two subsequent visible channels at 531.2 nm and 528.7 nm, thus the requirement for maximum reflectivity around 532 nm for the VLWP (Nott et al., 2012). The depolarization channel uses the small amount of residual 532 nm light which is transmitted through this VLWP filter on its way to the 607 nm channel, where it would normally be rejected by the 607 nm channel's interference filter.

5    During depolarization channel installation, a partially reflective optic was installed to redirect the residual 532 nm light into the depolarization channel, allowing the 607 nm light to continue on to the final PMT. The 607 nm channel optics were already well aligned and characterized at the time of depolarization installation. Therefore, a regular plate beamsplitter or dichroic mirror could not be used to pick off the light for the depolarization channel; this would have translated the transmitted 607 nm

10   light too much, and the downstream channel would have had to be realigned. A 3" CVI–Melles Griot 633 nm 50/50 pellicle beamsplitter is the most non-polarizing option available at 532 nm in reflectance, which still allows as much 607 nm light through as possible, with the smallest possible beam translation.





From the pellicle beamsplitter, the 532 nm light travels into the polarotor which is mounted atop the polychromator. It sits on a 2" diameter beamtube which is 3" tall. The diameter of the light beam is slightly larger than 2" at the pellicle and, due to a collimating lens just upstream of the pellicle, is slightly converging. The polarotor has an acceptance diameter of 20 mm, and an acceptance angle of 15°. By placing the polarotor a sufficient distance from the pellicle, the entire beam is accepted by the polarotor without the need for extra optics in between. It is also convenient to have the cables from the polarotor to the electronics rack be accessible without the need to open the polychromator.

Above the polarotor, there is a beamtube containing an interference filter, then a 1" diameter beamtube containing a 75 mm focusing planoconvex lens. Next, there is an adjustable focusing tube in which the Hamamatsu R7400-03 PMT is seated, which allows the active area of the photomultiplier tube to be positioned at an appropriate distance from the focusing lens. The interference filter specifications are: Andover Corporation Part Number 532FS02-25. 25 mm diameter, BW 1±0.2 nm centered at 532.0±0.2 nm, transmission greater than 45 %, $1 \times 10^{-4}$ average blocking from X-ray to far infrared wavelengths (Andover Corporation, 2015).

## 3 Theory for depolarization calculations

### 3.1 Traditional calculation methods

The depolarization ratio is the ratio of backscatter coefficients, $\beta$, for photons polarized perpendicular to those of the transmitted laser beam ($\perp$) and those polarized parallel to those of the laser beam ($\parallel$):

$$\delta = \beta_\perp / \beta_\parallel. \tag{1}$$

In the event that the atmosphere does not depolarize the beam there will be no photons backscattered with polarization different than the transmitted light, and therefore $\delta = 0$. In the case of complete depolarization, $\delta = 1$. Gimmestad (2008) advocates discussing lidar depolarization in terms of depolarization parameter $d$, defined in terms of light intensity $I$ by

$$d = I_{\text{unpol.}} / (I_{\text{pol.}} + I_{\text{unpol.}}), \tag{2}$$

instead of depolarization ratio $\delta$. This choice is sensible, as $d$ is the physical property of the atmosphere (see full derivation in following section) which lidar depolarization measurements seek to investigate. It is the portion of the total light intensity $I$ which has become depolarized through scattering. (Similar descriptions, called depolarization factor, are given as early as van de Hulst (1957)). The conversion between the quantities $d$ and $\delta$ is:

$$d = 2\delta / (1 + \delta). \tag{3}$$

Because lidars measure signals from PMTs, and not the backscatter coefficient directly, these equations for $d$ and $\delta$ must be reformulated in terms of lidar observables. Gimmestad (2008) demonstrates this development using Mueller Matrix algebra, with normalized matrices. The signals in the receivers, $S$, are individually "assumed to be calibrated", but no further details about these calibrations are provided. Presumably, this assumption considers the combined effects of all optics upstream of the





PMT and the gain of the PMT, acting together as a constant attenuation factor for each individual channel. If the factors differ between channels, the overall effect in the system as a whole is that of partial polarizer. Under these conditions, the equations for depolarization are given as:

$$\delta = k\frac{S_\perp}{S_\parallel},\tag{4}$$

$$d = \frac{2k\frac{S_\perp}{S_\parallel}}{1 + k\frac{S_\perp}{S_\parallel}} = \frac{2}{\frac{1}{k}\frac{S_\parallel}{S_\perp} + 1},\tag{5}$$

in which: $S_\perp$ is the signal measured by the perpendicular channel, $S_\parallel$ is the signal measured by the parallel channel, and $k = \frac{G_\parallel}{G_\perp}$ is the depolarization calibration constant, in which $G_\parallel$ is the gain (or attenuation) of the parallel channel, and $G_\perp$ is the gain (or attenuation) of the perpendicular channel. The second form for $d$ in Eq. (5) is easier to handle experimentally as each measurement appears only once and thus uncertainties may be considered uncorrelated.

In each equation, a single $k$ value determines the calibration. $k$ can be determined by introducing unpolarized light into the detector (i.e. setting $d = \delta = 1$), measuring the signals in each channel, and solving for $k$. The value of $k$ is generally defined as the ratio of gains in the PMTs for each channel. This choice precludes the possibility of cross talk, polarization rotation, and other optical effects in the receiver which may be present in any or all of the optical components. CRL uses a single PMT to measure both parallel and perpendicular channels on alternate laser shots. In the idealized case where the optics do not contribute to the polarization, $k$ would be unity. However, characterizing measurements suggest a value closer to $k = 21$ for CRL (Sect. 6.3), indicating that optics upstream of the polarotor are significantly polarizing. As this is the case, it seemed sensible to investigate potential additional optical contributions. If the overall impact of these optics is not exclusively that of a partial polarizer, showing for instance behaviour of a wave plate or a polarization rotator, then Eq. (4) and Eq. (5) are insufficient to describe the depolarization ratio and depolarization parameter for CRL. The following section uses Mueller Matrix algebra to re-derive the equation for $d$, allowing for optical effects in the upstream optics. The largest contributor to $k$ for CRL was found to be the Visible Long Wave Pass filter (Sect. 7.2, and Table 1), and the overall contribution of the optics is found to conform to that of a partial polarizer, rendering Eq. (4) and Eq. (5) appropriate for CRL.

### 3.2 More complete Mueller Matrix development

A Mueller Matrix approach was used to understand the signals being measured by the depolarization channel, with as few simplifications and assumptions as possible. In the case of atmospheric lidar operation, the original laser light is described as a Stokes vector which will be operated on by several optical elements, each of which can be described by a 4×4 matrix. We can thus determine the Stokes vector of the light which enters the PMTs, and predict the signal which will be measured in each of our channels: $S_\parallel$ (parallel channel measurement), and $S_\perp$ (perpendicular channel measurement).

In its most basic format, Mueller algebra functions as: $\boldsymbol{L'} = \mathbf{M}\boldsymbol{L}$, in which $\boldsymbol{L}$ and $\boldsymbol{L'}$ are the Stokes vectors of incident and emerging radiation, respectively, and $\mathbf{M}$ is the $4 \times 4$ real matrix effect of the instrument (Parke III, 1948). Stokes vectors add incoherently, and therefore the effect of any two components in series is equal to the product of their matrices. Thus, $\mathbf{M}$ can rep-





resent the product of two or more different optical matrices. Stated in terms of a matrix equation, the effects of the atmosphere $\mathbf{M}_{\text{atm}}$ and of the lidar's receiver optics $\mathbf{M}_{\text{receiver}}$ on the emitted laser light are: $\boldsymbol{I}_{\text{measured}} = \mathbf{M}_{\text{receiver}}\mathbf{M}_{\text{atm}}\mathbf{M}_{\text{transmitter}}\boldsymbol{I}_{\text{laser}}$.

30      The CRL laser emits horizontally linearly polarized light $\boldsymbol{I}_{\text{laser}} = I_{\text{laser}}[1\ 1\ 0\ 0]'$. The laser intensity $I_{\text{laser}}$ varies shot-to-shot with such quantities as laser voltage and flashlamp age. Using alternate laser shots to build up 300 shot sums in each parallel and perpendicular one minute lidar scan allows these channels to measure during simultaneous one minute periods. Taking many shots into each sum allows the short timescale ($<$ 1 second) laser voltage variations to average out in each measurement. Therefore, $I_{\text{laser}}$ can be considered the same for both channels for each measurement pair.

The optical backscattering effects of the atmosphere can be described as $\mathbf{M}_{\text{atm}}$. Some authors assume ensembles of spheres, and others the full normalized backscattering matrix for any shape of particle (Mishchenko and Hovenier, 1995). The latter is similar to the description in van de Hulst (1957), who describes a matrix specifically for backscattering of a cloud of asymmetrical particles with rotational symmetry, for which each particle has a mirror particle in equal numbers, and where there is no preferred orientation. Such is the case for the left matrix in Eq. (6), which can be fully described by two scalar

values $a_1$ and $a_2$. Particularly interesting are changes in polarization during the scattering events. Gimmestad (2008) introduces variable $d$ such that the scattering matrix is normalized to have an intensity of 1. For generality, we include here a gain factor, $b$. The gain factor is not stable long term, but for any given minute of data it will be constant for both channels. Under these assumptions $b$ cancels from the equations, demonstrating that the absolute scattering efficiency of the atmosphere has no effect on the measurements of depolarization.

$$\mathbf{M}_{\text{atm}} = \begin{pmatrix} a_1 & 0 & 0 & 0 \\ 0 & a_2 & 0 & 0 \\ 0 & 0 & -a_2 & 0 \\ 0 & 0 & 0 & a_1 - a_2 \end{pmatrix} = b \begin{pmatrix} 1 & 0 & 0 & 0 \\ 0 & 1-d & 0 & 0 \\ 0 & 0 & d-1 & 0 \\ 0 & 0 & 0 & 2d-1 \end{pmatrix} \tag{6}$$

In Eq. (6) the quantity $d$ is the depolarization parameter for the atmosphere above the lidar, as described in Eq. (5), and describes the extent to which the transmitted light has been depolarized by the atmosphere.

     The backscattered light already acted upon by $\mathbf{M}_{\text{atm}}$ passes through a first set of receiver optics which are shared by both the parallel and perpendicular beam paths: telescope, focus stage, beamsplitters, long-wave-pass filters, etc. We account for these

20 upstream optics together as a $4 \times 4$ matrix, $\mathbf{M}_{\text{upstream}}$, but we make no assumptions as to the values of the matrix elements:

$$\mathbf{M}_{\text{upstream}} = \begin{pmatrix} M_{00} & M_{01} & M_{02} & M_{03} \\ M_{10} & M_{11} & M_{12} & M_{13} \\ M_{20} & M_{21} & M_{22} & M_{23} \\ M_{30} & M_{31} & M_{32} & M_{33} \end{pmatrix} . \tag{7}$$

     There is also the possibility of an instrumental influence which varies with altitude $z$: Geometric overlap is such a term, and is the largest influence in CRL's height-dependent calibration vector. Hence, all height-dependent variations will be attributed


to geometric overlap, and we will subsequently call this the "overlap function", $O(z)$. The overlap varies with changes to the lidar's alignment to the sky, so calibration values must be applied to measurements made within an appropriate time window (as alignment can change slowly with laboratory temperature), and must be re-determined after every routine alignment adjustment procedure.

Next, the light passes through the Glan–Thompson prism of the Licel Polarotor. This prism acts as either a horizontal or 5  a vertical analyzing polarizer (for the parallel ($\mathbf{M}_{\parallel}$) and perpendicular ($\mathbf{M}_{\perp}$) measurement channels, respectively) depending on its orientation at a particular time. The laser beam has linear polarization in the horizontal direction. During setup, the "parallel" analyzer position was also oriented such that it can be represented as a horizontal polarizer (by aligning the parallel direction with the direction of maximum signal in a low depolarization sky).

$$\mathbf{M}_{\parallel} = \frac{1}{2}\begin{pmatrix} 1 & 1 & 0 & 0 \\ 1 & 1 & 0 & 0 \\ 0 & 0 & 0 & 0 \\ 0 & 0 & 0 & 0 \end{pmatrix} \quad \text{and} \quad \mathbf{M}_{\perp} = \frac{1}{2}\begin{pmatrix} 1 & -1 & 0 & 0 \\ -1 & 1 & 0 & 0 \\ 0 & 0 & 0 & 0 \\ 0 & 0 & 0 & 0 \end{pmatrix} \tag{8}$$

10  Finally, the light passes through the focusing lens and neutral density filter of the PMT tube, and onto the PMT itself. The lens is axially symmetric, and the neutral density filter is used at normal incidence. Therefore, the effect of these optics on any incident light will be identical regardless of incoming polarization orientation. These optics are downstream of the analyzing polarizer, so any rotation of the plane of polarization by these optics will have no impact on the signal registered by the PMT, and is therefore unimportant. The only effect of these optics is to reduce the amplitude of the signal by a constant 15  factor regardless of incoming polarization. They are well-described as a constant scalar factor. For lidars which have physically separate detectors or optical paths for the parallel and perpendicular channels, one such scalar factor will be required for each: $G_{\mathrm{PMT}\parallel}$ and $G_{\mathrm{PMT}\perp}$. These terms are identical and cancel out of the equations below for CRL, but will be included explicitly in the equations below as long as possible for generality.

These Mueller matrices combine to make an overall equation for each channel which describes the action of all optical 20  components on the light, and results in Stokes vectors $\boldsymbol{I}_{\parallel}$ (shown in full in Eq. (9)) and $\boldsymbol{I}_{\perp}$ (which differs from $\boldsymbol{I}_{\parallel}$ only by two minus signs in the polarizer matrix):

$$\boldsymbol{I}_{\parallel} = \frac{G_{\mathrm{PMT}\parallel}}{2}\begin{pmatrix} 1 & 1 & 0 & 0 \\ 1 & 1 & 0 & 0 \\ 0 & 0 & 0 & 0 \\ 0 & 0 & 0 & 0 \end{pmatrix}\begin{pmatrix} M_{00} & M_{01} & M_{02} & M_{03} \\ M_{10} & M_{11} & M_{12} & M_{13} \\ M_{20} & M_{21} & M_{22} & M_{23} \\ M_{30} & M_{31} & M_{32} & M_{33} \end{pmatrix} bO_{\parallel\perp}(z)\begin{pmatrix} 1 & 0 & 0 & 0 \\ 0 & 1-d & 0 & 0 \\ 0 & 0 & d-1 & 0 \\ 0 & 0 & 0 & 2d-1 \end{pmatrix}I_{\mathrm{laser}}\begin{pmatrix} 1 \\ 1 \\ 0 \\ 0 \end{pmatrix}. \tag{9}$$

The signal $S_{\parallel}$ measured by the lidar is the intensity element of the Stokes vector $\boldsymbol{I}_{\parallel}$:

$$S_{\parallel} = \frac{G_{\mathrm{PMT}\parallel}bO_{\parallel\perp}(z)I_{\mathrm{laser}}}{2}(M_{00} + M_{10} + (M_{01} + M_{11})(1-d)). \tag{10}$$





25  Similarly, $S_\perp$ is the intensity element of the Stokes vector $\boldsymbol{I}_\perp$:

$$S_\perp = \frac{G_{\mathrm{PMT}\perp} b O_{\|\perp}(z) I_{\mathrm{laser}}}{2}(M_{00} - M_{10} + (M_{01} - M_{11})(1 - d)). \tag{11}$$

Using the signals $S_\|$ and $S_\perp$ from above for the complete matrix description of the lidar, we solve for the depolarization parameter $d$ to learn about the atmosphere.

The simplest method for combining lidar signals $S_\|$ and $S_\perp$ into an equation to solve for the depolarization parameter comes

from creating the quantity

$$\frac{S_\| - S_\perp}{S_\| + S_\perp} = \frac{M_{10} + M_{11}(1 - d)}{M_{00} + M_{01}(1 - d)}, \tag{12}$$

realizing that $G_{\mathrm{PMT}\perp} = G_{\mathrm{PMT}\|} = G_{\mathrm{PMT}\|\perp}$ for CRL because both channels are physically the same PMTs. Solving for $d$ yields Eq. (13):

$$d = 1 - \frac{\frac{M_{10}}{M_{00}}(1 + \frac{S_\perp}{S_\|}) - (1 - \frac{S_\perp}{S_\|})}{\frac{M_{01}}{M_{00}}(1 - \frac{S_\perp}{S_\|}) - \frac{M_{11}}{M_{00}}(1 + \frac{S_\perp}{S_\|})}. \tag{13}$$

For calibration, we must determine the three instrument constants $\frac{M_{01}}{M_{00}}$, $\frac{M_{10}}{M_{00}}$, and $\frac{M_{11}}{M_{00}}$. Note that we do not require the $M_{xx}$ values individually, nor do we need to know the laser intensity.

### 3.3  Conditions under which the traditional equations are appropriate

Under the mathematical conditions $M_{01} = M_{10}$ and $M_{11} = M_{00}$, the traditional expression for $d$ (Eq. (5)) is equivalent to the more complete expression for $d$ (Eq. (13)). In the case that CRL met these conditions, it would acceptable to use Eq. (5) in

further calculations for this lidar. The matrix form of $\mathbf{M}_{\mathrm{upstream}}$ which is required for this condition describes a partial polarizer, which acts with gain $G_{\mathrm{up}\|}$ for light polarized in the parallel direction, and $G_{\mathrm{up}\perp}$ for that polarized perpendicularly. Under these simplified conditions, the relation between $k$ and $M_{xx}$ is:

$$\frac{G_{\mathrm{up}\|}}{G_{\mathrm{up}\perp}} = \frac{M_{00} + M_{01}}{M_{00} - M_{01}} = \frac{1 + \frac{M_{01}}{M_{00}}}{1 - \frac{M_{01}}{M_{00}}} = k. \tag{14}$$

Calibrations described in Sect. 5 demonstrate that the $M_{01} = M_{10}$ and $M_{11} = M_{00}$ conditions are met for CRL, allowing

Eq. (4) to be used for the calculation of depolarization ratio, and Eq. (5) for depolarization parameter.

### 4  Polarization and Depolarization generating calibration optics

Instrument calibration tests to determine $k$ and the $M_{xx}$ and $G_{xx}$ values for CRL are described in the following sections. To carry out these tests, some additional calibration optics must be temporarily added to the lidar.

1. Calibration cube polarizer: The generating polarizer is a Newport 10BC16PC.3 Pol Cube Beamsplitter, 532 nm, 25.4 mm,

Tp/Ts> 1000 : 1, a linearly polarizing cubic prism. It is placed immediately downstream of the focus stage and is rotated by



hand. The rotating mount has markings in $2°$ steps, and there is about half a step of uncertainty in either direction, hence $\pm 1°$ uncertainty in the rotation angles $\theta$.

The matrix describing the cube polarizer, allowing for an attenuation factor $G_{\text{cube}}$, is:

$$\mathbf{M}_{\text{cube}} = \frac{G_{\text{cube}}}{2} \begin{pmatrix} 1 & \cos 2\theta & \sin 2\theta & 0 \\ \cos 2\theta & \cos^2 2\theta & \frac{1}{2}\sin 4\theta & 0 \\ \sin 2\theta & \frac{1}{2}\sin 4\theta & \sin^2 2\theta & 0 \\ 0 & 0 & 0 & 0 \end{pmatrix}. \tag{15}$$

2. Glassine waxed paper depolarizer: Typically used to protect works of art, the depolarizing properties of Lineco Glassine
(Lineco Glassine Acid Free Tissue 16" $\times$ 20", 12 pack, Product number 448-1626) were found to be highly satisfactory. After one sheet the residual polarisation is less than $1\,\%$ (Polarization $= 0.009 \pm 0.006$), and two sheets in series eliminates the polarization completely, as tested by our group. The depolarizing properties were not affected by the product's exposure to damp, nor to wetting and subsequent drying out. This product was mounted in such a way as to be held relatively taut in a frame, or held gently in place by other mechanical means.

The matrix for a perfect depolarizer with an attenuation parameter $G_{\text{gl}}$ which is applicable to a real depolarizing optic (in this case, glassine waxed paper), this is:

$$\mathbf{M}_{\text{glassine}} = \frac{G_{\text{gl}}}{2} \begin{pmatrix} 1 & 0 & 0 & 0 \\ 0 & 0 & 0 & 0 \\ 0 & 0 & 0 & 0 \\ 0 & 0 & 0 & 0 \end{pmatrix} \tag{16}$$

3. Industrial kitchen grade waxed paper: The depolarizing properties of this product were used for CRL calibrations before we were aware of glassine, which turned out to be the superior material. The depolarizing properties of various brands of waxed 15 paper, and from batch to batch within a particular brand, vary widely; verification for each application is advisable.

## 5 Calibrations to determine whether traditional equations are acceptable

Introducing polarized lamp or laser light to the detector provides the calibration values which indicate whether the simple equations are satisfactory for CRL, by testing whether the conditions $M_{01} = M_{10}$ and $M_{11} = M_{00}$ are satisfied.

### 5.1 Physical setup of the rotating polarizer test used at CRL

A light source (lamp or backscattered laser) is directed through two layers of glassine depolarizer sheet into a polarization-generating optic and through to the lidar's receiver system. The polarization generator is able to be rotated through various angles $\theta$ with respect to the plane of polarization of the parallel channel, and signals are measured in the parallel and perpendicular channels as a function of this angle (Fig. 2). The glassine ensures that unpolarized light enters the polarizer, and thus we would begin with equal numbers of photons exiting the polarizer regardless of its orientation.



Ideally, as many optics as possible are included after the polarization generating calibration optic, so that the contributions of as many "upstream optics" as possible are included in the Mueller Matrix $\mathbf{M}_{\text{upstream}}$ during calibration. In practice, this is difficult to accomplish for practical reasons at CRL. A light source which illuminates the entire 1 m CRL telescope is available if we employ backscattered laser light. A 1 m depolarizing optic to initially depolarize the all the backscattered light received at the roof hatch level is also available (glassine waxed paper sheets). The problem is that no feasible polarizing

optic had the required properties: a 1 m diameter circle, which could be held completely flat, which could survive the harsh outdoor conditions of Arctic winter, which could be easily and repeatably rotated to the appropriate orientation, and which had sufficient optical polarization quality. A variety of setups using sheet polarizers were attempted to overcome these problems, and none produced satisfactory results. Neither did the use of a smaller 25 cm diameter sheet polarizer installed in a smaller aperture above the lidar's telescope, with the rest of the entrance to the primary mirror masked. Repeating the test with the

sheet polarizer held between the telescope's tertiary mirror and the focus stage worked better, but still relied on a suboptimal optical quality sheet polarizer.

By sacrificing the inclusion of both the lidar's telescope and focus stage in the calibration, the rotating polarizer test becomes possible at CRL. By the time the light reaches the entrance to the polychromator, the received light beam, originally 1 m diameter, is focused small enough to allow the use of the 25 mm polarizing cube beamsplitter of high optical quality described

in Sect. 4. It can be rotated precisely and is stably mounted on a kinematic rotation mount on a 2" beam tube which leads into the polychromator from the focus stage (location label 7 in Fig. 1).

Because the telescope and focus stage are being omitted in the test, there is no advantage to using lidar returns as the light source; a current-stabilized constant lamp source provides more signal with better control of the experimental setup. It also does not rely on specific atmospheric conditions. The lamp is installed on the telescope frame such that it shines through a glassine

depolarizing sheet held taut in a frame of foamcore between the tertiary telescope mirror and the focus stage (location label 8 in Fig. 1). The resulting unpolarized light is sent through the focus stage, then through the cube polarizer, which produces the linearly polarized light which is sent through the rest of the polychromator, including the rotating Polarotor polarizer, and into the PMT.

Omitting the first optics in the detector chain means that this test does not give us a whole-system understanding, although it

*does* allow us to say with certainty whether the downstream optics are contributing any non-simple-gain effects to the signals.

We pose the question, "If we consider the optics and detector starting after the focus stage, can we use the simplified Eq. (4) and Eq. (5) to find the calibration constant, and then to determine depolarization ratio and depolarization parameter?" This is answered in the remainder of Sect. 5. If yes, we can then ask "What is the best estimate for a polarization calibration constant which represents the entire system?", which is addressed in Sect. 6.

**5.2   Matrix description and results of polarized light calibration**

Linearly polarized light is introduced to the receiver. The orientation of the plane of polarization of this light is described as angle $\theta$ with respect to the plane of polarization of the parallel channel. Then measurements are made in the parallel and





perpendicular channels. In this scenario, the final Stokes vectors are:

$$I_{\parallel\theta} = \frac{G_{\mathrm{PMT}\parallel}}{2} \begin{pmatrix} 1\,1\,0\,0 \\ 1\,1\,0\,0 \\ 0\,0\,0\,0 \\ 0\,0\,0\,0 \end{pmatrix} \begin{pmatrix} M_{00}\ M_{01}\ M_{02}\ M_{03} \\ M_{10}\ M_{11}\ M_{12}\ M_{13} \\ M_{20}\ M_{21}\ M_{22}\ M_{23} \\ M_{30}\ M_{31}\ M_{32}\ M_{33} \end{pmatrix} \frac{G_{\mathrm{cube}}}{2} \begin{pmatrix} 1 & \cos 2\theta & \sin 2\theta & 0 \\ \cos 2\theta & \cos^2 2\theta & \frac{1}{2}\sin 4\theta & 0 \\ \sin 2\theta & \frac{1}{2}\sin 4\theta & \sin^2 2\theta & 0 \\ 0 & 0 & 0 & 0 \end{pmatrix} G_{\mathrm{gl}} \begin{pmatrix} 1 \\ 0 \\ 0 \\ 0 \end{pmatrix} I_{\mathrm{lamp}} \tag{17}$$

$$I_{\perp\theta} = \frac{G_{\mathrm{PMT}\perp}}{2} \begin{pmatrix} 1\,\text{-}1\,0\,0 \\ \text{-}1\,1\,0\,0 \\ 0\,0\,0\,0 \\ 0\,0\,0\,0 \end{pmatrix} \begin{pmatrix} M_{00}\ M_{01}\ M_{02}\ M_{03} \\ M_{10}\ M_{11}\ M_{12}\ M_{13} \\ M_{20}\ M_{21}\ M_{22}\ M_{23} \\ M_{30}\ M_{31}\ M_{32}\ M_{33} \end{pmatrix} \frac{G_{\mathrm{cube}}}{2} \begin{pmatrix} 1 & \cos 2\theta & \sin 2\theta & 0 \\ \cos 2\theta & \cos^2 2\theta & \frac{1}{2}\sin 4\theta & 0 \\ \sin 2\theta & \frac{1}{2}\sin 4\theta & \sin^2 2\theta & 0 \\ 0 & 0 & 0 & 0 \end{pmatrix} G_{\mathrm{gl}} \begin{pmatrix} 1 \\ 0 \\ 0 \\ 0 \end{pmatrix} I_{\mathrm{lamp}} \tag{18}$$

With corresponding signals:

$$S_{\parallel\theta} = G_{\mathrm{cube}}G_{\mathrm{PMT}\parallel}G_{\mathrm{gl}}\frac{I_{\mathrm{lamp}}}{4}\left(M_{00} + M_{10} + (M_{01} + M_{11})\cos 2\theta + (M_{02} + M_{12})\sin 2\theta\right) \tag{19}$$

$$S_{\perp\theta} = G_{\mathrm{cube}}G_{\mathrm{PMT}\perp}G_{\mathrm{gl}}\frac{I_{\mathrm{lamp}}}{4}\left(M_{00} - M_{10} + (M_{01} - M_{11})\cos 2\theta + (M_{02} - M_{12})\sin 2\theta\right). \tag{20}$$

There are very similar equations for the case in which we use backscattered laser light rather than lamp light.

The results of such a test from 5 March 2014 are plotted in Fig. 2. This plot shows the signals in the parallel channel (blue points) and in the perpendicular channel (red points) as a function of cube polarizer angle $\theta$. The cube polarizer was initially placed at an arbitrary angle to ensure that photons were visible in each channel. It was then rotated through a number of steps, spending several minutes at each angle. In total, it was rotated through just more than one full rotation, or $2\pi$ radians. The absolute angles were determined in post-processing, such that the maximum in the parallel channel is $\theta = 0$. All measurements for each angle $\theta$ have been combined for this plot. Photocounts are indicated in units of "photons per time bin per altitude bin", at a resolution of $1\,\mathrm{min} \times 7.5\,\mathrm{m}$ for each bin. There is approximately a $2°$ or 0.035 radian uncertainty in the angles when doing this calibration. Note the different scales for each: The overall signals in $S_{\parallel}$ far exceed the overall signals $S_{\perp}$.

We could attempt to estimate all 7 unknown terms in Eq. (19) and Eq. (20) by allowing them as free parameters in a fit to these signals, but for CRL, there is a better way: Signal values at some diagnostic angles simplify the equations a great deal by constraining certain calibration constants.

### 5.2.1   First constraint: $M_{02} = 0$ and $M_{12} = 0$

The signal equations, Eq. (19) and Eq. (20), are simplified a great deal if $M_{02} = 0$ and $M_{12} = 0$. This is the case if there is symmetry about $\theta = \pi/2$ in the curves of both of the signals in Fig. 2, and in particular if the signals at $\theta = \frac{\pi}{4}$ equal those at $\theta = \frac{3\pi}{4}$ for both the parallel and the perpendicular. The results of these measurements are given in Table 2.

For the parallel channel, the signals at these angles are:

$$S_{\parallel\theta = \frac{\pi}{4}} = G_{\mathrm{cube}}G_{\mathrm{PMT}\parallel}G_{\mathrm{gl}}\frac{I_{\mathrm{lamp}}}{4}\left(M_{00} + M_{10} + (M_{02} + M_{12})\right) \tag{21}$$

$$S_{\parallel\theta = \frac{3\pi}{4}} = G_{\mathrm{cube}}G_{\mathrm{PMT}\parallel}G_{\mathrm{gl}}\frac{I_{\mathrm{lamp}}}{4}\left(M_{00} + M_{10} - (M_{02} + M_{12})\right). \tag{22}$$





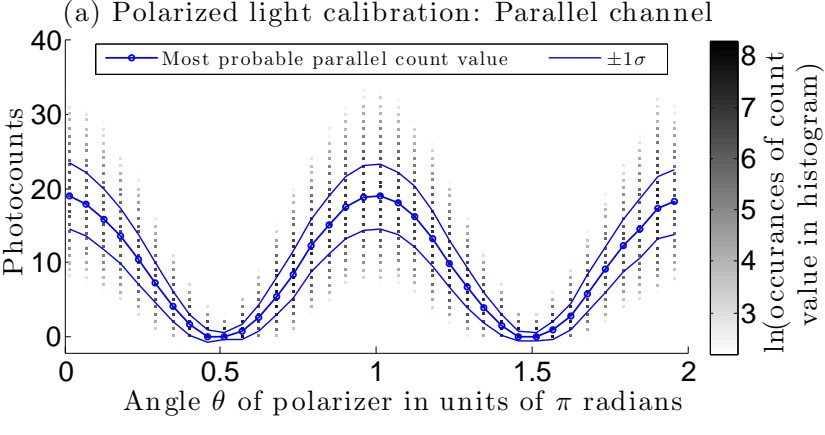

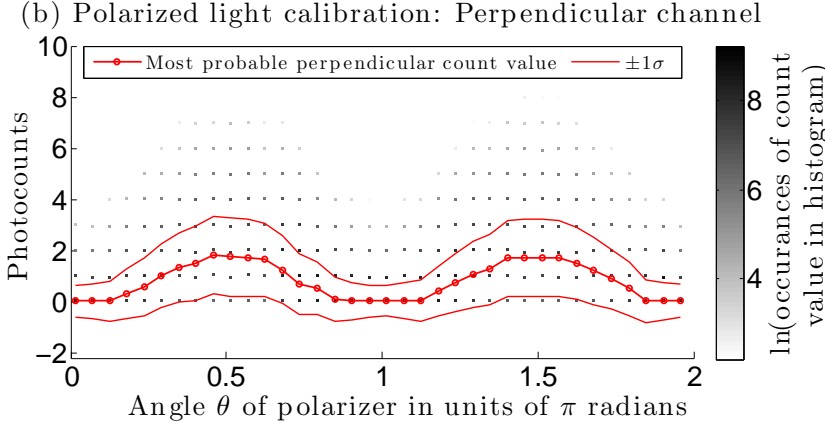

**Figure 2.** Polarized calibration measurements as a function of incident light polarization angle. At angle $\theta = 0$, the calibration cube polarizer is aligned completely with the parallel measurement channel. At $\theta = 0.5\pi$, it is aligned completely with the perpendicular channel. The greyscale points indicate the extent of scatter in the measurements for each channel. The colourbar indicates the natural logarithm of the number of data points at each location, which is the result of producing a histogram for each angle $\theta$. The truncated distribution of points (because only positive photons may be measured) is evident. Each histogram was fit with a 6th order polynomial in order to determine the location of the peak photocount rate. This often differed significantly from the mean of the photocount values at that angle, and was taken to be the most probable value. The most probable photocount values are given by the blue points (parallel channel; panel (a)) and red points (perpendicular channel; panel (b)). The thick blue and red lines trace these data points as a visual aid; they are not fit lines. The thin blue and red lines give the $\pm 1\sigma$ standard deviation of the point distribution at each angle. Note the different y-axis limits for each plot; the parallel count rates are far larger than the perpendicular count rates.

If our measurements are symmetric, with $S_{\parallel\theta=\frac{\pi}{4}} = S_{\parallel\theta=\frac{3\pi}{4}}$, then $M_{02} = -M_{12}$.





For the perpendicular channel,

$$S_{\perp\theta=\frac{\pi}{4}} = G_{\text{cube}}G_{\text{PMT}\perp}G_{\text{gl}}\frac{I_{\text{lamp}}}{4}\left(M_{00} - M_{10} + (M_{02} - M_{12})\right) \tag{23}$$

$$S_{\perp\theta=\frac{3\pi}{4}} = G_{\text{cube}}G_{\text{PMT}\perp}G_{\text{gl}}\frac{I_{\text{lamp}}}{4}\left(M_{00} - M_{10} - (M_{02} - M_{12})\right). \tag{24}$$

If our measurements are symmetric, with $S_{\perp\theta=\frac{\pi}{4}} = S_{\perp\theta=\frac{3\pi}{4}}$, then $M_{02} = M_{12}$.

Because the values for $M_{02}$ and $M_{12}$ must not change and are in common for parallel and perpendicular, then if *both* parallel

and perpendicular are symmetric, both $M_{02} = -M_{12}$ and $M_{02} = M_{12}$ must be true simultaneously, so $M_{02} = -M_{02}$ and

$M_{12} = -M_{12}$. Thus, $M_{02} = 0$ and $M_{12} = 0$.

This simplifies the calibration signal equations to:

$$S_{\|\theta} = G_{\text{cube}}G_{\text{PMT}\|}G_{\text{gl}}\frac{I_{\text{lamp}}}{4}\left(M_{00} + M_{10} + (M_{01} + M_{11})\cos 2\theta\right) \tag{25}$$

$$S_{\perp\theta} = G_{\text{cube}}G_{\text{PMT}\perp}G_{\text{gl}}\frac{I_{\text{lamp}}}{4}\left(M_{00} - M_{10} + (M_{01} - M_{11})\cos 2\theta\right) \tag{26}$$

Each channel's signal values at $\theta = \frac{\pi}{4}$ and $\theta = \frac{3\pi}{4}$ are equal within their respective uncertainties (Table 2). The mean signal

values are $S_\| = 9 \pm 1$ for parallel, and $S_\perp = 0.8 \pm 0.5$ for perpendicular, with uncertainties calculated using standard error

propagation. Therefore, both channels are symmetric about $\theta = \pi/2$, and the first simplification may be used. Hence, we see

that for CRL, $M_{02} = 0$ and $M_{12} = 0$ are valid for the conditions of this test.

At this point, it is possible to combine Eq. (25) and Eq. (26) into one equation which includes both signals and only the

desired calibration constants $\frac{M_{10}}{M_{00}}$, $\frac{M_{01}}{M_{00}}$, and $\frac{M_{11}}{M_{00}}$ which remain to be determined by a fit to the resulting curve:

$$\frac{S_{\|\theta} - S_{\perp\theta}}{S_{\|\theta} + S_{\perp\theta}} = \frac{\frac{M_{10}}{M_{00}} + \frac{M_{11}}{M_{00}}\cos 2\theta}{1 + \frac{M_{01}}{M_{00}}\cos 2\theta}. \tag{27}$$

Again, there is not a unique solution if all three calibration coefficients are free parameters in the fit. It is preferable to see

whether the signal measurements indicate any further constraints.

### 5.2.2 Second constraint: $M_{10} = M_{01}$ and $M_{00} = M_{11}$

Further simplifications may be made if the parallel and perpendicular channel signals each go to zero at their respective minima.

For parallel, this is at integer multiples of $\theta = \frac{\pi}{2}$, where $\cos 2\theta = -1$. For perpendicular, this is at $\theta = 0$ and at integer multiples

of $\theta = \pi$, where $\cos 2\theta = 1$. If this is the case, then

$$S_{\|\theta} = 0 = G_{\text{cube}}G_{\text{PMT}\|}G_{\text{gl}}\frac{I_{\text{lamp}}}{4}\left(M_{00} + M_{10} + (M_{01} + M_{11})(-1)\right) \tag{28}$$

$$0 = M_{00} + M_{10} - M_{01} - M_{11}, \tag{29}$$



and

$$S_{\perp\theta} = 0 = G_{\text{cube}}G_{\text{PMT}\perp}G_{\text{gl}}\frac{I_{\text{lamp}}}{4}\left(M_{00} - M_{10} + (M_{01} - M_{11})(1)\right) \tag{30}$$

$$0 = M_{00} - M_{10} + M_{01} - M_{11}. \tag{31}$$

In the case that *both* signals go to zero at their respective locations, Eq. (29) and Eq. (31) are equal, so $M_{10} = M_{01}$ and $M_{00} = M_{11}$.

This leaves the calibration signals as:

$$S_{\parallel\theta} = G_{\text{cube}}G_{\text{PMT}\parallel}G_{\text{gl}}\frac{I_{\text{lamp}}}{4}M_{00}(1 + \frac{M_{10}}{M_{00}})(1 + \cos 2\theta) \tag{32}$$

$$S_{\perp\theta} = G_{\text{cube}}G_{\text{PMT}\perp}G_{\text{gl}}\frac{I_{\text{lamp}}}{4}M_{00}(1 - \frac{M_{10}}{M_{00}})(1 - \cos 2\theta), \tag{33}$$

in which $\frac{M_{10}}{M_{00}}$ is the only calibration constant needing to be determined. Note that we do not need to know the value of $M_{00}$ or $M_{10}$ individually for calculating depolarization parameter $d$.

For CRL, histograms of the numbers of counts at each angle show that the most probable value for the perpendicular count rate $S_{\perp\theta=0}$ is zero photons per measurement interval. Similar histograms for the parallel channel show that it, too, goes to zero at its minimum (at $\theta = \pi/2$ rad). Fitted histogram values were used rather than means. The raw lidar photon counting data does not report any values less than zero counts, and noise will artificially increase the total rate. When examining signals larger than zero, the noise takes a Gaussian shape around the mean signal value. For situations in which the true signal is zero, a mean

of the measured signal will be reported as a larger value, thus not being indicative of the most probable photon counting result.

As the count rates do indeed go to zero at their respective minima, this second constraint is also appropriate for CRL: $M_{10} = M_{01}$ and $M_{00} = M_{11}$.

### 5.3 First result of rotating polarizer test: Traditional equations are appropriate for CRL

The value of $\frac{M_{10}}{M_{00}}$ can now be calculated from the calibration data (and this is done in Sect. 5.4), but this is not the most

important result from the polarized calibration test. Rather, what matters is that CRL's polychromator optics are acting only as a partial polarizer. $M_{01} = M_{10}$ and $M_{11} = M_{00}$ were identified in Sect. 3.3 as the necessary conditions for which the simpler traditional versions of the depolarization equations Eq. (4) and Eq. (5) are acceptable for CRL. These conditions were fulfilled for CRL. The calibration approach and equations for $d$ and $\delta$ used by others in the community is appropriate, despite CRL's many optics between the sky and the analyzing polarizer for the depolarization channel. This assumes that the telescope does

not contribute to these quantities in a significant way. This result is reasonable, as the reflectivity of all telescope mirrors are high.





## 5.4 Determining $\frac{M_{01}}{M_{00}}$ from polarized calibration

The results from the polarized light calibration may be extended further to calculate a preliminary value of $\frac{M_{10}}{M_{00}}$. Combining Eq. (32) and Eq. (33), we can solve directly for $\frac{M_{10}}{M_{00}}$, the only remaining calibration constant:

$$\frac{M_{10}}{M_{00}} = \frac{\cos 2\theta - \left(\frac{S_{\parallel\theta} - S_{\perp\theta}}{S_{\parallel\theta} + S_{\perp\theta}}\right)}{\cos 2\theta \left(\frac{S_{\parallel\theta} - S_{\perp\theta}}{S_{\parallel\theta} + S_{\perp\theta}}\right) - 1}. \tag{34}$$

In such a calculation, for measurements made at angles $\theta$ where either signal goes to zero, the result for $\frac{M_{10}}{M_{00}}$ is a zero-
divided-by-zero fraction. Thus, one must exclude such calibration angles.

This calculation was carried out for the test in Fig. 2. A histogram was made of the calculated $\frac{M_{10}}{M_{00}}$ values, with a peak at $\frac{M_{10}}{M_{00}} = 0.77 \pm 0.18$. However, this value has limitations, and is not a good representative value for the CRL system: This value is not representative of the whole receiver; rather the of the polychromator only. It includes no effects of the telescope or focus stage.

Because of these limitations, it is preferable to determine $\frac{M_{10}}{M_{00}}$, and also $k$, using an unpolarized light test, as demonstrated in the following section (Sect. 6). This determination of the constant employs well-established techniques and has lower uncertainty. Furthermore, it can include all lidar optics. Having measured the partial-polarizer-like form of the upstream optics Mueller Matrix using the polarized calibration test, we can proceed with confidence in the tests in the following section.

## 6   Determining $k$ and $\frac{M_{10}}{M_{00}}$ with traditional equations

The validity of the expressions in Eq. (5) (and therefore Eq. (4) also) has already been demonstrated for CRL in Sect. 5. Therefore these expressions can be used to determine $k$ and/or $\frac{M_{10}}{M_{00}}$ via calibrations in which $d = \delta = 1$. This mimics fully depolarized light returning from the sky. Two different methods were used to arrange a $d = \delta = 1$ calibration setup. The first method forces backscattered lidar light go through a depolarizing sheet of glassine waxed paper before being measured (Sect. 6.2). The second method involves shining a lamp at the detector through a depolarizing sheet of glassine (Sect. 7). Either
of these methods is preferable to using sky light alone without ensuring its total depolarization as it enters the lab. Even in atmospheric conditions which are thought to be depolarizing (e.g. clouds in which multiple scattering is expected, or ice clouds for which complete depolarization is expected), complete depolarization at all altitudes for the duration of the measurement cannot be ensured.

### 6.1   Mueller Matrix development of the calibration expressions for $\frac{M_{10}}{M_{00}}$ and $k$

The matrix equation for the intensity reaching the parallel channel, using the laser as the light source and a perfect depolarizer with attenuation parameter $G_{\mathrm{gl}}$ in front of the receiver is:





$$I_{\parallel} = \frac{G_{\mathrm{PMT}\parallel\perp}}{2} \begin{pmatrix} 1\,1\,0\,0 \\ 1\,1\,0\,0 \\ 0\,0\,0\,0 \\ 0\,0\,0\,0 \end{pmatrix} \begin{pmatrix} M_{00}\,M_{01}\,M_{02}\,M_{03} \\ M_{10}\,M_{11}\,M_{12}\,M_{13} \\ M_{20}\,M_{21}\,M_{22}\,M_{23} \\ M_{30}\,M_{31}\,M_{32}\,M_{33} \end{pmatrix} \frac{G_{\mathrm{gl}}}{2} \begin{pmatrix} 1\,0\,0\,0 \\ 0\,0\,0\,0 \\ 0\,0\,0\,0 \\ 0\,0\,0\,0 \end{pmatrix} bO_{\parallel\perp}(z) \begin{pmatrix} 1 & 0 & 0 & 0 \\ 0 & 1-d & 0 & 0 \\ 0 & 0 & d-1 & 0 \\ 0 & 0 & 0 & 2d-1 \end{pmatrix} I_{\mathrm{laser}} \begin{pmatrix} 1 \\ 1 \\ 0 \\ 0 \end{pmatrix}. \tag{35}$$

Thus, the signals in each channel are:

$$S_{\parallel\ d=1} = \frac{G_{\mathrm{PMT}\parallel\perp} G_{\mathrm{gl}}}{2} bO_{\parallel\perp}(z) I_{\mathrm{laser}} (M_{00} + M_{10}) \tag{36}$$

$$S_{\perp\ d=1} = \frac{G_{\mathrm{PMT}\parallel\perp} G_{\mathrm{gl}}}{2} bO_{\parallel\perp}(z) I_{\mathrm{laser}} (M_{00} - M_{10}). \tag{37}$$

The signals from the parallel and perpendicular channel are combined so that we can solve for $\frac{M_{10}}{M_{00}}$ with as many of the unknown factors cancelling out as possible:

$$\frac{S_{\parallel\ d=1} + S_{\perp\ d=1}}{S_{\parallel\ d=1}} = \frac{2M_{00}}{M_{00} + M_{10}} \tag{38}$$

$$\frac{M_{10}}{M_{00}} = \frac{2}{\frac{S_{\perp\ d=1}}{S_{\parallel\ d=1}} + 1} - 1. \tag{39}$$

To calculate $k$, the equation is:

$$k = \frac{S_{\parallel\ d=1}}{S_{\perp\ d=1}}. \tag{40}$$

Note that these equations for $\frac{M_{10}}{M_{00}}$ and $k$ work equally well for the case in which we use a lamp to illuminate the lidar as the several differences in the initial matrix equation cancel out in any case: There is no overlap function, no atmospheric matrix, and we use $I_{\mathrm{lamp}}$ instead of $I_{\mathrm{laser}}$, which also cancels out of the final calibration equations.

The constant $\frac{M_{10}}{M_{00}}$ and $k$ can be calculated from one another. $\frac{M_{10}}{M_{00}}$ tends to be more "forgiving" in terms of accuracy than $k$ does. A small percentage error in $\frac{M_{10}}{M_{00}}$ will yield a larger percent error in $k$. Thus, one must take care when making a selection of which to use despite the ease with which one may convert between them in the absence of estimates of their uncertainties.

## 6.2 Physical setup of unpolarized laser calibration to determine $\frac{M_{10}}{M_{00}}$ and $k$

To include all optics, the depolarized light should be introduced above the roof hatch window. Note that unlike the polarized calibration, we do have access to depolarizing optics which are practical to cover a $1\,\mathrm{m}$ diameter roof hatch window. However, as the lamp is not bright enough when placed in this location, backscattered laser light is better used to test this location.

In this setup, we obtain the depolarized light by running the lidar as usual, with the laser transmitted to the sky and scattered back to the lidar, but we interrupt the optical path of the receiver with a depolarizing sheet (glassine) before the backscattered light enters through the roof window of the detector.

Using a flexible material like glassine was important in the Arctic winter. When the roof hatch open is open, the glassine sheet is exposed to wind, blowing snow, cold temperatures, and any humidity that may be in the air. Several 16" x 20" sheets



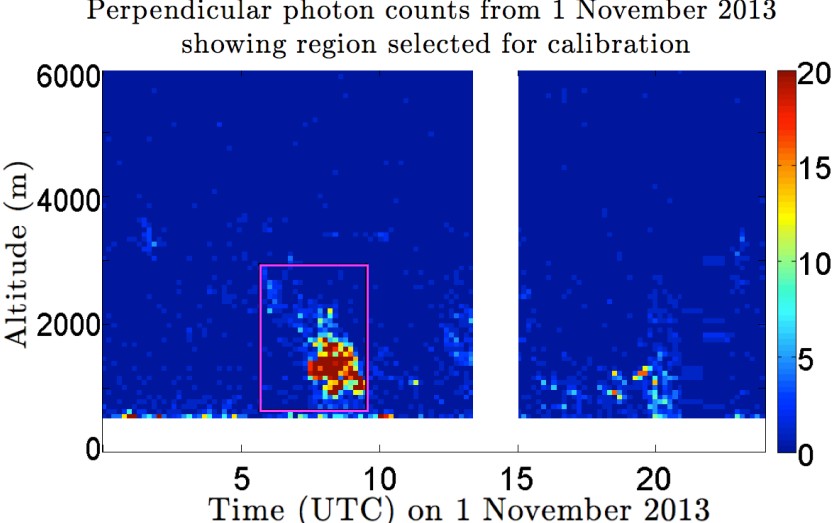

**Figure 3.** Colour plot of perpendicular channel coadded and corrected photocounts. Magenta box below 3000 m between 05:00 UTC and 10:00 UTC indicates the region of interest to be used in the calibration analysis.

were taped together with opaque black masking tape to create a larger sheet which could cover the 1 m diameter lidar roof window. To keep the photon count rates as high as possible during the test, only a single layer of glassine was used, although using two sheets in series ensures more complete depolarization. A 10 cm diameter circle was removed from the centre of the sheet so that the glassine would not obstruct the laser exit window. Glassine tears easily once its edge is compromised, so it was advantageous to pre-tape any line which was to be cut, and then to cut through the taped line. The sheet was placed over the roof window. A foam wedge with a circular hole was snugged down over the glassine sheet around the laser window such that the sheet would be secured and not blow into the laser beam. The four corners of the sheet were held down to the metal surface of the roof around the window, slightly below the level of the window's frame, pulling the glassine very gently taut over the window. Two 1x6 wooden planks with "feet" on each end provided sufficient tension, and these were wedged into place with foam.

## 6.3 Measurements for unpolarized laser calibration, using example from 1 November 2013

This test was carried out for over 22.8 h, between 22:40 UTC on 1 November 2013 and 21:00 UTC on 2 November 2013. Figure 3 is a plot of the perpendicular channel background-corrected photocount profiles for the entire calibration measurement, coadded with 10 x 10 binning to a resolution of 5 min × 37.5 m. Data from altitudes below 500 m is routinely rejected from CRL processing because of the differential geometric ovarlap function of CRL, and has been indicated here in white.

An appropriate time-altitude region of interest must be selected for the calibration. Regions with high-backscatter features, such as clouds, are desirable. The magenta box in Fig. 3 below 3000 m, between 05:00 UTC and 10:00 UTC shows the region





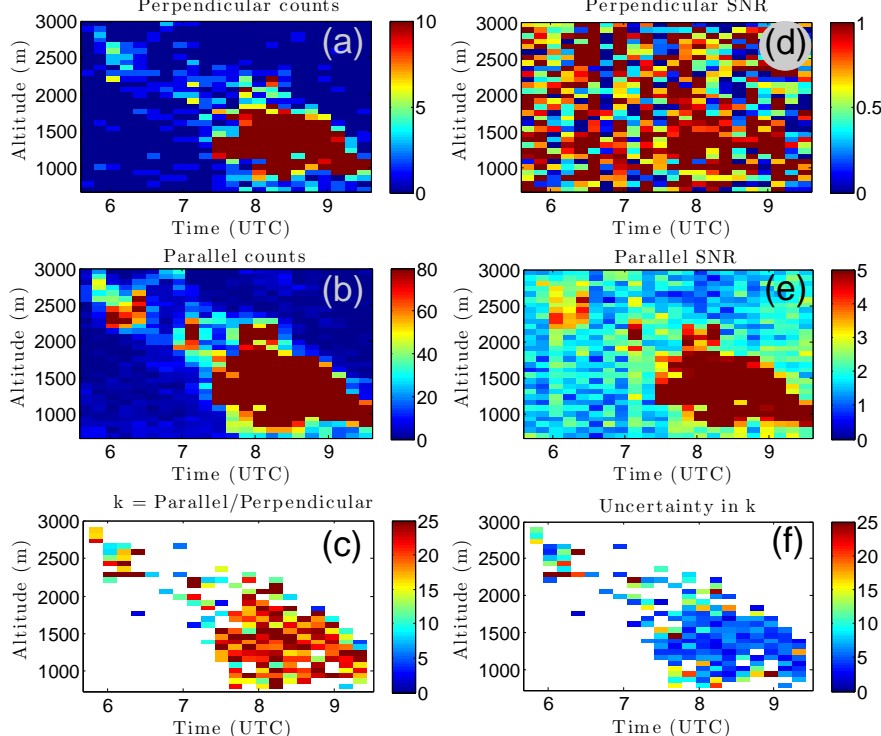

**Figure 4.** Parallel and perpendicular channel corrected photocounts within the calibration region of interest, with their associated signal-to-noise ratios (SNR) plotted to the right. Bottom left panel shows the ratio of the counts for each time-altitude point ($k$), with the associated uncertainties in $k$ shown at bottom right.

of interest selected for the calibration calculations. This region encompasses a cloud which remained between 1 and 3 km altitude over the lidar for several hours, as determined by contemporaneous measurements with the Millimetre Cloud Radar and 532 nm Visible Rayleigh Elastic CRL lidar channel measurements. Analysis for calibration was attempted on the whole data set as well as just the region of interest which contains the cloud. It was repeated for a variety of coadding times and altitude resolutions. All resolutions provided similar results.

5    Regions with fewer than one photon per time-altitude bin in either channel, and/or signal-to-noise ratios below 0.1 for perpendicular, and 1.5 for parallel, are rejected from the calculation. Because of the attenuation from the depolarizing sheet, many areas have zero counts in the perpendicular channel, even at low altitudes. The top left (a) and centre left (b) panels of Fig. 4 plot the parallel and perpendicular photon counts from the region of interest with their respective signal-to-noise ratios plotted to the right in panels (d) and (e).

10   Once the measurements have been quality controlled, there are several approaches for determining $k$.





1. Calculate a $k$ value individually for each time-altitude bin (Fig. 4, panel (c), with associated uncertainty in panel (f)), and then combine $k$ values into an overall mean constant. This method gives values of $k = 20.2 \pm 0.6$. $\frac{M_{10}}{M_{00}} = 0.89 \pm 0.05$ is similarly calculated from a mean of individual values of $\frac{S_\perp}{S_\parallel}$.

2. Calculate a summed or mean count value for the whole calibration time-altitude space in each of the two channels, and then take the ratio of these to calculate one $k$ representative of the whole region. The result using this method $k = 21.0 \pm 0.2$, and $\frac{M_{10}}{M_{00}} = 0.910 \pm 0.002$.

The two methods give values which are equal to within their uncertainties. As the uncertainty is smaller using the second approach, this method is selected.

The overall calibration factors for CRL are therefore $\frac{M_{10}}{M_{00}} = 0.910 \pm 0.002$ and $k = 21.0 \pm 0.2$, calculated using all detector optics. This is quite different from the value of $k = 1$ which would be expected for an ideal depolarization lidar in which no upstream optics were interfering with the polarization of returned light.

## 7   Determining contributions of individual optics

To determine which receiver optics contribute most to CRL's high value of $k$, further tests were carried out. These tests determined that the Visible Long Wave Pass optic is by far the largest contributor.

### 7.1   Physical setup of unpolarized lamp calibration

A lamp was placed directly upstream of the focus stage, similar to the setup of Sect. 5.1. A depolarizing sheet was installed at various locations within the detector to ensure that all light proceeding from that point was completely unpolarized. This test started with the depolarizing sheet directly before the polarotor. It was moved sequentially upstream, placed between any two optics where there was room to safely insert it, up to and including right in front of the lamp, upstream of the focus stage (Fig. 1, marked as numbers from 1 through 8). Industrial kitchen grade waxed paper was used for this test, mounted between two frames of foam core to keep it rigid. Measurements were made with the polarotor in operation as usual. Mean values of parallel and perpendicular counts were used for each calculation of $k$ and $\frac{M_{10}}{M_{00}}$.

### 7.2   Results: Lamp test $\frac{M_{10}}{M_{00}}$ and $k$, setting $d = 1$

The lamp tests were carried out over a 2 h period on 1 April, 2013. The results are listed as Tests 1 through 8 in Table 1. The laser calibration test result from Sect. 6.3 is listed as Test 9 in this table, for comparison purposes.

The tests indicate which of our optics are contributing most to the large overall calibration factor of the system. The first test location (Test 1) is as far downstream as it was possible to begin. All subsequent optics (interference filter, focus lens, etc) are in a closed beamtube where it is not feasible to insert a depolarizer. The measurements here showed equal amounts of light in the parallel and perpendicular channels, and $k = 1$. This indicates that the collection of optics in this beamtube, and the PMT,





are not contributing to the value of $k = 21$ measured for the whole lidar. The "$k$" described for CRL must therefore include optics in the system, and not only the gain of the PMT.

Subsequent tests moving upstream each time indicate that most of the optics in the polychromator are indeed partially polarizing the returned lidar beam, some favouring attenuation of parallel-polarized light (decrease in $k$ and $\frac{M_{10}}{M_{00}}$), and others attenuation of perpendicular-polarized light (increase in $k$ and $\frac{M_{10}}{M_{00}}$).

The largest contributor to the overall $k$ value is the VLWP dichroic (compare Test 3 to Test 5). Its individual contribution is $k_{\text{VLWP}} = 6$. Other large contributors are the lenses at the entrance to the polychomator ($k_{\text{poly entrance}} = 2.2$; compare Test 6 to Test 7) and the telescope and roof window ($k_{\text{telescope+window}} = 3.12$; compare Test 8 to Test 9).

The unpolarized lamp tests of Test 7 at the entrance to the polychromator, with $\frac{M_{10}}{M_{00}} = 0.719 \pm 0.001$, were made at the same location as the polarized calibration tests of Sect. 5.2, with $\frac{M_{10}}{M_{00}} = 0.77 \pm 0.18$. These values are, as expected, equal to within

their uncertainties. This provides a link between the two calibration methods.

Tests 7 and 8 took place on either side of the focus stage. Their $\frac{M_{10}}{M_{00}}$ and $k$ results change only by a small amount. Therefore, the focus stage does not contribute significantly to the whole-lidar values.

## 7.3   Discussion of results from unpolarized lamp tests

The tests in Sect. 7.2 are important for CRL future planning. First, they indicate that we *must* calibrate for $k$ by placing the

depolarizing optic at the beginning of the optical chain. Many lidar groups choose to use calibration lamps part way through their system, rather than using a lamp which scans or is projected over the whole entrance aperture at the first optic of the system. (One notable exception to this trend is the lidar group at Howard University led by Prof. Venable. See, for example Venable et al. (2011), with a mapping lamp applied to a water vapour lidar. CRL unfortunately does not have the capacity to install such a system at this time). Tests become easier to do as one moves the optic downstream in the detector for several

reasons: Any optic placed at the beginning of the detector chain must be as large as the first optic itself. In the case of CRL, this means a circle with diameter $1\,\text{m}$. Optics of this size are expensive and unwieldy, and are sometimes impossible to obtain. Any optic placed at the beginning of the detector chain will necessarily be outdoors, and will be exposed to the elements. In Eureka this includes temperatures colder than -50° C, significant wind, blowing snow, and working on a roof. If a lamp is used for illumination, power is also needed on the roof, which is inconvenient. Going downstream brings the optic inside, and

makes the required optics smaller. Any optics placed between the telescope and the focus stage must be about $25\,\text{cm}^2$. Optics after the focus stage may be as small as $25\,\text{mm}^2$. Smaller optics are easier to rotate in a controlled manner (e.g. for polarizer calibrations). At what cost to the calibration and science do these practical advantages come? For CRL, $k$ changes by a factor of 3.4 between the entrance to the polychromator and the entrance to the entire system; the more convenient calibration is insufficient. A second use for these test measurements is that they allow us to see which optics would be most advantageous to

remove or upgrade the next time we change optics in the lidar. Naturally, those such as the VLWP filter, which contributes the most to the reduction of the perpendicular signal, would be most advantageous to change.





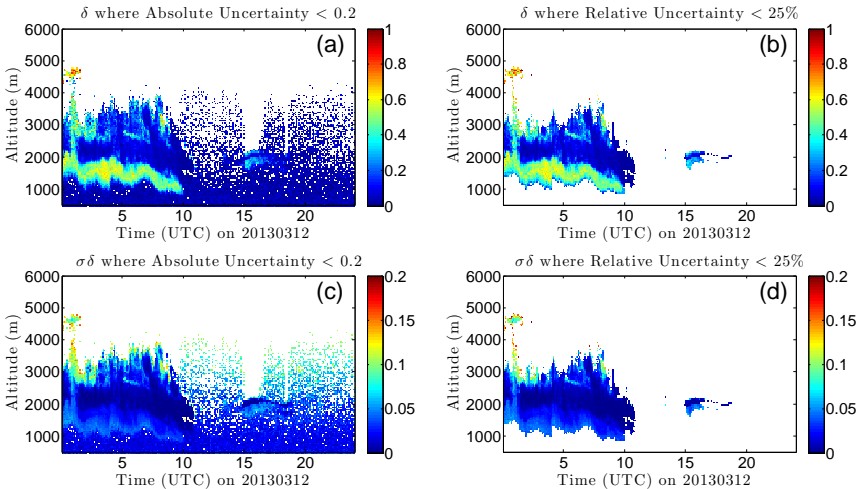

**Figure 5.** Atmospheric depolarization ratio measurements from 12 March 2013, with associated absolute uncertainties in units of depolarization ratio. In the left panels of $\delta$ (panel (a)) and $\sigma\delta$ (panel (c)), all data points with absolute uncertainty greater than 0.2 have been eliminated. In the right panels of $\delta$ (panel (b)) and $\sigma\delta$ (panel (d)) , points with relative uncertainty greater than 25 % have been eliminated instead. Resolution is 5 min × 37.5 m.

## 8 Sample atmospheric measurements

Using the best determination of the calibration constant, $k = 21.0 \pm 0.2$, (Sect. 6.3), we can determine $\delta$ and $d$ using Eq. (4) for a day's measurements to show the performance of CRL. Here we use measurements obtained on 12 March 2013, which we chose because two distinct cloud morphologies are present, as are a variety of signal levels in both depolarization channels, and because some particular places in the plot require special interpretation, which is discussed below. Figures 5 and 6 show

the depolarization ratio, depolarization parameter, and the uncertainties and relative errors for each. Many data points have uncertainties on the order of 10 % and smaller.

Just below 2 km altitude, a region of high depolarization is evident with low uncertainty. This implies that this region of the cloud is icy rather than made of liquid droplets. As altitude increases in the cloud, the depolarization drops. Is this because the cloud has suddenly turned into liquid droplets? Perhaps, but there are a few other factors to consider. First, the uncertainty

is higher in these regions, but this does not tell the whole story. This calculated uncertainty expresses *only* the uncertainty in the calculated result from Eq. (4) based on the number of photons returned in each measurement channel, and the statistical uncertainty in each of these measurements. As Eq. (4) is not applicable in circumstances of multiple scattering, any value calculated using this equation for time-altitude locations experiencing such scattering will not be valid as a proxy for particle phase – despite our (possibly precise) ability to calculate it. In the example presented here, the assumption of no multiple

scattering is decreasingly trustworthy high in the cloud, as photons have to pass through the thick cloud below twice. In future,





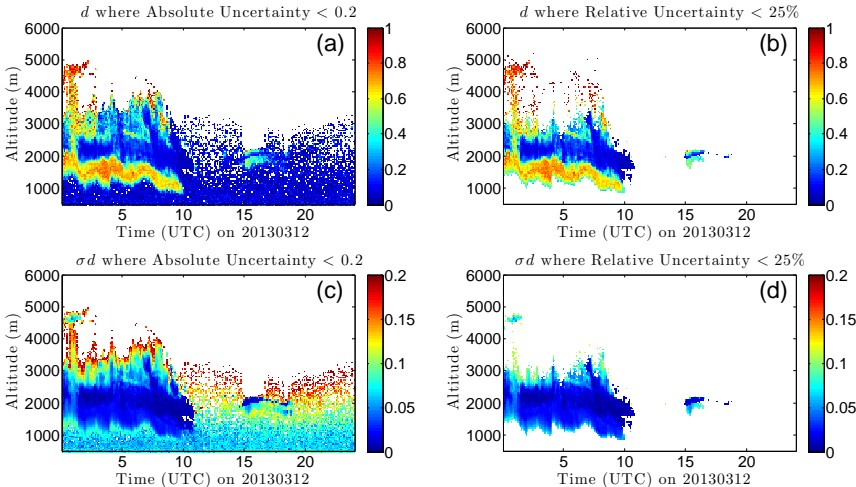

**Figure 6.** Atmospheric depolarization parameter measurements from 12 March 2013, with associated absolute uncertainties in units of depolarization parameter. In the left panels of $d$ (panel (a)) and $\sigma d$ (panel (c)), all data points with absolute uncertainty greater than 0.2 have been eliminated. In the right panels of $d$ (panel (b)) and $\sigma d$ (panel (d)) , points with relative uncertainty greater than 25 % have been eliminated instead. Resolution is $5\,\mathrm{min} \times 37.5\,\mathrm{m}$.

it would be well to quantify the added uncertainty due to the likelihood of multiple scatters. As this is beyond the scope of this paper, care must be taken when interpreting the depolarization values, even those with low measurement uncertainty. A further possible contributing factor is that the two channels may have differing amounts of extinction if the transmission function of the atmosphere is polarization-dependent. The most trustworthy depolarization values are those for which we have both low uncertainty *and* reasonable confidence that we have a low extinction single-scattering situation for every photon involved in

the data for that time-altitude bin. Examples of such locations in this example night are the lower portions of the large cloud before 10:00 UTC, and, to a smaller degree, the entire small cloud between 15:00 UTC and 20:00 UTC.

It may be most sensible to cut out regions which are not providing trustworthy depolarization values based on criteria discussed above. The values could be cut based either on relative error or on absolute error. The latter is chosen in this situation because a measurement of a depolarization parameter $d = 0.01 \pm 0.01$ is still meaningful, despite having 100 % relative error.

Interpretations deal with cutoffs between different values of $d$ and the measurements need to be sensitive enough to discern this cutoff, without eliminating an excessive number of measurements. Compare the data coverage of the upper left panel (panel (a)) in either Fig. 6 or Fig. 5 with the coverage of its corresponding upper right panel (panel (b)). Far more low-depolarization values are kept by cutting off using the absolute uncertainty rather than the relative uncertainty, without losing any interpretation confidence. Extra information is available at $3\,\mathrm{km}$ just after 05:00 UTC: There are distinct regions of increased depolarization

parameter which are retained when cutting depolarization parameter on absolute uncertainty rather than on relative uncertainty.





Although these are above thick cloud, and so multiple scattering may influence the interpretation of the specific values of $d$, the relative values can still be instructive, so it is useful to retain these. Features such as liquid layers within otherwise frozen clouds, or frozen parts within liquid clouds, would be detectable in similar situations, and regions of aerosols within clear air as well. Depending on the application, it may be of use to some lidar users to know simply that the cloud is inhomogeneous in cloud particle phase, or that an aerosol layer is present, without the specific microphyisical details which are usually available

with a well-constrained absolute depolarization parameter value.

Further coadding to lower resolution would help with the data coverage by increasing the number of perpendicular photons per bin. On some dates, measurements can be retained in this way up to 8 to 10 km altitude. However, this can only be carried out to a certain point, after which the low resolution depolarization measurements will be misleading, as any instances of thin liquid layers (low $d$ and $\delta$) residing within an ice cloud (high $d$ and $\delta$) would, at low resolution, show a smooth region with

intermediate values of $d$ and $\delta$ which are not actually present anywhere within the binned region. For this reason, measurements are kept at as high a resolution as possible, while still retaining as many measurement points as possible.

## 9   Future work

CRL can now make depolarization parameter measurements at a precision of $< 10\,\%$ uncertainty in clouds at a resolution of $5\,\mathrm{min} \times 37.5\,\mathrm{m}$ (Sect. 8), despite the less than optimum optical configuration of CRL. The major difficulty for CRL is in

receiving sufficient perpendicular-polarized photons at the depolarization PMT, as indicated by the very large calibration value of $k = 21$ found in the system. There are several possibilities for improvement of the depolarization measurements: Changes to the depolarization parameter calculation method, and changes to lidar hardware.

### 9.1   New analysis method

We have developed a new depolarization calculation method by extending the Mueller Matrix instrument characterization

method shown in this paper to a third CRL measurement channel: The unpolarized 532 nm Rayleigh Elastic channel. This method will be discussed in detail in a companion paper (McCullough et al., 2017), and is available in McCullough's PhD thesis (2015). In the new method, low resolution traditional $d$ values, as calculated in the present paper, are used to create a nightly calibration profile. This calibration profile feeds into an alternate expression for $d$ which depends only on the high-count-rate parallel and unpolarized channel measurements. The improved calculation method produces depolarization parameters with

similar uncertainties as the traditional method, but at much higher resolutions of $2\,\mathrm{min} \times 15\,\mathrm{m}$, and with far fewer data points lost to low signals, and therefore better measurement coverage of the atmosphere above the lidar. Alternately, the results can be expressed at similar resolution to the traditional depolarization products for CRL, but with much higher precision.

### 9.2   Hardware upgrades

Hardware upgrades are the other option for improving the CRL depolarization measurements. They are detailed here because

any or all of these would improve both calculation methods for $d$ by increasing signals in the perpendicular channel, and



because they are of use in any lidar which does not have access to an unpolarized channel at the same wavelength as its depolarization channels. In the context of the calibrations in this paper, hardware improvements seek to reduce the value of $k$ by increasing the number of perpendicular photons which reach the depolarization PMT.

Specifically, any of the following hardware changes would improve signals in the perpendicular channel. First, the laser's polarization could be rotated by 90°, allowing the collection optics in the polychromator to suppress the large parallel signal while enhancing the small perpendicular signal. Second, putting a quarter wave plate at the entrance to the polychromator would also help balance the signals for the same reason stated above. Third, replacing the VLWP filter with one which is less polarizing, or less polarizing in the perpendicular-suppressing direction would reject as few as possible the perpendicular photons which enter the telescope. Fourth, using two depolarization PMTs would allow for different gain settings individually optimized for the parallel and perpendicular channels. CRL's setup uses a single PMT to make both parallel and perpendicular measurements, and there is no switching of PMT gain settings between laser shots. Therefore, the gain setting must be a compromise of what is best for the high-signal parallel measurements and the low-signal perpendicular measurements. In practice means a lower gain setting is required to optimize the parallel channel to avoid PMT saturation. The high dynamic range of the combined analogue and photon counting of the Licel recorders helps somewhat, but having two signals of more comparable levels would be a better choice. Fifth, moving the depolarization PMT further upstream in the detector, perhaps with a pick-off beamsplitter at the beginning of the optical chain, would remove many of the partially-polarizing optics from the path to the depolarization PMT, thus rejecting fewer of the perpendicular photons than is currently the case. Any of the CRL hardware improvements here described would improve depolarization measurements made using the traditional equations for CRL or for any similar lidar, and would also improve measurements made using the newly-developed three-channel calculation technique.

Throughout this work, $k$ values far from unity have been presented as being undesirable. There can, however, be advantages to a setup with $k \neq 1$, albeit with $k < 1$ rather than $k > 1$ as CRL has. Consider that the scattering properties of the atmosphere lead to maximum $d$ and $\delta$ values of 1 for total depolarization, and 0 for no depolarization. For $d = 1$, half of the backscattered light reaching the roof window is parallel, and the other half perpendicular. For $d = 0$, all of the light is parallel, and none is perpendicular. Therefore, the maximum perpendicular signal ever possible to be backscattered to the lidar's roof window and through the receiver is only *half* the maximum parallel signal ever possible. With a whole-system calibration value $k = 1$, the maximum measured perpendicular signal will also be half of the maximum measured parallel signal. Values of $k > 1$ suppress the perpendicular signal even further. Therefore, $k > 1$ values for CRL are leading to the preferential attenuation of already-smaller signals. If CRL instead had $k$ values just as far from unity, but with $k < 1$, the larger parallel signal would instead be preferentially attenuated. This analysis leads to the interesting conclusion that for CRL, $k < 1$ optics would lead to better signal-to-noise ratios overall, and lower uncertainty in products derived from ratios of the two signals.

The new analysis technique makes it unlikely that the physical changes to CRL will be carried out in the near future, as they are less critical to the success of the CRL depolarization measurements than they would be if the traditional method was the only calculation possibility available. Therefore, all efforts have been directed toward the validation of the new analysis method.



## 10 Conclusions

Depolarization measurement capability has been added to the CRL lidar at Eureka, Nunavut in the Canadian High Arctic. Characterization tests have demonstrated that the traditional depolarization ratio and depolarization parameter equations and their calibration methods are appropriate to be used for CRL, despite the lidar not having been built specifically to make polarization measurements.

Mueller algebra was applied to determine suitable calibration constants to allow the calculation of the depolarization parameter. Using a similar Mueller algebra exercise, these calibration methods may be applied to other lidars whose optics may in fact contribute different unwanted optical effects. This method allows upgrades of depolarization capability to other lidars which were not originally designed specfically for polarization measurements.

The calibration values for CRL are $\frac{M_{10}}{M_{00}} = 0.910 \pm 0.002$ and $k = 21.0 \pm 0.2$. Using these values, an example 24-hour long atmospheric measurement demonstrates the depolarization ratio and depolarization parameter for the sky above Eureka on 12 March 2013, at a resolution of $5\,\mathrm{min} \times 37.5\,\mathrm{m}$. For the test night, a thick partially frozen cloud was present before $10{:}00\,\mathrm{UTC}$, and the reduced reliability of the depolarization measurements farther into the thick cloud are evident as multiple scattering becomes important. A non-frozen cloud with regions of higher depolarization material above it was present after $15{:}00\,\mathrm{UTC}$. CRL was also found to be sensitive to the low depolarization regions of clear air below the $15{:}00\,\mathrm{UTC}$ cloud.

This work has resulted in a functioning, well-characterized depolarization measurement system at CRL. Since this work, a new calculation technique based on similar calibration principles and Mueller Matrix algebra has been developed for CRL. This takes advantage of CRL's Visible Rayleigh Elastic unpolarized channel and produces depolarization parameter measurements to similar precision, but at 5 to 10 times higher resolution, and with better depolarization coverage of time and space. These improvements will be detailed in an upcoming publication.

*Data availability.* Data used in this paper available upon request from corresponding author (e.mccullough@dal.ca).

*Author contributions.* E. M. McCullough: Installation and calibration of depolarization hardware. Data analysis and development of method. Writing of analysis MATLAB code. Manuscript preparation. This work formed part of McCullough's doctoral thesis. R. J. Sica: Supervision of doctoral thesis. Contribution to manuscript preparation. J. R. Drummond: Principal Investigator of PEARL laboratory. Contribution to manuscript preparation. T. J. Duck: Principal Investigator of CRL lidar at the time of this work. Development of the CRL laboratory. G. J. Nott: Original design of depolarization channel for CRL. Instruction in installation and initial data processing in Python. Extensive discussions regarding calibration and analysis. Contribution to manuscript preparation. J. Hopper: Maintenance and testing of depolarization hardware. Lidar operations. C. P. Thackray: Lidar operations. Assistance in the laboratory. Contribution to manuscript preparation. C. Perro: Instruction on lidar operations. Lidar operations. J. Doyle: Instruction on lidar operations. Lidar operations. K. A. Walker: Principal Investigator of polar sunrise ACE/OSIRIS Arctic validation campaigns of which these CRL measurements are one component.





30    *Competing interests.* The authors declare that they have no conflict of interest.

*Acknowledgements.* PEARL has been supported by a large number of agencies whose support is gratefully acknowledged: The Canadian
Foundation for Innovation; the Ontario Innovation Trust; the (Ontario) Ministry of Research and Innovation; the Nova Scotia Research and
Innovation Trust; the Natural Sciences and Engineering Research Council; the Canadian Foundation for Climate and Atmospheric Science;
Environment and Climate Change Canada; Polar Continental Shelf Project; the Department of Indigenous and Northern Affairs Canada;
5   and the Canadian Space Agency. This work was carried out during the Canadian Arctic ACE/OSIRIS Validation Campaigns of 2010, 2011,
2012 and 2013, which are funded by: The Canadian Space Agency, Environment and Climate Change Canada, the Natural Sciences and
Engineering Research Council of Canada, and the Northern Scientific Training Program. This particular project has also been supported by
NSERC Discovery Grants and Northern Supplement Grants held by J. R. Drummond, R. J. Sica, and K. A. Walker, and the NSERC CREATE
Training Program in Arctic Atmospheric Science (PI: K. Strong). In addition, the authors thank the following groups and individuals for their
10  support during field campaigns at Eureka: PEARL site manager Pierre Fogal; Canadian Network for the Detection of Atmospheric Change
(CANDAC) operators: Mike Maurice, Peter McGovern, Alexei Khmel, Paul Leowen, Ashley Harret, Keith MacQuarrie, Oleg Mikhailov,
and Matt Okraszewski; and the Eureka Weather Station staff.





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




**Table 1.** Measuring calibration constants $\frac{M_{10}}{M_{00}}$ and $k$ using a depolarizer at various locations in the polychromator, illuminated with unpolarized lamp light. Test numbers correspond with locations marked in Fig. 1. The ratio of each $k$ value compared to the previous value is given by $\frac{k_i}{k_{i-1}}$. This quantity gives the contribution of each optic to the cumulative $\frac{M_{10}}{M_{00}}$ and $k$ values shown in that row, and is effectively the "$k$" value for that individual optic. A value of $\frac{k_i}{k_{i-1}} = 1$ indicates no partial polarization contribution (i.e. an ideal optic for depolarization measurement purposes). Test 9 is included for comparison purposes.

| Test | Light source location | $\frac{M_{10}}{M_{00}} \pm \sigma$ | $k \pm \sigma$ | $\frac{k_i}{k_{i-1}}$ | Contribution of additional optic included in each test |
|---|---|---|---|---|---|
| 1,4 | Directly upstream of polarotor | $0.001 \pm 0.004$ | $1.002 \pm 0.002$ | - | None. We expect $k = 1$ here, and it is. |
| 2 | Upstream of pellicle | $-0.250 \pm 0.002$ | $0.601 \pm 0.003$ | 0.600 | Moderate relative parallel attenuation by pellicle. |
| 3 | Upstream of collimating lens | $-0.211 \pm 0.003$ | $0.652 \pm 0.004$ | 1.085 | Minimal relative perpendicular attenuation. |
| 5 | Upstream of the Visible Long Wave Pass (VLWP) dichroic | $0.561 \pm 0.003$ | $3.56 \pm 0.03$ | 5.598 | Very large relative perpendicular attenuation by VLWP. |
| 6 | Upstream of the UV Long Wave Pass (UVLWP) dichroic | $0.472 \pm 0.003$ | $2.78 \pm 0.02$ | 0.78 | Small relative parallel attenuation by UVLWP. |
| 7 | At entrance of the polychromator | $0.719 \pm 0.001$ | $6.12 \pm 0.02$ | 2.20 | Large relative perpendicular attenuation by collimating optics at polychromator entrance. |
| 8 | Upstream of the focus stage | $0.741 \pm 0.001$ | $6.73 \pm 0.01$ | 1.10 | Minimal relative perpendicular attenuation by focus stage. |
| 9 | Upstream of roof window (from laser calibration in Sect. 6.3) | $0.910 \pm 0.002$ | $21.0 \pm 0.2$ | 3.12 | Large relative perpendicular attenuation by telescope and roof window. |

**Table 2.** Parallel and perpendicular signals for angles diagnostic of symmetry in the measurements. Each signal is effectively the same at $0.25\pi$ and $0.75\pi$ radians and integer multiples thereof.

| Angle $\theta$ (rad) | $S_\parallel \pm \sigma_\parallel$ | $S_\perp \pm \sigma_\perp$ |
|---|---|---|
| $-0.25\pi$ | $9.358 \pm 2.5$ | $0.666 \pm 1.3$ |
| $0.25\pi$ | $9.629 \pm 2.5$ | $0.693 \pm 1.3$ |
| $0.75\pi$ | $9.269 \pm 2.5$ | $0.701 \pm 1.3$ |
| $1.25\pi$ | $9.091 \pm 2.5$ | $0.834 \pm 1.3$ |
| $1.75\pi$ | $9.875 \pm 2.5$ | $0.789 \pm 1.3$ |
| $2.25\pi$ | $8.919 \pm 2.5$ | $0.830 \pm 1.3$ |