# Peer review of "Depolarization calibration and measurements using the CANDAC Rayleigh-Mie-Raman Lidar at Eureka, Canada"

_Atmospheric Measurement Techniques, 2017_

## Referee Comment (RC1) · Anonymous Referee #1 · 11 Apr 2017

Referee Comments

*Depolarization measurements using the CANDAC Rayleigh-Mie-Raman Lidar at Eureka, Canada*
Emily M. McCullough et al.

General Comments

This paper describes a very challenging retrofit of a depolarization channel to an existing lidar, in which optical elements upstream of the polarization analyzer had a major effect on the ratio of the parallel and perpendicular signals; there were a limited number of places in the optical path where test instruments could be inserted; and only a small fraction of the received light (<3%) was available for the measurements. In addition, some of the measurements required working above the roof hatch in an extremely cold and windy environment.

The history of the CANDAC lidar and the motivation for adding a depolarization channel are introduced well, and the paper is well organized and very well written. It includes very thorough and detailed explanations of experimental procedures and data analyses. It is fun for an experimentalist to read, because of the wealth of detail (including items such as the properties of two kinds of waxed paper and the use of foam core frames to hold sheets of them) and the fact that the Mueller matrix algebra is explicit and hence easy to follow.

A lengthy series of measurements showed that the perpendicular signal was suppressed by a factor of 21 relative to the parallel signal, and the optical components with the largest suppressing effects were identified. This part of the paper deserves a few more explanatory comments. There are many elements in the optical train with an incidence angle of 45 degrees. These are the classic cause of a polarization dependence, and a competent optical engineer will take each one into account to maximize the parallel signal, which necessarily minimizes the perpendicular signal. Was the CANDAC lidar designed this way, and is this the basic reason that the suppression factor is so high? In particular, the roof hatch/telescope system was a main contributor. By looking up the reference Nott, et al. (2012), one discovers that the telescope is Newtonian. Is not the secondary mirror the likely culprit? On the other hand, the focus stage has *four* 45-degree mirrors and yet it was found not to be a major contributor, surprisingly. What type of mirrors are used in it? Some comments on these issues would be most welcome. Also, when describing waxed paper as a depolarizer, it would be good to mention that it is also a highly scattering material, so that when the entire roof hatch window is covered with it, the received lidar signal is greatly reduced.

Unfortunately, the paper is an amalgam of obsolete and modern treatments in the lidar literature that perpetuates an old and misleading notation and terminology based on the notion that non-spherical particles in the air backscatter light that is polarized either parallel to the transmitted beam polarization or perpendicular to it. This idea is consistent with Eq. (1), in which parallel and perpendicular subscripts are attached to the backscatter coefficient. The depolarization ratio $\delta$ is defined as the ratio of "photons returned with polarization perpendicular to that of the transmitted laser beam, to those returned with polarization

parallel to that of the transmitted laser beam" (page 2 line 17). Do cloud particles rotate the polarization of part of the backscattered light by exactly 90 degrees? The authors say as much, for example in this sentence: "For d =1, half of the backscattered light reaching the roof window is parallel, and the other half perpendicular." (page 25 line 22). This notion is unphysical, of course, but the notation in Eq. (1) and the associated terminology were standard in the lidar literature until 2008, when C.J. Flynn et al. brought the appropriate Mueller matrix to the attention of the lidar community, enabling a reexamination by G.G. Gimmestad (as the authors correctly point out) and sparking a transition to analysis methods and terminology for lidar that is consistent with the rest of optical physics and scattering theory. There is now no reason to continue the obsolete and incorrect rubbish, and this problem in the paper is easily corrected with a few edits, as detailed in the next section of this review. Incidentally, the authors appear to lump the mutually exclusive pre- and post-2008 techniques together under the label "traditional", so this term should be re-visited everywhere it appears in the paper.

An experimental paper on a retrofit to a unique lidar is necessarily somewhat arcane, and so the authors include appropriate references to other lidar depolarization instrumentation papers that tie into a larger body of work and hence make the paper of wider interest. In this vein, this quite recent one could be added:

> Freudenthaler, V., About the effects of polarising optics on lidar signals and the Δ90-calibration, *Atmos. Meas. Tech.*, 2016, 9, 4181-4255, doi: 10.5194/amt-9-4181-2016

This reference was not available when the work described in the paper was done of course, but it is a quite general treatment of the effects of lidar system optics on depolarization signals, and it includes a calibration procedure for such lidars. The 74-page paper is very comprehensive, and the techniques described in it can be applied to a large variety of lidar systems. The Freudenthaler paper will likely be useful to anyone interested in the CANDAC paper. The authors might comment on how their work does or does not fit in with the analysis and calibration procedures in this reference.

Specific Comments

As background to these comments, here are the key facts for understanding lidar backscatter from randomly-oriented particles:

A. The two classes of "photons returned" are
   a. with polarization parallel to that of the transmitted laser beam, and
   b. unpolarized.
B. One-half of the unpolarized light goes through the perpendicular polarization analyzer and *one-half of it also goes through the parallel analyzer*.

These key facts are completely consistent with the Stokes and Mueller matrix formalism in the paper, as is easily verified by inspection and Eq. (5) can be derived from them with simple

algebra, without recourse to the formalism. The following lines need to be revised for consistency with the key facts:

Page 2 lines 17-21: for Schotland et al., it was really the ratio of the cross pol signal to the parallel pol signal, with proper calibration. Flynn did, in fact, have the definition of *d* wrong in words, but it is the fraction of the backscattered light that is unpolarized.

Page 3 lines 5-6: change it to …and that which is returned unpolarized.

Page 5 lines 16-17 & Eq. (1): change the words to be correct and change the equation to be

$$\delta = \frac{kS_{\perp}}{S_{\parallel}}.$$ That's all $\delta$ has ever really been. The signals are all we have to work with!

Page 24 line 15 – delete the words perpendicular-polarized.

Page 25 line 7 – replace perpendicular with unpolarized.

Page 25 lines 22-24 – this was cited under General Comments. Re-write in light of the key facts.

Technical Corrections

Page 7 line 8 – "in equal numbers" is redundant and should be deleted.

Page 8 line 1 – isn't "to the sky" redundant/ unnecessary?

Page 10 line 11 - delete second "this".

Page 11 line 3 – change "the all the" to all the.

Page 13 Fig. 2 – Photocounts were defined on the previous page as "photons per time bin per altitude bin", This should be mentioned in the caption, and the readers would like to know how many altitude bins were included in the data.

Page 16 line 18 should read …to go through …

Page 22 Fig. 5. The parameter $\sigma\delta$ has not been defined.

Page 24 line 6 – change photons to photocounts.

Page 25 line 12 "… in practice this means a lower gain …" (missing word).

---

## Referee Comment (RC2) · Anonymous Referee #2 · 10 Jun 2017

**Referee comments: amt-2017-76**

**Depolarization measurements using the CANDAC Rayleigh-Mie-Raman Lidar at Eureka, Canada**

This work presents in detail a particular methodology to add a depolarization channel to a lidar without change any of the previous optic configuration and apply the Müller matrix theory to track the polarizing effect of the lidar optics. Despite there are many studies which shows the 'best practices' to measure lidar depolarization — in fact, Section 9.2 shows how the hardware could be improved—, I understand that it is not always possible to make the best setup but the optimal one according to the circumstances. Therefore, I consider interesting how the authors face the complexity of the setup and the methodology can help to scientific community to face similar experimental setups. Considering the scopes of the journal, I recommend this paper for publication but the authors should strongly consider the following comments.

I would like to suggest a change of the title of the manuscript. By now, the title highlights the 'depolarization measurements' as a key goal of the manuscript but at the end, the depolarization measurements are only present in Section 8 and not discussed from a scientific point of view. Since the main result of the paper is the depolarization calibration, I would recommend a title fitting the content of the paper and an improved introduction fitting the 'real scope'.

The lidar depolarization technique, and specially the depolarization calibration, is an important research field. However, reading the introduction, I have the feeling that this is the first time that the depolarization is calibrated. I strongly recommend a brief state of the art about the calibration procedures and studies (Alvarez et al., 1999; Snels et al. 2009; Freudenthaler et al., 2010; Bravo-Aranda et al., 2013; and references there in), highlighting why these methods are not applicable to this lidar or how an existing calibration method was adapted. Additionally, authors should include other references of interest such as the general theory based on Stokes-Müller formalism recently presented by Freudenthaler, 2016. In other words, the introduction is too straightforward to me since it is not consider the state of the art of the lidar depolarization technique up to date.

Regarding the structure of the manuscript, I suggest to move the paragraph P2L6-P2L16 to a section called 'Site and lidar description'. Since this work is part of the thesis of the first author, I'll take the opportunity to 'remind' that the introduction should gather the state of the art and the explanations about why the work is necessary and useful.

Bravo-Aranda et al, 2016 quantifies the systematic errors on the depolarization measurements from the non-calibrated parts of the lidar but the polarization effect of the Newtonian telescopes with 90° fold mirrors is not evaluated. The calibration values presented in this study demonstrates that the combination 'window roof + Newtonian telescopes' strongly affect the depolarization measurements (Table 1, ki/(ki-1) = 3.12) and thus, it should be highlighted as an interesting result of this paper.

Minor comments:

**P2L12: 1064 nm is not available?**

P3L11: The use of a single PMT for the parallel and perpendicular measurement is presented as an advantage. However, in Section 9.2, *'using two depolarization PMTs would allow for different gain settings individually optimized for the parallel and perpendicular channels'* is presented as a hardware change that would improve the perpendicular signal. May the authors clarify which is the best configuration?

P3L23: Does licel report the temporal stability of this device?

**P5L1: 'atop' -> typo?**

P5L15: I suggest to include a comment about the different definitions (Cairo, 1999). The concept of 'photons polarized perpendicular' is 'old-fashion' and has already demonstrated wrong. Please, revise the paper considering the explanation of Gimmestad 2008.

P17L21: I understand that authors use the depolarization sheet to isolate the polarizing effect since *there is no alternative* but, in any case, I would appreciate the technical specifications of the depolarization sheet (glassine). From the phrase 'To keep the photon count rates as high as possible during the test, only a single layer of glassine was used, although using two sheets in series ensures more complete depolarization' (P18L26), I would say that depolarizer is not perfect ( $d \neq 1$ ). Was the depolarization degree of the sheet measured? Any information in this regard would be great for the scientific community (accurate measurements of the polarizing characteristics of the *depolarization sheet* are not so common). Did authors consider use the equations to find the effect of an almost perfect *depolarization sheet*?

P22L15-P23L0: I agree with the authors that the effect of the multiple scattering on depolarization measurements has to be evaluated but is out of the scope of this work. Nevertheless, some references on this regard will be appreciated by the readers.

P25L23: 'half of the backscattered light reaching the roof window is parallel...' Parallel to what? This way to understand the depolarization is dangerous. The polarization state of the photons is not binary (parallel/perpendicular). The parallel and perpendicular signal *with respect to the polarizing component of the emitted laser beam* is the way we measured the received light. Please, revise the whole manuscript.

---

## Referee Comment (RC3) · Anonymous Referee #3 · 12 Jun 2017

General Comments:

The paper presents a method to calibrate the newly installed depolarization channels of the CRL lidar instrument at Eureka, Nunavut. The depolarization channels make use of only a small fraction of the available light collected by the lidar instrument since initially the lidar was not designed to retrieve depolarization products. The study is designed to overcome this drawback by calibrating these channels.

The calibration method for this study implies the use of additional optics to test the receiving unit. As presented in the manuscript, the optical components situated "up-stream" the depolarization analyzer have significant influence on the lidar products.

[Figure]

When performing this calibration, the effects on the depolarization channels are removed with some degree of accuracy. The paper starts with a detailed description of the instrument together with the motivation for adding depolarization capabilities to the instrument. The paper gives descriptions of experimental setups and optical constrains of the instrument, but also an introduction on the Mueller matrix formalism. Still some algebra in the theoretical description is not fully explained and some comments are insufficient. Explanations on why some simplifications are used are missing and part of the optical chain is ignored without any additional details. This part should be improved if the paper is published. Also the state of the art on depolarization lidars and their calibration can be improved.

The results show a calibration value over 20, meaning that the perpendicular signal was drastically reduced within the lidar optics – this result should be further discussed. Comments and conclusions on the calibration technique were also expected in the concluding part of the paper.

Specific comments:

Abstract:

P01 L05: "well-characterized lidar channels" - how are the channels well characterized and how does a new depolarization channel influence this characterization? Suggest replacing with "To reduce its impact on the existing lidar channels,...."

P01 L09: "within ±20% ..." - is this value sufficient to express the results of the calibration? Is this value similar both for low and high depolarization layers? If this is the case than this should be stated (perhaps not in the abstract but in the paper itself)

Introduction:

P01 L21: "the phase of cloud particles" – needs a reference

Installation of depolarization hardware:

[Figure]

P03 L09-11: "This design reduces the number of differences between the hardware of both depolarization channels because the backscattered light traverses identical optics and uses the same photomultiplier tube. Given" - yes this design reduces the number of differences due to identical optics but can include additional errors from the rotation of the polarizer. What is the rotation accuracy of this module and the stability? A short comment on this issue would be welcomed.

Polarotor:

P03 L21: "90 degree" – a comment on the rotation accuracy of the polarotor should be included? How does this accuracy influence the results?

P03 L22: "Two recording buffers are used in the Licel Transient Recorder, one for parallel and one for perpendicular photocount profiles." - is this comment necessary?

P03 L23-24: "The extinction ratio of the polarizer was characterized by the manufacturer to be $5 \times 10-5$ or smaller " - it is also given by the accuracy of the rotation angle. This section should better describe the polarotor.

Figure 1 shows a collimating lens between points 2 and 3. Could the authors describe the purpose of this lens.

P04 L08: could you provide more detail on how was this channel characterized

P04 L08-11: "installation. Therefore, a regular plate beamsplitter or dichroic mirror could not be used to pick off the light for the depolarization channel; this would have translated the transmitted 607nm light too much, and the downstream channel would have had to be realigned." - But still a collimating lens was introduced before the 607/532p and c channels. How does this lens affect the alignment? Why is the collimating lens placed in front of the pellicle beamsplitter and not after the splitter? This way the lens would only affect the 532.1nm channel.

P05 L05: "angle of 15degree" – please mention "full angle"

P05 L06-07: "It is also convenient to have the cables from the polarotor to the electronics rack be accessible without the need to open the polychromator." – is this comment really necessary?

P07 L30: Ilaser purity - was this verified by any experimental measurement. According to Freudenthaller 2016, laser emission is not 100% polarised.

P07 L04: - rotation of the plane of polarization of the laser with respect to a reference (usually the polarizer separator) must also be accounted for. For a rotation of Mreceiver and Mtransmitter with respect to the laser polarization, the collected polarized light could be also altered. Some comments must be included. The study should also take into account these effects (Bravo-Aranda, J. A. 2016)

P07 L05: "The optical backscattering effects of the atmosphere can be described as Matm." - Mtransmitter is initially mentioned but left out further in the study. According to Nott et al 2012 - the number of emission optics is significant and could influence the polarization purity of the emitted light (this added to the assumption that the laser unit is emitting 100% polarized light). Comments should be included to explain these assumptions.

P07 L12: "The gain factor is not stable long term, but for any given minute of data it will be constant for both channels" - please give more details

P08 L01-02: "The overlap varies with changes to the lidar's alignment to the sky" – consider reformulation

P08 L07-09: "During setup, the "parallel" analyzer position was also oriented such that it can be represented as a horizontal polarizer (by aligning the parallel direction with the direction of maximum signal in a low depolarization sky)." - As stated before, also the rotation of the optics (Mtransmitter and Mreceiver) must also be accounted for. For Mtransmitter and Mreceiver rotation with respect to the laser polarization, collected polarized light could be altered.

P08 L19-21: "These Mueller matrices combine to make an overall equation for each channel which describes the action of all optical components on the light, and results in Stokes vectors (shown in full in Eq. (9)) and I (which differs from only by two minus signs in the polarizer matrix):"

The author must include a comment on the effects of the polarotator rotation uncertainties. An extra comment is also required for the rotation of the laser polarization purity.

The author provided information on how the rotation of the laser plane of polarization is corrected but no information on possible rotation of the receiving optics is provided.

Again more detail should be given to the "... action of all optical elements ...". Many optical components are excluded (transmission and part of the receiving optics) - this should be clearly mentioned. Suggest changing to: "A simplified version of the overall equations combine to make an ....... " - ...... including ideal laser polarization purity, no emission optics, no laser rotation, no optics rotation, no retardation effects, ideal polarizer and so on. How will these simplifications affect the final results?

P09 L02-03: "lidar, we solve for the depolarization parameter d to learn about the atmosphere" - consider reformulating

P09 L04: "to solve for the" - consider reformulating

P09 L10: Another option would be to use the three signal calibration since the lidar instrument is able to measure the total, parallel and cross 532.1nm components. This will be a nice add-on to this study: a comparison between the current calibration and the suggested calibration.

Polarization and Depolarization generating calibration optics

Title - consider reformulating

P09 L25: "It is placed immediately downstream of the focus stage and" - please indicate

on the picture

P10 L08-09: "This product was mounted in such a way as to be held relatively taut in a frame, or held gently in place by other mechanical means." – Too vague, consider reformulating

P10 L24: Fig 2 - why do we have such different photoncounts for the two channels? Why do you have a setup counting so few photons per time-bin? Could this be improved by expanding the time bin?

P11 L05: "properties: a 1m diameter circle" - Consider reformulating

P11 L10: "held between the telescope's tertiary mirror and the focus stage worked better" - please indicate on Fig 2 (label 7?)

P11 L12: "By sacrificing the inclusion of both" - what do you mean by sacrificing the inclusion?

P11 L15: "It" should be replaced by -> The cube beamsplitter

P11 L17: "test, there is no advantage to using lidar returns as the light source"- usually when using the lidar return as the light source we take into account the height dependency of the optics (Freudenthaler 2016): Light collected from different heights have different lightpaths and different incidence angles in the receiving modules. this must be taken into account since the lidar optics are not polarization independent - see fig 2 - count number is strongly different for the same initial light source. This issue should be reconsidered.

P11 L25: "optics are contributing any non-simple-gain effects to the signals" - please reformulate. It is not clear.

P11 L26-27: ""If we consider the optics and detector starting after the focus stage, can we use the simplified Eq. (4) and Eq. (5) to find the calibration constant, and then to determine depolarization ratio and depolarization parameter" - the three signals

calibration would include all optics in the receiving unit of the lidar instrument. This should be a viable option for this study.

P12 L07: "There are very similar equations for the case in which we use backscattered laser light rather than lamp light." - This is not fully correct if we consider the laser polarization purity, laser rotation, emission optics

P12 L11-12: "The absolute angles were determined in post-processing, such that the maximum in the parallel channel is 0." - This data combined with information on the actual position of the cube could be used to extract the laser rotation relative to the parallel detection. Was this study performed?

P12 L14: 7.5m - if the light source is a lamp placed on the telescope frame, the range dependence should be inexistent. Under these assumptions, the altitude bin could be extended to several hundred meters or even km to improve the photocounts number. By using this method Fig 2b would have a much better fit in the n*pi regions (n=0,1,2).

P12 L14: "There is approximately a 2 degree or 0.035 radian uncertainty" – Is this uncertainty taken into account in the study?

P12 L14: "angles" - what angles? The calibrator or polarotor - it should be the calibrator, right?

P12 L15: "The overall signals in S parallel far exceed the overall signals S cross." - Why? What does this indicate?

P12 L16-17: "by allowing them as free parameters in a fit to these signals," - Does this return a unique solution?

P12 L20-22: please give more details on why is this

P13 L01: "If our measurements are symmetric, with ..." - A detailed explanation must be included. What are the considerations that are the base of this result? This explanation must be detailed in the manuscript.

[Figure]

P14 L04-05: please reformulate also including detailed explanations

P14 L10-11: "The mean signal values are ... perpendicular" - is this relevant?

P14 L20: "the parallel and perpendicular channel signals each go to zero" - Fig 2 b shows the number of photon counts as a function of incident light polarization angle. The low amount of photons used for each point makes it difficult to perform a fit on the data. The mean values around n*pi (n=0,1,2) are clearly forced to be zero. We can see six values around pi that are zero. To clearly say that "........perpendicular channel signal each go to zero" it is mandatory to have a higher amount of photons. This may also apply to Fig 2 a. This could be accomplished if the author increases the time and height bins

P15 L14-15: "For situations in which the true signal is zero, a mean of the measured signal will be reported as a larger value, thus not being indicative of the most probable photon counting result." - This is one reason why the number of photoncounts must be increased either by increasing the time interval or by increasing the altitude window to hundred of meters or even to km

P15 L24-25: "This assumes that the telescope does not contribute to these quantities in a significant way." - is this assumption based on any measurement? We see that the telescope includes many 45 degree mirrors. This should have a significant influence.

P15 L25-26: "This result is reasonable, as the reflectivity of all telescope mirrors are high." - How does this statement exclude any depolarization effects that the optics may have on the collected light?

P16 L08-09: "It includes no effects of the telescope or focus stage." - In the upper paragraph the author stated that: ".... This assumes that the telescope does not contribute to these quantities in a significant way. This result is reasonable, as the reflectivity of all telescope mirrors are high." This paragraph states that the results are not representative since the telescope is not included. These two statements contradict each other.

Please reconsider the statements.

P16 L10: "Because of these limitations ..... k, using an unpolarized light test" - unpolarized light was also used in the upper section. Please reformulate so that the reader clearly understand what the author mean by this statement.

P16 L12-13: "optics. Having measured the partial-polarizer-like form of the upstream optics Mueller Matrix using the polarized calibration test, we can proceed with confidence in the tests in the following section." – consider reformulating

P16 L20: "it enters the lab" – could be changed to: is collected by the instrument

P16 L22: "complete depolarization is" - what is complete depolarization? is this complete for cases when the depolarization ratio is 1? Is this condition satisfied in ice clouds?

P17 L23: "Using a flexible material like glassine was important in the Arctic winter." – is this sentence important?

P18 L03-09: is this explanation really necessary? Could this be replaced with a comment on what were the requirements of the setup?

P18 L14: "and has been indicated here in white." – Please remove

P20 L01-07: this section could be reduced.

Suggestion: "Different approaches showed that the best retrieval method for the assessment of k was to .....(2.)......

P20 L17-18: "directly before the polarotor. It was moved sequentially upstream, placed between any two optics where there was room to safely insert it, up to and including right in front of the lamp, upstream of the focus stage" – suggest changing to -> "before the polarotor and then moved sequentially upstream ."

P20 L19: "Industrial kitchen grade waxed paper was used for this test" - since the
author presented a better solution for this material - the test should at least include a comparison between the two and a explanation on why this material is still presented in the manuscript.

P21 L15: "Many lidar groups" - please provide examples

P21 L28: "the more convenient calibration is insufficient." - What does this mean?

P21 L30: "remove or upgrade the" – suggest changing to -> "change the"

P21 L30: "we change optics in" – suggest changing to -> We upgrade

Figure 5: "25%" – how did you estimate this value?

P22 L02: "on 12 March 2013," - this date is prior to the calibration date. It must be stated that the calibration factors are constant and can be used for measurements collected before the actual calibration date.

P22 L03: "because two distinct cloud morphologies are present" - A simpler example should be used at the beginning since the aim is to demonstrate the performance of the instrument.

P22 L10: "regions, but this does not tell the whole story" - is this part necessary?

P22 L10: "This calculated uncertainty expresses only the uncertainty in the calculated result from Eq. (4)" – consider reformulating

P22 L14: "valid as a proxy for particle phase – despite our (possibly precise) ability to calculate it." - consider excluding this section

P22 L15: "is decreasingly trustworthy high in the cloud" – consider reformulating

Figure 6: could this figure be included in fig 5 with two color scales for the two parameters?

P23 L01-03: "A further possible contributing factor is that the two channels may have differing amounts of extinction if the transmission function of the atmosphere is

polarization-dependent." – It is not clear what the author meant by this statement. More explanations are required.

P23 L11 – P24 L03: "Compare the ....  as well" - please rephrase.  This section is meaningful to the study but care must be given in phrasing the message.

P24 L07-10: "However, this can only be carried out to a certain point, after which the low resolution depolarization measurements will be misleading, as any instances of thin liquid layers (low d and delta) residing within an ice cloud (high d and delta) would, at low resolution, show a smooth region with intermediate values of d and delta which are not actually present anywhere within the binned region.  " - Please divide this section into several sentences. It is relatively hard to follow the message.

P24 L16-17: "There are several possibilities for improvement of the depolarization measurements: Changes to the depolarization parameter calculation method, and changes to lidar hardware." – Please rephrase

P24 L22-27: Please rephrase. Too vague

P24 L29 – P25 L02: "Hardware....channels." – Please rephrase

P25 L03-19: is there really a necessity for this paragraph?

P25 L20: "Throughout this work, k values far from unity have been presented as being undesirable.  There" – Is this the transmitted message?  As a reader, I do not understand this message from the manuscript.

P25 L31-34: is this section necessary for the manuscript?

Conclusions - detailed information on the calibration method and its particularities must be presented. Conclusions on what is the most polarization sensitive optics in the lidar receiving unit must also be included.

P26 L11: "night" to "period"

P26 L12-13: "and the reduced reliability of the depolarization measurements farther into the thick cloud are evident as multiple scattering becomes important." – Please rephrase

P26 L13: "above it was present" – please rephrase

P26 L14: "sensitive to the low" – suggest changing to -> "sensitive also in low"

P26 L15: "functioning, well-characterized depolarization" – ambiguous

P26 L15-19: consider excluding from the manuscript

Technical corrections:

P05 L19: beam,

P06 L12: "and solving for k." – consider reformulating

P06 L21: "allowing for optical effects in the upstream optics" – consider reformulating

P09 L14: "would" to would be

P10 L05: Product to product

P11 L03: "A 1m depolarizing optic to initially depolarize the all the backscattered" to A 1m depolarizing optic to initially depolarize all the backscattered...

P11 L05-07: "which could be held completely flat, which could survive the harsh outdoor conditions of Arctic winter, which could be easily and repeatably rotated to the appropriate orientation, and which had sufficient optical polarization quality." – to -> which could be held completely flat and could maintain within required characteristics even in harsh outdoor conditions of Arctic winter. The depolarizing optic must be easily and repeatable rotated to the appropriate orientation and should have sufficient optical polarization quality.

P11 L27 parameter?".

P12 L15: "each" –> each of the two channels

P15 L23: "by others in the community" to -> in the remote sensing community

P16 L08: "rather the of the polychromator" -> one extra "the"

P17 L23: When the roof hatch open is open – please review

———————————————

---

## Author Response (AR1)

**Author response to reviewer comments**

**Atmos. Meas. Tech. Discuss., doi:10.5194/amt-2017-76-RC3, 2017**

We would like to thank the three reviewers of this paper for their detailed comments, corrections and recommendations.  We believe that the incorporation of their reviews has improved the paper.

**Original title:**
Depolarization measurements using the CANDAC Rayleigh-Mie-Raman Lidar at Eureka, Canada
Emily M. McCullough1,2a,*, Robert J. Sica1, James R. Drummond2, Graeme Nott2,4a, Christopher Perro2,
Colin P. Thackray2, Jason Hopper2, Jonathan Doyle2, Thomas J. Duck2, and Kaley A. Walker3

**New title:**
Depolarization calibration and measurements using the CANDAC Rayleigh-Mie-Raman Lidar at Eureka, Canada
Emily M. McCullough1,2a,*, Robert J. Sica1, James R. Drummond2, Graeme Nott2,4a, Christopher Perro2,
Colin P. Thackray2, Jason Hopper2, Jonathan Doyle2, Thomas J. Duck2, and Kaley A. Walker3

Reason: Title change requested by Reviewer 2 (item R2a, in general comments) because the paper focuses largely on calibration rather than atmospheric measurements.

**Notes about the organization of this document:**

The comments of all three reviewers have been combined to keep this document organized by page and line number of the discussion paper, so far as possible.

Reviewers identified as:      R1 = Reviewer 1
        R2 = Reviewer 2
        R3 = Reviewer 3

In the case of similar or overlapping content in the comments (e.g. a general comment is satisfied by the resolution of a specific comment, or if one response/action addresses multiple comments), we give the full details of the change *only* at the location of the comment which which required the largest changes. All other related comments make reference to the item with the full response, identified by comment number. The full response may be earlier or later in the document than the comment in question.

**General comments** are those for which there is no particular page or line number, and/or which affect many parts of the paper. These are addressed first.

General comments are identified as:      R1a,b,c = Reviewer 1 first comment, 2nd comment, 3rd comment, ...

**Specific comments** refer to specific page and line numbers of the discussion paper, and are addressed next. Comments from all reviewers are given together in order of page and line number.  Author comments are clearly denoted amongst these specific reviewer comments.

To retain consistency with each reviewer's individual page and line number system, specific comments are identified as:      R1: Page 2 lines 17-21: = Reviewer 1, Page 2, Lines 17-21.
        R2:  P2L12: = Reviewer 2, Page 2, Line 12.
        R3: P01 L05: = Reviewer 3, Page 1, Line 5.
        Author Comment: P01 L05 = An additional correction made by the authors.

Black text (hereafter):
Reviewer comment [with additional surrounding text for necessary context given in square brackets]

Blue text:
Response = Our comments to the reviewers and the public
Action = What we have done to change this in the text of the article

**General comments:**

R1a:  A lengthy series of measurements showed that the perpendicular signal was suppressed by a factor of 21 relative to the parallel signal, and the optical components with the largest suppressing effects were identified. This part of the paper deserves a few more explanatory comments. There are many elements in the optical train with an incidence angle of 45 degrees. These are the classic cause of a polarization dependence, and a competent optical engineer will take each one into account to maximize the parallel signal, which necessarily minimizes the perpendicular signal. Was the CANDAC lidar designed this way, and is this the basic reason that the suppression factor is so high? In particular, the roof hatch/telescope system was a main contributor. By looking up the reference Nott, et al. (2012), one discovers that the telescope is Newtonian. Is not the secondary mirror the likely culprit? On the other hand, the focus stage has *four* 45-degree mirrors and yet it was found not to be a major contributor, surprisingly. What type of mirrors are used in it? Some comments on these issues would be most welcome.

Response: CRL was not designed to have depolarization measurements taken into account, so yes - it was designed to optimize reflection for the parallel signals, as usual in optical design. The depolarization channels were not originally intended to be part of the main lidar receiver. If we were designing a depolarization lidar from scratch, this is of course not the design we would choose! The lidar telescope is a Dall-Kirkham design. It is a modified Cassegrain, and contains a concave elliptical primary mirror, a convex spherical secondary, and its tertiary mirror is flat. The focus stage does have four 45 degree mirrors, but these are arranged in two planes such that the polarization effects from two mirrors are undone by the others. See item R3: P15 L24-25 for further response on this issue. Since 2013 we have made calibration measurements to isolate effects of the telescope from those of the roof window, but these have not been fully analyzed yet, and are thus not included in this paper.

Action: See item R3: P15 L24-25 for changes to text.

R1b:  Also, when describing waxed paper as a depolarizer, it would be good to mention that it is also a highly scattering material, so that when the entire roof hatch window is covered with it, the received lidar signal is greatly reduced.

Response: Agreed.

Action: Will add a sentence to Page 10 line 9. Refer to specific comments R1 P10 L3 for response.

R1c:  Unfortunately, the paper is an amalgam of obsolete and modern treatments in the lidar literature that perpetuates an old and misleading notation and terminology based on the notion that non-spherical particles in the air backscatter light that is polarized either parallel to the transmitted beam polarization or perpendicular to it. This idea is consistent with Eq. (1), in which parallel and perpendicular subscripts are attached to the backscatter coefficient. The depolarization ratio delta is defined as the ratio of "photons returned with polarization perpendicular to that of the transmitted laser beam, to those returned with polarization parallel to that of the transmitted laser beam" (page 2 line 17). Do cloud particles rotate the polarization of part of the backscattered light by exactly 90 degrees? The authors say as much, for example in this sentence: "For d =1, half of the backscattered light reaching the roof window is parallel, and the other half perpendicular." (page 25 line 22). This notion is unphysical, of course, but the notation in Eq. (1) and the associated terminology were standard in the lidar literature until 2008, when C.J. Flynn et al. brought the appropriate Mueller matrix to the attention of the lidar community, enabling a reexamination by G.G. Gimmestad (as the authors correctly point out) and sparking a transition to analysis methods and terminology for lidar that is consistent with the rest of optical physics and scattering theory. There is now no reason to continue the obsolete and incorrect rubbish, and this problem in the paper is easily corrected with a few edits, as detailed in the next section of this review.

Response: We have made major modifications, particularly to subsection 3.1, which has been reworked into the introduction.  Further, see the responses to specific comment items  R1: Page 2 lines 17-21, R1: Page 3 lines 5-6, R2: P5L15, R1 Page 25 lines 22-24, and general comment R1d. Items R1e, R2c and R3b1 also request an expanded literature review, so this item will cover those changes as well.

Action: See responses to specific line items listed above, which have specific details. In addition to those, here are the modifications to Section 1, Introduction:

At P02 L16, delete all text after the sentence "Adding 532nm linear depolarization capabilities to this instrument is an economical way to add additional capacity to study Arctic clouds, in concert with other instruments at PEARL such as the Millimetre Cloud Radar (Moran et al., 1998), the E-AERI interferometer (Mariani et al., 2012), and the Starphotometer (Baibakov et al., 2015)." until the next section break.

Beginning in the same paragraph, directly after that sentence, insert the following new text (which now references Figure 1 (the diagram of the receiver) much earlier in the paper:

"To preserve continuity in the long-term data sets from other CRL channels, no existing optics were altered or removed during the installation of the depolarization channels. Figure 1 is a diagram of the CRL's receiver, showing the seven original measurement channels, and indicating the locations of the new pellicle beamsplitter, Polarotor rotating Glan-Thomson prism, interference filter, focusing lens, and photomultiplier tube of the 532.1 nm depolarization channel. CRL uses a single PMT to measure light of two polarization planes on alternate laser shots, with a laser repetition rate of 10 Hz. This is similar to Platt (1977), which operated with a laser repetition rate of 1 to 5 Hz. CRL's higher repetition rate means that the assumption of simultaneous measurements in both polarization planes is reasonable. As the original lidar optics were not chosen for their polarization properties, the optical design of the CRL has made the calibration of the depolarization measurements challenging."

A new section "1.1 Depolarization lidar theory" now contains the previous Equations 2, 3, 4, 5. These are now numbered Equations 1, 2, 3, 4. The previous Equation 1 has been eliminated from the document. The equations for d and delta are better introduced and described, according to requests from Reviewer 1. All descriptions of polarization, including nomenclature and concepts, have been re-worked to match the language used in Gimmestad 2008. These changes are reflected in the remainder of the paper as well.

A new section "1.2 Literature review of depolarization calibrations" contains entirely new text, with the exeption of the paragraph about the early development of Mueller Matrix Algebra by Mueller and Parke III, which was retained from the discussion paper. The expanded literature review addresses suggestions from all three reviewers. It now provides an overview of existing calibration approaches, and specifics are used to show how each of these applies, or does not apply, to the CRL lidar and our calibrations.

A new section "1.3 Mueller Matrix calibration goals for CRL" contains new text. This shows the new calibration methods which are used for CRL in the present paper, and gives an overview of the rest of the paper.

The full new text for sections 1.1, 1.2, and 1.3 are as follows:

**1.1 Depolarization lidar theory**

With the new depolarization capabilities, we aim specifically to investigate "the atmospheric phenomena which change the polarization state of the light received by a lidar relative to the state of the transmitted light" (Gimmestad, 2008). Depending on the optical qualities of the particles, a population of randomly oriented identical particles in the atmosphere should either a) not change the polarization state of the light (i.e. all light from that population will be returned polarized parallel with respect to the state of the transmitted light), or b) should cause the light to become completely unpolarized on its return. There may be more than one population of particles present in any given scattering volume. The calculation to determine the change in polarization requires a ratio of the intensity of light which is returned unpolarized to the total intensity of light which is returned in any and all polarization states (Flynn et al., 2008; Gimmestad, 2008). Expressed in this manner, the quantity of interest is d, the depolarization parameter: the portion of the total light intensity I which has become depolarized through scattering. Similar descriptions, called depolarization factor, are given as early as van de Hulst (1957). The depolarization parameter is defined as: $d = I_{unpol.}/(I_{pol.} + I_{unpol.})$. (1)
In the event that the atmosphere does not depolarize the beam, there will be no intensity returned with polarization different than the transmitted light, and therefore $d = 0$. In the case of complete depolarization, $d = 1$.

Because lidars measure signals from photomultiplier tubes (PMTs), and not the backscattered light intensity

directly, the equation for d and must be reformulated in terms of lidar observables. Gimmestad (2008) demonstrates this development using Mueller Matrix algebra, with normalized matrices. Two quantities are measured. The first is the signal in a channel which uses a polarization analyzer to admit light polarized parallel to the polarization plane of the transmitted laser beam (the "parallel" channel), and signal in a channel which uses a polarization analyzer to admit light polarized perpendicularly to this plane (the "perpendicular" channel). In this way, in the absence of any complicating factors, for linearly polarized transmitted light, the parallel channel will be sensitive to half of the backscattered light which has been unpolarized during scattering ( 1/2 Iunpol. ) and all light which remains polarized during scattering (Ipol.). The second signal is that in the perpendicular channel, which will be sensitive only to half of the unpolarized light ( 1/2 Iunpol. ), and none of the light which remains polarized when backscattered. In Gimmestad's paper, the signals in the receivers, S, are individually "assumed to be calibrated", but no further details about these calibrations are provided. Presumably, this assumption considers the combined effects of all optics upstream of the PMT and the gain of the PMT, acting together as a constant attenuation factor for each individual channel. If the factors differ between channels, the overall effect in the system as a whole is that of partial polarizer.

Under these conditions, the equation for depolarization parameter is given as:

$$d = \frac{2kS_\perp}{S_\parallel + kS_\perp} = \frac{2k\frac{S_\perp}{S_\parallel}}{1 + k\frac{S_\perp}{S_\parallel}} = \frac{2}{\frac{1}{k}\frac{S_\parallel}{S_\perp} + 1} \quad (2)$$

in which: $S_\perp$ is the signal measured by the perpendicular channel, $S_\parallel$ is the signal measured by the parallel channel, and $k = G_\parallel/G_\perp$ is the depolarization calibration constant, in which $G_\parallel$ is the gain (or attenuation) of the parallel channel, and $G_\perp$ is the gain (or attenuation) of the perpendicular channel. The third form for d in Eq. (2) is easier to handle experimentally as each measurement appears only once and thus uncertainties may be considered uncorrelated.

Historically, "depolarization" has also referred to δ, the depolarization ratio. This quantity is proportional to the ratio of the perpendicular signal $S_\perp$ to the parallel channel $S_\parallel$ (e.g. Hohn, 1969; Schotland et al., 1971; Liou and Schotland, 1971; Freudenthaler et al., 2009). The depolarization parameter d is directly relatable to the expression for depolarization ratio, δ, through the same signal measurements and the same calibration constant:
$\delta = kS_\perp/S_\parallel$. (3)
The conversion between the quantities d and δ is: $d = 2\delta/(1 + \delta)$. (4)

A variety of expressions for "depolarization" are described in Cairo et al. (1999). The δ described in the current paper is most closely related in meaning to the Pal and Carswell (1973) "volume linear depolarization ratio" cited therein, but it is not strictly equivalent because no claims are made here about the connection between δ and backscatter coefficient. Rather, δ is defined here only as a function of measured quantities. Gimmestad (2008) provides motivation for moving away from all δ descriptions, pointing out that d is consistent with the rest of optical physics and scattering theory. Expressing depolarization as d has since been adopted in such publications as Hayman and Thayer (2009, 2012) and Neely III et al. (2013). Results in the present paper will be provided in terms of both d and δ so that readers working under either paradigm can readily make use of the figures and calculations.

The expressions for d and δ of Eq. (2) and Eq. (3) are all referred to in this paper as "traditional" in the sense that in each equation, a single k value determines the calibration.

**1.2 Literature review of depolarization calibrations**

The calibration constant k can be determined by introducing unpolarized light into the detector (i.e. setting d = δ = 1) or, equivalently, light polarized at ±45◦ with respect to each of the planes of polarization for the detectors (also sets d = δ = 1) and measuring the signals in each channel. k is then the ratio of the two signals. The location of calibration optics or lamps within the lidar determines how much of the system will be characterized through the calibration. The most strict meaning of k is the ratio of gains of the detectors, if the polarization state of the calibration light is defined directly before the polarizing beamsplitter. A wider interpretation for the meaning of k can include relative gains resulting from other receiver optics if the polarization state is defined earlier in the receiver, and can include relative gains of transmitter optics if the polarization state for the test is set within the

Author response to reviewer comments: Atmos. Meas. Tech. Discuss., doi:10.5194/amt-2017-76-RC3, 2017

transmitter. The laser is assumed to be completely linearly polarized. The orientation of the parallel and perpendicular polarization analyzers must be correctly set at 90◦ to each other, and oriented correctly with respect to the usual polarized returns from the transmitted laser beam, before the calibrations for k commence. Using k as the only calibration factor ignores the possibility of any retarding and rotating effects which may exist in the transmitter and receiver, in all of the optical components. This is more likely to be appropriate for lidars which have few receiver optics before the polarization analyzer (e.g. Wang et al., 2009), and is less likely to be appropriate for lidars which have many receiver optics which are not optimized for polarization measurements, such as the CRL.

Some groups begin the calibration for k with lidar returns from an atmospheric region which is assumed not to depolarize the light (or to depolarize only a known minimal amount as a result of molecular scattering) for the duration of the calibration. Then a half waveplate is included in the receiver to control the orientation of the polarized backscattered light as it enters the detectors, aligning it at ±45◦ with respect to both polarization analyzers (Spinhirne et al., 1982; McGill et al., 2002; Alvarez et al., 2006; David et al., 2012; Wang et al., 2009; Freudenthaler et al., 2009; Neely III et al., 2013; Bravo-Aranda et al., 2013; Freudenthaler, 2016). This type of calibration is not sensitive to polarization effects in the transmitter optics, nor to any optics upstream of the half wave plate. This typically means omitting at least the telescope, and sometimes more optics. A notable exception is Neely III et al. (2013), which has a waveplate optic in the roof window.

In an alternate version of the half waveplate calibration, this waveplate may be placed in the transmitter to control the orientation of the plane of polarization of the laser light transmitted to the sky (Liu and Wang, 2013; Neely III et al., 2013; Freudenthaler, 2016; Bu et al., 2017). Eloranta and Piironen (1994) uses a pockels cell in their laser to the same effect. Locating the calibration optic in the transmitter is a method which includes as many optics as possible in the calibration.

All calibrations using polarized light must be concerned with obtaining the correct orientation of the waveplate (or other relevant optic). Spinhirne et al. (1982) and some lidar examples in Freudenthaler et al. (2009) arrange the waveplate as well as possible such that the output is at +45◦ and/or −45◦, and report the calibration measurements only from those specific angles. Other groups show improved results for k by calculating its value at a variety of waveplate rotation angles, then using a fit to determine the optimal rotation angle from which to use the calibration values (Alvarez et al., 2006; Hayman and Thayer, 2009; Snels et al., 2009; Liu and Wang, 2013; Bu et al., 2017). Of those in the latter case, methods using both +45◦ and −45◦ measurements together (calibrations 90◦ apart from one another), can have error terms which compensate well for one another in the event that the waveplate is misaligned by the same amount in each case (Freudenthaler et al., 2009, some systems in Freudenthaler, 2016).

Sassen and Benson (2001) use a different method for simulating a $d = \delta = 1$ situation. They introduce unpolarized lamp light to their detector from the point of focus of the telescope. This calibration method is not sensitive to any polarization effects in the transmitter optics or the telescope, but the polarization state of the calibration light is well-known, and there are no calibration optic rotation angles to introduce errors.

CRL calibrations for k use a sheet of depolarizing material in the receiver. This can be placed at a variety of locations within the receiver. The results of these tests provide the motivation for this paper. When using a depolarizing sheet directly upstream of the polarotor (see Fig. 1, location 1 and 4; same location as half wave plate used in e.g. Alvarez et al. (2006); Wang et al. (2009); Freudenthaler et al. (2009)), we find that k = 1. This is exactly as expected. Effectively, this is an estimate for the strict version of k: the ratio of PMT gains – and CRL uses the same PMT for each depolarization channel. Repeating the calibration measurements with a depolarizing sheet at the entrance to the receiver roof window suggest a value closer to k = 21.0 ± 0.2 for the whole CRL receiver (Sect. 6.3), indicating that optics upstream of the polarotor are significantly polarizing. Clearly a more thorough instrument depolarization characterization is required for the CRL. If our optics are so highly polarizing, they may carry other optical consequences as well, which Eq. (2) and Eq. (3) are insufficient to describe.

Various approaches are available in the literature to account for non-ideal depolarization lidar hardware, each with their own assumptions. Some calibrations are tests with a temporarily-installed optic. These allow for calibration factors to be determined, which will then be applied to regular measurements made without the

temporary calibration optic in place. Other calibrations consist of adjusting compensation optics (typically by adjusting their rotation angle) which will remain in the lidar during regular measurements. Some of these are the same optics used for the calculations of k. At present, the lidar of Neely III et al. (2013) seems most capable of a whole-system calibration. Their calibration optics exist in multiple places within the transmitter, and multiple places within the receiver. The laser light can be rotated directly as it exits the laser, again as it exits the laboratory, again as it enters the telescope, and altered yet again as the light enters the PMTs. Further, the lidar was designed to make depolarization measurements, and optics could be selected and oriented with this in mind, as indicated in Hayman and Thayer (2012). Their liquid crystal variable retarder has some effects related to laboratory temperature which must be considered, but the authors have accounted for these. No other lidar of which we are aware has all of these capabilities. The more common calibrations each assess only some of the possible complications. Two are discussed briefly here.

First, Sassen and Benson (2001) allow calibration for the effect of angular misalignment between the transmitter and receiver planes of polarization in their measurements and calibrations. Other groups have introduced methods of optical compensation for such an angular misalignment: The half wave plates used in the transmitters of Spinhirne et al. (1982); Liu and Wang (2013); Bu et al. (2017), in the receivers of McGill et al. (2002); Wang et al. (2009); Freudenthaler et al. (2009), and in both transmitter and receiver of Neely III et al. (2013) for the k calibration remain in the lidars. During regular measurements, the optics are aligned such that a maximum of non-depolarized backscattered light is directed to the appropriate channel. These calibrations to account for angular mismatch between transmitted and received planes of polarization generally depend on a "known" sky depolarization of aerosol-free molecular-only scattering (e.g. Platt (1977)) or liquid-droplet-only stratospheric clouds (e.g. Adachi et al. (2001), requiring additionally a total backscattering ratio measurement). Bravo-Aranda et al. (2016) and Freudenthaler (2016) analyze the effects of transmitter-receiver angular misalignment on uncertainties in the retrieved atmospheric depolarization values. The calculations therein are less relevant for CRL because both studies include the error that is induced by leaving an extra compensation optic in the lidar (the half-wave plate). The CRL (similarly to e.g. Alvarez et al. (2006)), does not leave any calibration optics in the optical path during routine measurements. Thus, the uncertainties due to angular mismatch must be dealt with a different way. CRL carries out a polarotor start delay test in clear sky (see Sect. 2.3) to ensure angular alignment between transmitted and received planes, and an assessment of uncertainty is carried out using a simple model.

Second, lidars using a polarizing beamsplitter to separate received light to two separate PMTs have to account for different efficiencies for each plane of polarization in their parallel and perpendicular channels, one being reflected and the other trans- mitted through the beamsplitter. These calculations are integral to some recent works (Liu and Wang, 2013; Bravo-Aranda et al., 2016; Freudenthaler et al., 2009), but are not relevant for CRL, in which both parallel and perpendicular measurements are made using the transmitted beam of light through the polarotor.

In contrast to the methods discussed to address these two complications, which tend to be dealt with individually, the CRL's approach in this paper is to determine the optical effects of the receiver as a whole. The recent papers on depolarization calibration are moving away from a scalar description, and are moving toward a vector description of light, with matrix algebra describing the optical effects of the sky (Kaul et al., 2004; Hayman and Thayer, 2009, using Mueller Matrix algebra) and of the lidar itself (Biele et al., 2000; Hayman and Thayer, 2009, 2012; Neely III et al., 2013; Freudenthaler, 2016, using Mueller Matrix algebra and Bu et al., 2017, using Jones matrices).

The Mueller Matrix algebra upon which this technique relies was introduced as lectures and conference proceedings by Hans Mueller in the early 1940s (e.g. Mueller (1946a, b, 1948)). These and his previous works (Mueller, 1943a, b) remain difficult to obtain, and those available (e.g. in summary report Bush (1946), which describes the design and use of the shutter described in Mueller (1943a)) do not explicitly demonstrate the matrix algebra. A better and more available source describing all of the Mueller Matrix algebra in considerable detail is the thesis of Mueller's PhD student, Nathan Grier Parke III (Parke III, 1948).

In Hayman and Thayer (2012), there is a rigorous mathematical development of the Mueller matrices for lidar instrument optical contributions of various sorts. This is followed in a similar way by Freudenthaler (2016) and Bravo-Aranda et al. (2016), which use Mueller Matrix algebra to work out the expected signals for a sample of

calibration and measurement situations, including errors. In those papers, and in Bu et al. (2017), many of the contributing matrices and vectors are considered to be standard forms for well-understood optics. In that sense, these works are a detailed forward-looking development intended to account for possible errors in known parameters (e.g. introducing a term for error in the rotation of transmitter with respect to receiver, and then determining it for their lidar). For CRL, we also take the Mueller Matrix approach. We particularly follow the lead of Hayman and Thayer (2009, 2012) with regards to the mathematics, but with the opposite perspective: We initially presume to know nothing about the elements of the receiver optics Mueller Matrix, and then we measure them.

**1.3 Mueller Matrix calibration goals for CRL**

For the CRL, our approach is to use Mueller Matrix mathematics to more fully diagnose the optical properties of CRL's receiver as a whole, similar to the approach taken by Di et al. (2016) and Liu and Wang (2013). We do not require the specific contributions of each receiver optic in order to understand our measurements for d. Nor do we need to split the matrices into equivalent standard optics (e.g. Hayman and Thayer (2012), which describes optics as combinations of retarders, polarizers, etc.). Neither do any of the specific examples given in Freudenthaler (2016) adequately describe the CRL.

The first goal for this paper is to use Mueller Matrix algebra to re-derive the equation for d, including calibration terms which describe the arbitrary optical effects of the upstream optics. These terms allow the collection of upstream optics to be represented using the most general single Mueller Matrix possible (See Sect. 3.1, Eq. (12)). We make no prior assumptions regarding rotation, retardation, polarization properties of the optics. The beamsplitter and laser polarizations are assumed to be ideal in our expressions.

The second goal is to use calibrations to measure the relevant matrix elements for the upstream optics which will indicate whether or not the overall impact of CRL's optics is that of a partial polarizer. If not, and it shows behaviour similar to that of a wave plate or a polarization rotator, then Eq. (2) and Eq. (3) are insufficient to describe the depolarization parameter and depolarization ratio for CRL, and the full equations for d from Sect. 3.1 will be required for routine measurements. The main tests introduce light of known polarization to the detector at a variety of rotation angles. This is generated by putting unpolarized light through a polarizing cube beamsplitter. Some compromises must be made, as we must at times exclude the telescope and focus stage from our calibrations, similar to calibrations made by Platt (1977); Spinhirne et al. (1982); Sassen and Benson (2001); Wang et al. (2009); Bravo-Aranda et al. (2013, 2016) and others. With careful characterisation of the remainder of the lidar receiver, we show that the overall contribution of the optics is indeed found to conform to that of a partial polarizer, rendering Eq. (2) and Eq. (3) appropriate for CRL.

Third, we find the appropriate calibration constant k for the whole receiver, including the telescope and focus stage, and using the laser as a light source, and using a sheet depolarizer to force all light entering the receiver to be unpolarized.

Fourth, we carry out additional unpolarized-light tests to determine the contributions that individual optics make to the overall large k value for CRL. The largest contributor to k for CRL was found to be the Visible Long Wave Pass filter (Sect. 7.2, and Table 1).

Finally, we demonstrate the use of the CRL's newly calibrated depolarization capability by showing some example measurements of ice clouds from 12 March 2013 in Sect. 8. The result for the CRL is a new depolarization data product tied into a scientifically significant long-term measurement record, all without compromise to the continued acquisition of the original types of data. To date, linear depolarization measurements have been made for four polar sunrise seasonal campaigns at Eureka: 2013, 2014, 2016, and 2017 (no measurements were obtained during 2015 because no operator was available).

R1d: Incidentally, the authors appear to lump the mutually exclusive pre- and post-2008 techniques together under the label "traditional", so this term should be re-visited everywhere it appears in the paper.

Response: In the context of this paper, *both* Gimmestad 2008 and the depolarization ratio methods are traditional. We recognize that the development in Gimmestad 2008 is a major step forward in describing

depolarization lidar in terms of scattering physics. Nevertheless, they retain something in common which is relevant for CRL: Both cases assume that a constant gain factor multiplied onto each signal suffices for calibration (k in depolarization ratio, and not specified by a variable in d, other than saying the signals are proportional to intensity. And k according to some sources does not include anything other than a ratio of PMT gains specifically - which in our case is 1). We were not sure that would be sufficient for CRL. What about rotation? Retardation? We definitely need at the very least to expand k to allow it to include effects of other optical elements. Why should we assume that we have an instrument matrix in which all on-diagonal elements are equal, and all off-diagonal elements are zero? The "traditional" therefore is the assumption that we don't have anything non-constant-gain, and nothing other than an overall "partial polarizer effect" happening in the instrument matrix. In contrast, the "non-traditional" material in this paper is the part which allows the instrument matrix to have elements of any value, which we will discover. Of course, we end up finding out that the simpler "traditional" equations are applicable for CRL. But the development of the more general situation stands.

Action: We modified section 3.1, in which we added the sentence "These expressions for $d$ and $\delta$ are both referred to here as "traditional" in the sense that in each equation, a single $k$ value determines the calibration." Then Section 3.1 has been reworked into the introduction. See item R1: Page 2 lines 17-21 for the full new text.

R1e: Add reference. An experimental paper on a retrofit to a unique lidar is necessarily somewhat arcane, and so the authors include appropriate references to other lidar depolarization instrumentation papers that tie into a larger body of work and hence make the paper of wider interest. In this vein, this quite recent one could be added: Freudenthaler, V., About the effects of polarising optics on lidar signals and the Δ90-calibration, *Atmos. Meas. Tech*., 2016, 9, 4181-4255, doi: 10.5194/amt-9-4181-2016 This reference was not available when the work described in the paper was done of course, but it is a quite general treatment of the effects of lidar system optics on depolarization signals, and it includes a calibration procedure for such lidars. The 74-page paper is very comprehensive, and the techniques described in it can be applied to a large variety of lidar systems. The Freudenthaler paper will likely be useful to anyone interested in the CANDAC paper. The authors might comment on how their work does or does not fit in with the analysis and calibration procedures in this reference.

Response: Thank you for bringing this paper to our attention. We will add it into the revised literature review (New Section 1.2), and point out specifically how the CRL requires a slightly different approach to that given in Freudenthaler 2016. We comment also in the new Section 2.3 (Polarotor Start Delay calibration), about the machined timing holes for the polarotor disk, "One consequence of this arrangement is that although there may be error in the absolute angle during each measurement, the parallel and perpendicular channels are exactly 90◦ apart", which is a helpful comment for readers who are also familiar with the Freudenthaler Δ90-calibration.

Action: See the response to item R1c for the full new literature review. This forms the new Section 1.2 of the paper. See also response to item R2c. The response to specific item R3: P03 L09-11 gives the full new text of Section 2.3.

R1f Background Info comment: As background to these comments, here are the key facts for understanding lidar backscatter from randomly-oriented particles:
A. The two classes of "photons returned" are
a. with polarization parallel to that of the transmitted laser beam, and
b. unpolarized.
B. One-half of the unpolarized light goes through the perpendicular polarization analyzer and *one-half of it also goes through the parallel analyzer*.
These key facts are completely consistent with the Stokes and Mueller matrix formalism in the paper, as is easily verified by inspection and Eq. (5) can be derived from them with simple algebra, without recourse to the formalism.

Response: Thank you for the clear explanation of the specific nomenclature you wish us to include in the paper. All polarization nomenclature has been reworked in the paper.

Action: See responses to individual items, e.g. R1c, R1d, R1: Page 2 lines 17-21, R1: Page 3 lines 5-6, R2: P5L15, R1: Page 25 lines 22-24, and similar.

R2a: I would like to suggest a change of the title of the manuscript. By now, the title highlights the 'depolarization measurements' as a key goal of the manuscript but at the end, the depolarization measurements are only present in Section 8 and not discussed from a scientific point of view. Since the main result of the paper is the depolarization calibration, I would recommend a title fitting the content of the paper and an improved introduction fitting the 'real scope'.

Response: The current title is "Depolarization measurements using the CANDAC Rayleigh-Mie-Raman Lidar at Eureka, Canada". We will change the title.

Action: Suggested new title: Depolarization calibration and measurements using the CANDAC Rayleigh-Mie-Raman Lidar at Eureka, Canada

R2b: Bravo-Aranda et al, 2016 quantifies the systematic errors on the depolarization measurements from the non-calibrated parts of the lidar but the polarization effect of the Newtonian telescopes with 90¡ fold mirrors is not evaluated. The calibration values presented in this study demonstrates that the combination 'window roof + Newtonian telescopes' strongly affect the depolarization measurements (Table 1, ki/(ki-1) = 3.12) and thus, it should be highlighted as an interesting result of this paper.

Response: The other reviewers were likewise interested in this topic. Please see the more detailed responses in items R1a and the new text in item R3: P15 L24-25.

Action: Please see new text in item R3: P15 L24-25.

R2c: I strongly recommend a brief state of the art about the calibration procedures and studies (Alvarez et al., 1999; Snels et al. 2009; Freudenthaler et al., 2010; Bravo-Aranda et al., 2013; and references there in), highlighting why these methods are not applicable to this lidar or how an existing calibration method was adapted. Additionally, authors should include other references of interest such as the general theory based on Stokes-Mller formalism recently presented by Freudenthaler, 2016. In other words, the introduction is too straightforward to me since it is not consider the state of the art of the lidar depolarization technique up to date. Since this work is part of the thesis of the first author, I'll take the opportunity to 'remind' that the introduction should gather the state of the art and the explanations about why the work is necessary and useful.

Response: We agree. Thank you for the suggestion to include these references. When the focus of the paper changed from the atmospheric measurements to having more to do with calibration, we neglected to provide sufficient review of the techniques to date. We will remedy this. The literature review has now been expanded in the introduction, given in full in the response to item R1c. Item R3: P07 L04 also suggests a Bravo-Aranda et al. reference (2016). Section 3.1 has been moved and modified into the introduction as well, so that the literature review can make use of the definitions of d and \delta when describing the various calibrations and formalism.

Action: Please see item R1c for the completely reworked introduction, including a literature review in the new Section 1.2 which includes the references suggested here by Reviewer 2.

R3a: Still some algebra in the theoretical description is not fully explained and some comments are insufficient. Explanations on why some simplifications are used are missing and part of the optical chain is ignored without any additional details. This part should be improved if the paper is published.

Response: Please see the responses to Reviewer 3's specific comments, particularly items R3 P08 L19-21 part d and location P11 L24 - 29 in the text. We have also modified the introduction considerably, which covers the theoretical description of depolarization (the new Section 1; see item R1c).

Action: Please see the responses to Reviewer 3's specific comments.

R3b1: Also the state of the art on depolarization lidars and their calibration can be improved.

Response: We will improve the literature review section of the manuscript. Please see the response to item R1c

for the full new introductory section including literature review (new Section 1.2 in particular).

Action: See response to item R1c.

R3b2: The results show a calibration value over 20, meaning that the perpendicular signal was drastically reduced within the lidar optics – this result should be further discussed.

Response: We feel that Section 7 addresses specifically the latter point as thoroughly as possible with the tests we were able to do in 2013. We should add some sentences near the start of the manuscript to further make this point. The high value of k was in fact the motivation for the more detailed calibrations which we carried out.

Action: None.

R3c: Comments and conclusions on the calibration technique were also expected in the concluding part of the paper.

Response: The conclusion will be changed to address this. See response to item R3: P25 L31-34b for specific changes.

Action: See response to item R3: P25 L31-34b.

**Specific comments:**

R3: P01 L05: "well-characterized lidar channels" - how are the channels well characterized and how does a new depolarization channel influence this characterization? Suggest replacing with "To reduce its impact on the existing lidar channels,...."

Response: Reference to (Doyle et al., 2011; Nott et al., 2012). is given on page 2, line 13. The reference to (Nott et al., 2012) is repeated on page 3, line 14. These references describe the characterization of the other lidar channels in detail. We will replace the sentence with different wording.

Action: Replace the sentence "To reduce its impact on the existing, well-characterized lidar channels, the depolarization hardware was placed near the end of the receiver cascade." with the sentence "To minimize disruption in the existing lidar channels and to preserve their existing characterization so far as is possible, the depolarization hardware was placed near the end of the receiver cascade.".

Author Comment: P01 L07: Changed verb tense in abstract such that the work done in this paper is in present tense.

Action: Change "were used" to "are used in the sentence "Calibrations and Mueller matrix calculations were used to determine and mitigate the contribution of these upstream optics on the depolarization measurements." such that it now reads "Calibrations and Mueller Matrix calculations are used to determine and mitigate the contribution of these upstream optics on the depolarization measurements."

R3: P01 L09: "within +/-20% ..." - is this value sufficient to express the results of the calibration? Is this value similar both for low and high depolarization layers? If this is the case than this should be stated (perhaps not in the abstract but in the paper itself)

Response: Thank you for bringing up this point. The 20% value is an oversight from a previous version of the paper in it was in direct reference to a plot of relative uncertainty in d, which has since been removed. We prefer now to write in terms of absolute uncertainty, but neglected to write the percent as a decimal value. A value of \sigma = 0.2 is appropriate, as per the cutoff value shown in Fig. 5 and Fig. 6. The plots in Figure 6 show the absolute uncertainty associated with each value of d. The dominant source of uncertainty in d is the statistical uncertainty from photon counting. Values of d for which the count rates are low will have an associated uncertainty which is high. (The form of uncertainty as a function of photocounts is visually similar to $y = 1/(x)$, but closer to the origin of the plot, so could likely be modeled as $y = 1/(x-a)$ -b or similar.) In the context of the plots here, the low count rate areas are those without clouds and regions at high altitude. In Figures 5 and 6, points with higher depolarization points for any given value of total backscatter (backscatter plots not shown) have lower uncertainty, simply because these result from measurements with higher photon count rates in the perpendicular channel. We will modify this sentence in the abstract. To address the portion of the comment "Is this value similar both for low and high depolarization layers?", we feel that this is addressed initially by the plots in Fig. (5) and Fig. (6), from which the uncertainties may be directly read for each atmospheric and depolarization value situation. To clarify, we will specify a typical uncertainty within the clouds shown (+/-0.05, which is about 10% relative uncertainty at the depolarization values present for those clouds), and will point out that the uncertainty differs based on whether or not the measurement is within or without a cloud.

Action: a) Replace the sentence "The results show that with appropriate calibration, indications of cloud particle phase (ice vs. water) are now possible to precision within 20 % uncertainty at time and altitude resolutions of 5 min 37.5 m, with higher precision and higher resolution possible in select cases" with the sentence "The results show that with appropriate calibration, indications of cloud particle phase (ice vs. water) through the use of the depolarization parameter are now possible to a precision of +/- 0.05 absolute uncertainty (+/- 10% relative uncertainty) within clouds at time and altitude resolutions of 5 min and 37.5 m respectively, with higher precision and higher resolution possible in select cases. The uncertainty is somewhat larger outside of clouds at the same altitude, typically with absolute uncertainty +/- 0.1.". This has been updated in the conclusion section, also.

Author Comment: P01 L18: Wording improvement

Action: Modify the sentences "The evaluation of cloud phase in models requires more observational datasets in order to improve (Shupe, 2011), with phase transitions being of particular interest (Kalesse et al., 2016). Therefore, measurements of cloud particle phase (ice vs. water) are necessary in order to more fully understand the radiation balance of the Arctic atmosphere." such that they now read "In order to develop models with improved fidelity of the cloud phase, more observational measurement datasets are required (Shupe, 2011), with phase transitions being of particular interest (Kalesse et al., 2016). Measurements of cloud particle phase (ice vs. water) are therefore necessary in order to more fully understand the radiation balance of the Arctic atmosphere."

R3: P01 L21: "the phase of cloud particles" – needs a reference

Response: Fair comment. The full sentence is "Liquid droplets can exist well below 0 degrees C, so cloud temperature is not sufficient to determine the phase of cloud particles."

Action: We will add a selection of example references which show this to be the case. Add these to the sentence, which will then read: "Liquid droplets can exist well below 0 degrees C, so cloud temperature is not sufficient to determine the phase of cloud particles (e.g. \cite{ShupeBAMS2008}, \cite{DeBoerJAS2009}, \cite{CurryJC1996}, \cite{SassenBAMS1991},\cite{WestbrookRMS2013})."

The new references to be included are:
M. Shupe, J. Daniel, G. de Boer, E. Eloranta, P. Kollias, C. Long, E. Luke, D. Turner, and J. Verlinde, "A focus on mixed-phase clouds: The status of ground-based observational methods," Bulletin of the American Meteorological Society, 2008
G. D. Boer, E. Eloranta, and M. Shupe, "Arctic mixed-phase stratiform cloud properties from multiple years of surface-based measurements at two high-latitude locations," Journal of the Atmospheric Sciences, vol. 66, pp. 2874–2887, 2009.
J. Curry, W. Rossow, D. Randall, and J. Schramm, "Overview of Arctic cloud and radiation characteristics.," Journal of Climate, vol. 9, pp. 1731–1764, 1996.
K. Sassen, "The polarization lidar technique for cloud research: a review and current assessment," Bulletin of the American Meteorological Society, vol. 72, pp. 1848– 1866, December 1991.
C. D. Westbrook and A. J. Illingworth, "The formation of ice in a long-lived supercooled layer cloud," Quarterly Journal of the Royal Meteorological Society, vol. 139, pp. 2209–2221, October 2013.

R2: P2L6-P2L16: Regarding the structure of the manuscript, I suggest to move the paragraph P2L6-P2L16 to a section called 'Site and lidar description'.

Response: We feel that both options (leaving section as-is, or having a dedicated section) are reasonable. It was difficult to have sufficient motivation for the rest of page 2 following line 16 without keeping this section early in the paper. It is precisely the implementation of CRL's depolarization capability as an "afterthought" which makes this paper novel, rather than a simple repeat of work done by others. Thus, this bit of detail about the lab and the instrument should remain as early as is practical in the manuscript. Further, the laboratory is in a remote location. This has bearing on every other item in the paper, and the reader must be aware of this large constraint when reading follow-on sections. We have rearranged Section 1 (Introduction) considerably to accommodate some other requests by reviewers, and have added to Section 2. These changes may improve the organization from the perspective of this request as well.

Action: None specifically to address this request. We have introduced Figure 1 (receiver diagram) earlier, so that both the site and instrument are able to be referenced when giving the literature review. Equations 1, 2, 3, and 4 (previously numbered 2, 3, 4, and 5) have been moved into the introduction as well. This is particularly important for showing how calibrations from the literature are not appropriate to apply "as-is" for CRL. We have collected more information into Section 2 regarding the installation of the Polarotor. Therefore, more of the introductory theory is collected into Section 1, while more of the specifics of the depolarization installation hardware is collected into Section 2.

R2: P2L12: 1064 nm is not available?

Response: The 1064 nm laser light is separated out of the frequency-doubled and frequency-tripled laser beams, and is directed into a beam dump. No 1064 nm light is transmitted to the sky, and none is measured by the lidar. (Incidentally, no 532 nm light is transmitted to the sky from the 355 nm laser, either).

Action: None.

R1: Page 2 lines 17-21: for Schotland et al., it was really the ratio of the cross pol signal to the parallel pol signal, with proper calibration. Flynn did, in fact, have the definition of *d* wrong in words, but it is the fraction of the backscattered light that is unpolarized.

Response: We will make major changes to the paper to emphasize the modern terminology for lidar depolarization, following primarily the example of Gimmestad 2008. To this end, the entire end half of the introduction has changed (P02 L16 - P03 L2). The literature review has been expanded, and now also includes the text from Section 3.1.

Action: Section 1 now has an ameliorated literature review, with material from section 3.1 now included so as not to repeat the same information twice. The full changes to the introductory Section 1 are extensive. The specific changes pertaining to this reviewer comment are given here.

Particular to the present reviewer response, replace the two sentences "The basic quantity upon which lidar depolarization calculations are based is the ratio of photons returned with polarization perpendicular to that of the transmitted laser beam, to those returned with polarization parallel to that of the transmitted laser beam (e.g. Hohn (1969); Schotland et al. (1971); Liou and Schotland (1971)). This quantity is known as the depolarization ratio, \delta ." with the sentences "Historically, "depolarization" has also referred to δ, the depolarization ratio. This quantity is proportional to the ratio of the perpendicular signal $S_\perp$ to the parallel channel $S_\parallel$ (e.g. Hohn, 1969; Schotland et al., 1971; Liou and Schotland, 1971; Freudenthaler et al., 2009).".

Also pertaining to this response, replace the two sentences "An alternate expression for depolarization is the ratio of photons returned with polarization perpendicular to that of the transmitted laser beam, to the total number of returned photons of any polarization (Flynn et al., 2008; Gimmestad, 2008). This alternative to the depolarization ratio is called the depolarization parameter, d." with the sentences "The calculation to determine the change in polarization requires a ratio of the intensity of light which is returned unpolarized to the total intensity of light which is returned in any and all polarization states (Flynn et al., 2008; Gimmestad, 2008).

R1: Page 3 lines 5-6: change it to …and that which is returned unpolarized.

Response: At the time this work began, the main references were to traditional d = k Sperp/Spara nomenclature. We felt that the sentence in question was still valid for this section, in light of Gimmestad 2008 nomenclature for d, because the unpolarized backscattered contribution to the total lidar signal may be decomposed into parallel and perpendicular components.The CRL lidar does measure signals of light which is polarized parallel and perpendicular (after the polarization analyzers, of course), regardless of the fact some of the parallel signal, and all of the perpendicular signal, come from light which is unpolarized when it hits the roof window. We will clarify by modifying the present sentence, and adding another.

Action:  Change the sentence "To make depolarization measurements, the lidar must be able to distinguish between backscattered light which is polarized parallel to the outgoing laser light, and that which is returned polarized perpendicularly." to read "To make depolarization measurements, the lidar must be able to distinguish between backscattered light which is polarized parallel to the outgoing laser light, and that which is returned unpolarized.". Include a new sentence directly afterward which reads "To this end, we make measurements in two orthogonal polarization planes: One parallel to the polarization plane of the outgoing laser light, and the other perpendicular to this plane.".

R3: P03 L09-11: "This design reduces the number of differences between the hardware of both depolarization channels because the backscattered light traverses identical optics and uses the same photomultiplier tube. Given" - yes this design reduces the number of differences due to identical optics but can include additional errors from the rotation of the polarizer. What is the rotation accuracy of this module and the stability? A short

comment on this issue would be welcomed.

Response: These effects were considered when carrying out the calculations, and they do not have a large effect on the results. These were not initially shown in the manuscript in order to make the manuscript more concise. As there is interest from the reviewers in this topic, we will add these details back in. Licel does not provide information about the rotation accuracy nor the stability of this module. We trigger the whole lidar from the polarotor. This means that the laser pulses will remain synced with the rotation of the polarotor, even if the speed of the polarotor changes. More polarotor details will be added to Section 2, as a new subsections 2.3 and 2.4. Items R3: P07 L04 and R3: P03 L23-24, and others also affect the content of these subsections.

Action: More polarotor details will be added to Section 2, as a new subsection after page 5 line 13 in the discussion paper: "2.3 Start Delay calibration to define the "parallel" polarotor rotation angle", and as a new subsection "2.4 Effects of polarotor rotation angle errors on depolarization measurements".  The full added text is:

**2.3  Start Delay calibration to define the "parallel" polarotor rotation angle**

The polarotor is set to spin at a 2.5 Hz (1.11 ms/degree of rotation). It can be set to trigger the laser from any polarizer angle. Timing in the polarotor is controlled by a timing disk with four indicators 90° apart, each of which can trigger the laser. Thus, once we set the rotation angle such that trigger 1 corresponds with "parallel", triggers 2, 3, and 4 correspond automatically to perpendicular, parallel, and perpendicular respectively, each 90° of rotation after the other. One consequence of this arrangement is that although there may be error in the absolute angle during each measurement, the parallel and perpendicular channels are exactly 90° apart.

In practice for the CRL, the reference rotation angle is controlled by the "start delay" time between the trigger pulse from the polarotor and the time the laser fires. We perform a calibration to define the angle of rotation at which the polarotor's prism needs to be in order for the parallel measurement channel to correspond with the polarized backscatter returns from the laser beam. In this manner we effectively align our polarizer with the plane of polarization of the laser as it is transmitted to the sky. Start delay calibration measurements are made in a dark, clear sky, where the depolarization parameter should be approximately 0. Measurements are taken in both depolarization channels for several minutes at each of many start delay settings. The optimal start delay setting occurs at the location of maximum contrast, where as much of the clear sky signal as possible enters through the parallel channel, and as little as possible enters via the perpendicular channel. In Eureka we work on a campaign basis, so it is not possible to wait months for the perfect clear day to do this measurement. We reduce the effects of any clouds, aerosols, or other depolarizing particle, by using only values from a certain altitude range of interest which does not include clouds, and by using the Polarization-Independent Rayleigh Elastic channel as a check to indicate times where our calibration may be invalid. It is helpful to divide the photon count rates of parallel and perpendicular by the count rates in the polarization-independent channel to eliminate the effect of laser power variations. If the sky depolarization is not truly zero for this test, some systematic error will be induced in all subsequent calibrated measurements of sky depolarization.

From the test, we obtain two cosine curves of photocounts as a function of polarotor start delay: One for parallel, and the other for perpendicular, 180° out of phase with one another. A fit to each curve allows us to find the start delay value.

Our start delay test is carried out in start delay steps of 2560 µs, with allowable start delay values between 20 µs and 419000 µs, as this has sufficient resolution to determine the correct start delay while requiring few enough settings that the calibration may be carried out during a single night. This test needs to be repeated any time the polarotor is uninstalled or reinstalled into the polychromator.

**2.4  Effects of polarotor rotation angle errors on depolarization measurements**

If we consider the angular start delay error to be equal to one-half of our measurement step size of 2560 µs, corresponding to 1.408° of error, we have the following errors in d: The worst case scenario is that for which d = 0. There, a 1.408° angle error corresponds to an error of ±0.0006 . For d = 1, there is no uncertainty in d from this source. For a typical value of d = 0.2 the uncertainty is ±0.0005.

Next, we investigated the effect of the rotation of the polarizer during each laser shot measurement and found that has negligible impact on our interpretation of d. Again considering the worst offending case, that in which d = 0: A two-way maximum photon travel time for 16000 altitude bins is 0.0008 s, during which the polarotor rotates through 0.88∘ . The induced change in d (in units of d) results in an uncertainty in d such that d = 0.0000±0.0005. The more diagnostic case d = 0.2 results in d = 0.2 ± 0.0004.

When combined, the rotational errors of the polarotor considered in this section contribute a maximum of total error of 2.288∘, making the error from these sources approximately equal to ±0.003 for typical measured values of d. To put this in context, other errors (see Sect. 8) contribute ±0.1 to ±0.2.

R2: P3L11: The use of a single PMT for the parallel and perpendicular measurement is presented as an advantage. However, in Section 9.2, *'using two depolarization PMTs would allow for different gain settings individually optimized for the parallel and perpendicular channels'* is presented as a hardware change that would improve the perpendicular signal. May the authors clarify which is the best configuration?

Response: The "best" configuration depends on the initial physical conditions one has available in a given lidar, and on the overall goals for that instrument. The best configuration for CRL as a whole is to leave the optics as-is, with the polarotor, and to use a different calculation technique which nicely gets around the hardware problem. The optimal hardware change for depolarization measurements would be to remove the VLPW optic and/or insert one of the compensating optics at the entrance to the polychromator. Thus, the suppression of the perpendicular-polarized beam would be less, and we would take better advantage of the polarotor and single PMT setup. This would have a much greater effect than changing the gain settings on the PMTs. However, that is not a compromise we are willing to make, considering all other scientific goals for the instrument. We could, as specified, use two depolarization PMTs instead. On its own, the benefits from the increase in perpendicular photon counts would outweigh the negatives of having to calibrate out the different gain settings of the PMTs. In that sense, two PMTs would be preferable to the polarotor.  However, CRL currently has as many PMTs as it has electronic support to handle. In order to add an extra PMT, we would need to install an additional Licel PMT module, more Licel Transient Recorder modules, have the electrical power and computational power to support this, etc. If these limitations were not considerations, *and* we still did not want to remove the VLWP, or add a compensating optic upstream, then adding a 2nd PMT would make sense. For any other lidar, the particular issues will be different, and the net gain of either option needs to be evaluated on an individual lidar basis.

Action: Change the sentence "Fourth, using two depolarization PMTs would allow for different gain settings individually optimized for the parallel and perpendicular channels." such that it now reads "Fourth, using two depolarization PMTs would allow for different gain settings individually optimized for the parallel and perpendicular channels but this change to the configuration is not possible with the current electronics.".

Author comment P03 L19-21: There is an error in the sentence "This trigger signals the laser to fire and the detectors to record every time the prism rotates through 450 degrees, which corresponds to the prism's acceptance plane being rotated by 90 degrees." We would like to correct this. In actuality, the polarotor has triggered the laser to fire every 90 degrees of rotation during the work in this paper (and continues to operate as such currently).

Action: Change the sentence "This trigger signals the laser to fire and the detectors to record every time the prism rotates through 450 degrees, which corresponds to the prism's acceptance plane being rotated by 90degrees" to read "This trigger signals the laser to fire and the detectors to record every time the prism rotates through 90 degrees."

R3: P03 L21: "90 degree" – a comment on the rotation accuracy of the polarotor should be included? How does this accuracy influence the results?

Response: See response to item R3: P03 L09-11, above.

Action: Same as action for item R3: P03 L09-11, above.

R3: P03 L22: "Two recording buffers are used in the Licel Transient Recorder, one for parallel and one for perpendicular photocount profiles." - is this comment necessary?

Response: It is probably not necessary. We do not feel that it detracts from the manuscript. We intend to keep this comment, as this is quite an instrument- and hardware- focused paper. It will give some indication of what is needed to set up a similar system both for those lidar groups already working with Licel instrumentation, and those for whom the Licel products are unfamiliar. The paired recording buffers operate quite differently to the buffers used for every other measurement PMT in the CRL.

Action: None.

R2: P3L23: Does licel report the temporal stability of this device?  [question in reference to the polarotor]

Response: Licel does not provide information about the rotation accuracy nor the stability of this module. See response to item R3: P03 L09-11, above.

Action: Same as action for item R3: P03 L09-11, above.

R3: P03 L23-24: "The extinction ratio of the polarizer was characterized by the manufacturer to be 5X10^-5 or smaller " - it is also given by the accuracy of the rotation angle. This section should better describe the polarotor.

Response: See response to item R3: P03 L09-11, above.

Action: Same as action for item R3: P03 L09-11, above.

R3: P04 L0: Figure 1 shows a collimating lens between points 2 and 3. Could the authors describe the purpose of this lens.

Response: The collimation lens shown between locations "2" and "3" in Figure 1 was present in the lidar before the design and installation of the depolarization channel. It was not added during the depolarization modifications to the lab. Its purpose is to collimate the light entering the 607 nm channel. It also happens to slightly converge the light beam entering the 532.1 nm depolarization channel's polarotor, which is convenient. The beam diameter is too large at the location of the pellicle beamsplitter to entirely fit through the polarotor's prism. From the action of the existing collimating lens, by the time the depolarization channel's beam reaches the top of the polychromator, it has reduced in diameter sufficiently to fit entirely through the acceptance area and angle of the polarotor's prism. Thus, by placing the polarotor as far as was practical from the collimation lens (i.e. by mounting the polarotor on top of the polychromator rather than inside it), we take advantage of the reduced beam size for this channel and do not need an additional collimating optic between the pellicle and the polarotor.

Action: Modify caption of Figure 1 to include the sentence: "The new hardware consists of the pellicle beamsplitter, the Polarotor, and the interference filter, focusing lens, and PMT, to the right of the polarotor."

R3: P04 L08: could you provide more detail on how was this channel characterized

Response: Reference to (Doyle et al., 2011; Nott et al., 2012) is given on page 2, line 13. The reference to (Nott et al., 2012) is repeated on page 3, line 14. These references describe the characterization of the other lidar channels in detail. We will repeat the references here.

Action: Add references for the sentence "The 607nm channel optics were already well aligned and characterized at the time of depolarization installation." such that it will now say "The 607nm channel optics were already well aligned and characterized at the time of depolarization installation \citep{Doyle2011,Nott2012}."

R3: P04 L08-11: "installation. Therefore, a regular plate beamsplitter or dichroic mirror could not be used to pick off the light for the depolarization channel; this would have translated the transmitted 607nm light too much, and

the downstream channel would have had to be realigned." - But still a collimating lens was introduced before the 607/532p and c channels. How does this lens affect the alignment? Why is the collimating lens placed in front of the pellicle beamsplitter and not after the splitter? This way the lens would only affect the 532.1nm channel.

Response: This lens pre-dates the depolarization channel. See response to item R3: P04 L0, above.

Action: Same as action for item R3: P04 L0, above.

R2: P5L1: 'atop' -> typo?

Response: Not a typo. This word is as intended, and is appropriate in this sentence. To increase clarity, we will rephrase this.

Action: Change "atop" to "on top of"

R3: P05 L05: "angle of 15degree" – please mention "full angle"

Response: We understand these two terms ("acceptance angle" and "full angle") to be equivalent. We will modify the term to include "full" in the description of the angle.

Action: Replace sentence "The polarotor has an acceptance diameter of 20 mm, and an acceptance angle of 15 degrees" with a new sentence which reads "The polarotor has an acceptance diameter of 20 mm, and a full acceptance angle of 15 degrees".

R3: P05 L06-07: "It is also convenient to have the cables from the polarotor to the electronics rack be accessible without the need to open the polychromator." – is this comment really necessary?

Response: The authors agree that this is not strictly necessary. However, given the nature of this installation, in which practical concerns for existing equipment proved to be the limiting factors, it seems fitting to include comments of a practical nature when discussing motivation for decisions regarding the placement of new components.

Action: None.

R2: P5L15: I suggest to include a comment about the different definitions (Cairo, 1999). The concept of 'photons polarized perpendicular' is 'old-fashion' and has already demonstrated wrong. Please, revise the paper considering the explanation of Gimmestad 2008.

Response: We will do so, and follow the notation given in Gimmestad 2008. Thank you for calling our attention to the Cairo 1999 paper. We will include a comment.

Action: The introduction is now considerably changed, and the literature review has been expanded. The reference to Cairo 1999 is now included in Section 1.1, particularly in the new text which reads: "A variety of expressions for "depolarization" are described in Cairo et al. (1999). The $\delta$ described in the current paper is most closely related in meaning to the Pal and Carswell (1973) "volume linear depolarization ratio" cited therein, but it is not strictly equivalent because no claims are made here about the connection between $\delta$ and backscatter coefficient. Rather, $\delta$ is defined here only as a function of measured quantities. Gimmestad (2008) provides motivation for moving away from all $\delta$ descriptions, pointing out that d is consistent with the rest of optical physics and scattering theory. Expressing depolarization as d has since been adopted in such publications as Hayman and Thayer (2009, 2012) and Neely III et al. (2013). Results in the present paper will be provided in terms of both d and $\delta$ so that readers working under either paradigm can readily make use of the figures and calculations.".

R1: Page 5 lines 16-17 & Eq. (1): change the words to be correct and change the equation to be delta = k(S_perp/S_para). That's all has ever really been. The signals are all we have to work with!

Response: This has been changed. The entire contents of Subsection 3.1 has been altered to focus on the expressions in terms of d, with the tie-in to the older $\delta$ as the add-on. The edits for items R2 P5 L15 and R3: P06 L12 are also included. All of the material for Subsection 3.1 has been moved to the expanded literature review section in the introduction.

Action: See item R1: Page 2 lines 17-21, which provides all new text for this section.

R3: P05 L19: beam,

Response: Agreed. As a result of item R1: Page 5 lines 16-17 & Eq. (1), the entirety of subsection 3.1 has been reworked and moved to the introduction, but this edit for item R3: P05 L19 has been included in the modifications. See that item for the new wording.

Action: A comma will be added after the word "beam". in the sentence "In the event that the atmosphere does not depolarize the beam there will be no photons backscattered with polarization different than the transmitted light, and therefore \delta = 0. In the case of complete depolarization, \delta = 1.". See item R1: Page 2 lines 17-21, which provides all new text for this section.

R3: P06 L12: "and solving for k." – consider reformulating

Response: We will change the wording. As a result of item R1: Page 5 lines 16-17 & Eq. (1), the entirety of subsection 3.1 has been reworked and is now included in the introduction. This edit for item R3: P06 L12: has been included in the modifications.

Action: See item R1: Page 2 lines 17-21 for new wording in the introduction.

R3: P06 L21: "allowing for optical effects in the upstream optics" – consider reformulating

Response: We have reformulated this.

Action: Change the sentence "The following section uses Mueller Matrix algebra to re-derive the equation for d, allowing for optical effects in the upstream optics." to read "The first goal for this paper is to use Mueller Matrix algebra to re-derive the equation for $d$, including calibration terms which describe the arbitrary optical effects of the upstream optics. These terms allow the collection of upstream optics to be represented using the most general single Mueller matrix possible (See Sect. 3.1, Eq.12 )".

R3: P07 L30 (3rd line on page): Ilaser purity - was this verified by any experimental measurement. According to Freudenthaller 2016, laser emission is not 100% polarised.

Response: The laser's polarization purity was investigated from a location partway through the transmitter system. A cubic polarizing beamsplitter was placed in front of the power meter, and the laser was redirected through it. The laser power was measured as a function of beamsplitter angle. The results were as one would expect for a reasonably polarized laser source. This is not an optimal test because five mirrors (including the diverging mirror) and the laser-quality roof window insert are neglected. However, it was the best test which could be carried out at the CRL lab during the years this work was accomplished.

Action: a) Change the sentence <<The CRL laser emits horizontally linearly polarized light Ilaser = Ilaser[1 1 0 0]'>> to read <<The CRL laser emits horizontally linearly polarized light, nominally Ilaser = Ilaser[1 1 0 0]'>>.
b) After the sentence: "Therefore, Ilaser can be considered the same for both channels for each measurement pair.", include several new sentences which read " The polarization of the laser was measured with a cubic polarizing beamsplitter on a kinematic mount placed in front of a power meter which is permanently mounted on the optical table. A mirror wheel redirects the beam into the power meter from a location partway through the transmitter. The precise location can be found in Nott et al. (2012). Thus, the polarization tests take into account all but four transmission optics on the optical table, as well as the one extra optic of the redirection mirror. This location is not ideal, but is accessible in this installation. The first three elements of the resulting normalized Stokes vector [I , Q, U, V ]′ were calculated at two laser powers. At lower power: [I,Q,U,V ]′ =

[1±0,1.00±0.02,0.02±0.01,V ±σV]′. At higher power: [I,Q,U,V ] = [1±0,0.97±0.01,0.00±0.01,V±σV]′. Our test was not directly sensitive to the value of V. Since Q^2+U^2+V^2 ≤I^2,we can surmise that |V | ≤ 0.243. We consider Ilaser = Ilaser[1 1 0 0]′ to be a reasonable laser Stokes vector for the calculations in this paper."

R3: P07 L04: - rotation of the plane of polarization of the laser with respect to a reference (usually the polarizer separator) must also be accounted for. For a rotation of Mreceiver and Mtransmitter with respect to the laser polarization, the collected polarized light could be also altered. Some comments must be included. The study should also take into account these effects (Bravo-Aranda, J. A. 2016)

Response: This is accounted for with the initial timing delay setting of the polarotor during the installation and characterization procedures of the polarotor. Provided that the polarization of the transmitted laser beam as it exits the roof window remains a linear polarization, it is not important for our method that we know how this relates to the laser's initial polarization plane in the real world. Instead, we take the outgoing to-the-sky polarization plane as the reference polarization for all analyses, and call it "horizontal", or "parallel". Then we initialize the polarotor such that its rotation angle at maximum backscatter in a clear sky is the "parallel" setting for the polarotor. The minimum is 90 degrees from this. A more logical location to include these details is in Section 2, so some text will be added there to address this and other related issues raised by the reviewers.

Action: See item R3: P03 L09-11 for the full new text of sections 2.3 and 2.4 about the start delay test and its uncertainty.

R3: P07 L05: "The optical backscattering effects of the atmosphere can be described as Matm." - Mtransmitter is initially mentioned but left out further in the study. According to Nott et al 2012 - the number of emission optics is significant and could influence the polarization purity of the emitted light (this added to the assumption that the laser unit is emitting 100% polarized light). Comments should be included to explain these assumptions.

Response: See response to item R3: P07 L30 (3rd line on page), above.

Action: Same as action for item R3: P07 L30, above.

R1: Page 7 line 8 – "in equal numbers" is redundant and should be deleted.

Response: This is the wording used in (van de Hulst, 1957). The authors see that this is redundant wording.

Action: "in equal numbers" will be deleted.

R3: P07 L12: "The gain factor is not stable long term, but for any given minute of data it will be constant for both channels" - please give more details [Note from the authors: This is referring to gain factor b which makes up part of the atmospheric scattering matrix Matm.]

Response: Matm is a matrix which has been normalize by a factor b. As the atmosphere changes above the lidar, the absolute number of photons scattered back to the lidar will change, and this is reflected in Matm via a change in the value of b. However, both channels (parallel and perpendicualr) are measuring simultaneously. Unless b is alternating between two physical states repeatedly at precisely a multiple of 10 Hz, both channels will experience the same physical atmospheric state *as each other* for a one-minute measurement. (i.e. there could be a change in the atmosphere over the one-minute, but both channels would experience this in precisely the same way, in terms of b).

Action: None.

R1: Page 8 line 1 – isn't "to the sky" redundant/ unnecessary?

Response: We do not believe this to be redundant, because there is also the alignment on the optical bench to consider. The overlap function would vary with changes in optical bench alignment as well of course, but this does not change frequently and is not the issue at hand when discussing the long-term stability of the overlap function O(z) in this context. The lidar's beams are realigned to the sky approximately daily, while on-the-bench

alignments occur less than once per year. However, Reviewer 1 was not the only reviewer to request a change in wording, so we will address this to make it clearer.

Action: "to the sky" will be removed from the present sentence.

R3: P08 L01-02: "The overlap varies with changes to the lidar's alignment to the sky" – consider reformulation

Response: See response to item R1: Page 8 line 1, above.

Action: See response to item R1: Page 8 line 1, above.

R3: P08 L07-09: "During setup, the "parallel" analyzer position was also oriented such that it can be represented as a horizontal polarizer (by aligning the parallel direction with the direction of maximum signal in a low depolarization sky)." - As stated before, also the rotation of the optics (Mtransmitter and Mreceiver) must also be accounted for. For Mtransmitter and Mreceiver rotation with respect to the laser polarization, collected polarized light could be altered.

Response: See response to item R3: P07 L04: and R3: P03 L09-11, above, and item R3: P08 L19-21 part d:, below. We did not have the capability to verify ellipticity in our measurements. We will do so in the future if the option presents itself.

Action: See item R3: P03 L09-11 for the full new text of sections 2.3 and 2.4 about the start delay test and its uncertainty.

R3: P08 L19-21 part a: "These Mueller matrices combine to make an overall equation for each channel which describes the action of all optical components on the light, and results in Stokes vectors (shown in full in Eq. (9)) and I (which differs from only by two minus signs in the polarizer matrix):" The author must include a comment on the effects of the polarotator rotation uncertainties.

Response: See response to item R3: P03 L09-11, above.

Action: Same as action for item R3: P03 L09-11, above.

R3: P08 L19-21 part b: An extra comment is also required for the rotation of the laser polarization purity.

Response: See response to item R3: P07 L30 (3rd line on page):, above.

Action: See response to item R3: P07 L30 (3rd line on page):, above.

R3: P08 L19-21 part c: The author provided information on how the rotation of the laser plane of polarization is corrected but no information on possible rotation of the receiving optics is provided.

Response: Related to item R3: P07 L04, above, and R3: P08 L19-21 part d, below.

Action: See responses to item R3: P07 L04, above, and R3: P08 L19-21 part d, below.

R3: P08 L19-21 part d: Again more detail should be given to the "... action of all optical elements ...". Many optical components are excluded (transmission and part of the receiving optics) - this should be clearly mentioned. Suggest changing to: "A simplified version of the overall equations combine to make an ....... " - ...... including ideal laser polarization purity, no emission optics, no laser rotation, no optics rotation, no retardation effects, ideal polarizer and so on. How will these simplifications affect the final results?

Response: Related to item R3: P07 L04, above, and others. We feel that this is not the appropriate location in the paper to address this issue. We agree that there are simplifications for CRL in the field tests of Section 5 and Section 6: Skipping the telescope and roof window in the receiver, and we are not 100% sure that the laser light remains linearly polarized as it exits the roof window of the transmitter. However: a) Rotation of and initial laser

purity of polarization are accounted for (via tests of laser purity, and by aligning the receiver to the *light as it exits the roof*, not the light *as it exits the laser;* b) Receiver matrix M accounts for all receiver optics' rotation, retardation, non-ideal polarizer (except for the polarotor's prism, which is otherwise accounted for in the matrix equation), etc. in Equation 9. At the present location in the paper (Equation 9), the compromises and assumptions made in Sections 5 and 6 are not yet relevant. With better materials at our disposal, equation 9 is still correct - it's only Sections 5 and 6 that would change.

In locations such as P11 L24 - 29, many of the simplifications that we have had to make are spelled out. Further, we demonstrate that *for all the matrix elements which matter for the calculation of d*, most of these effects are nil, for most of the optics involved. We do not find all matrix elements, but we do not need to - we only need to understand the signals in our PMTs, which we do. It is regrettable that some optics had to be omitted from the analysis.

Response: We will make the point more clearly about which aspects of the calibrations are simplified at appropriate locations in the text. For example, see item R3: P15 L24-25.

Action: For example, see item R3: P15 L24-25.

R3: P09 L02-03: "lidar, we solve for the depolarization parameter d to learn about the atmosphere" - consider reformulating

Response: We will shorten this sentence, and combine with the following paragraph.

Action: Replace the sentence "Using the signals Spara and Sperp from above for the complete matrix description of the lidar, we solve for the depolarization parameter d to learn about the atmosphere." and the paragraph break following it so that the text between Eqn 11 and Eqn 12 will now read:  "Using the signals Spara and Sperp from above for the complete matrix description of the lidar, we solve for the depolarization parameter d. The simplest method for combining lidar signals Spara and Sperp into an equation to solve for the depolarization parameter comes from creating the quantity" (no period at the end, because Eqn 12 forms part of the sentence).

R3: P09 L04: "to solve for the" - consider reformulating [Full sentence: "The simplest method for combining lidar signals Sk and S? into an equation to solve for the depolarization parameter comes from creating the quantity..."]

Response: The rest of the sentence is awkward as well. We will rephrase this.

Action: Remove the portion of the sentence "The simplest method for combining lidar signals Sk and S? into an equation to solve for the depolarization parameter comes from creating the quantity..." and replace it with a portion which reads "The simplest method for combining lidar signals Sk and S? into an equation for depolarization parameter comes from creating the quantity...".

R3: P09 L10: Another option would be to use the three signal calibration since the lidar instrument is able to measure the total, parallel and cross 532.1nm components. This will be a nice add-on to this study: a comparison between the current calibration and the suggested calibration.

Response: The authors are pleased that Reviewer 3 brings up this point. There is a follow-on paper nearly ready for submission which takes advantage of the polarization-independent ("total") 532.1 nm Rayleigh Elastic channel. It turns out that the calibrations are not as simple as might be expected (in fact, the best versions rely on the results of the present paper), but they do work. This point is raised later in the paper, in Section 9.1.

Action: None at this location in the paper.

R3: P09 L14: "would" to would be

Response: Typo. We will address this, and also change the tense of the sentence.

Action: change the sentence "In the case that CRL met these conditions, it would acceptable to use Eq. (5) in further calculations for this lidar." to read "In the case that CRL meets these conditions, it will be acceptable to use Eq. (2) in further calculations for this lidar."

R3: P09 L21: Title "Polarization and Depolarization generating calibration optics" - consider reformulating

Response: The authors agree that this wording is awkward, and will revise it.

Action: The name of this section will be revised to "Optics for generating polarized and depolarized light during calibrations".

R3: P09 L25: "It is placed immediately downstream of the focus stage and" - please indicate on the picture

Response:  This is indicated in more detail on Page 11, line 16. We will specify the location here on Page 9 as well as it precedes the later description.

Action: The specification in brackets will be added to this sentence: "... downstream of the focus stage (location label 7 in Fig. 1)..."

R3: P10 L05: Product to product

Response: Agreed.

Action: Will change the capital "P" in "Product number" to a lower case "p".

R3: P10 L08-09: "This product was mounted in such a way as to be held relatively taut in a frame, or held gently in place by other mechanical means." – Too vague, consider reformulating

Response: The precise choice depends on the test we are using the glassine for at any given time. Specific examples of calibrations using the depolarizing sheet are given later in the paper. For the test in Section 5.1 (Physical setup of the rotating polarizer test used at CRL), for example, a foamcore frame was used. There is a photograph of this in McCullough's PhD thesis. Conversely, for the test in Section 6.2 (Physical setup of unpolarized laser calibration to determine M10/M00 and k) a much larger sheet was placed over the roof window. A detailed description of this is written in that section.

Action: Change sentence "This product was mounted in such a way as to be held relatively taut in a frame, or held gently in place by other mechanical means." to read "This product was mounted in such a way as to be held relatively taut in a frame, or held in place by other mechanical means, depending on the specific calibration test. See \ref{McCullough2015 PhD thesis} for photographs of the arrangements used for the calibrations in this paper."

R1: Page 10 Lines 10 and 15 (provided as a general comment): Also, when describing waxed paper as a depolarizer, it would be good to mention that it is also a highly scattering material, so that when the entire roof hatch window is covered with it, the received lidar signal is greatly reduced.

Response: We will add the comment suggested by the reviewer, in two places.

Action: a) Refer first to the response to item R3: P10 L08-09:, which edits the first part of the relevant paragraph. b) Following the new sentence "This product was mounted in such a way as to be held relatively taut in a frame, or held gently in place by other mechanical means, depending on the specific calibration test. See \ref{McCullough2015 PhD thesis} for photographs of the arrangements used for the calibrations in this paper.", include a new sentence which reads "Glassine is highly scattering material, so when the entire roof hatch window is covered with it, the received lidar signal is greatly reduced". c) Following the sentence which ends "verification for each application is advisable.", add a new sentence which reads "As for glassine, waxed paper is a highly scattering material which greatly reduces received lidar signals when they are measured with the waxed paper in place."

R1: Page 10 line 11 - delete second "this".

Response: Agreed.

Action: The second "this" will be deleted, to make the sentence read "...optic (in this case, glassine waxed paper), is:"

R3: P10 L24: Fig 2 - why do we have such different photoncounts for the two channels? Why do you have a setup counting so few photons per time-bin? Could this be improved by expanding the time bin?

Response: This is addressed in great detail in Section 7, "Determining contributions of individual optics". The main contributor is the Visible Long Wave Pass optic which suppresses the perpendicular signal rates far more than it suppresses the parallel signal rates. This optic was installed in the lidar before the decision was made to add depolarization measurement capability. Consequently, the polarization qualities of this VLWP optic (and it is highly polarizing) were not a factor in its selection. We did not know its polarization characteristics before beginning the upgrades for depolarization in 2010, and were surprised to find that it had such an impact on the measurements. In order not to compromise any "existing" measurement capability of the lidar, we decided not to replace this optic with something less polarizing, but instead to work around this problem with calibrations (shown in this paper) and a new calculation method (which will be shown in a follow-on paper to be submitted to AMT very soon). The suggestion to expand the time bin is a good one, however this would overwhelm the polarization-independent 532.1 nm channel (results in the follow-on paper). Further, we wanted to keep as many controllable parameters as we could the same as during operating conditions.

Action: None

R1: Page 11 line 3 – change "the all the" to all the.

Response: Agreed.

Action: The first "the" will be deleted, to make the sentence read "...initially depolarize all the backscattered light..."

R3: P11 L03: "A 1m depolarizing optic to initially depolarize the all the backscattered" to A 1m depolarizing optic to initially depolarize all the backscattered...

Response: Agreed. See response to R1: Page 11 line 3.

Action: See response to R1: Page 11 line 3.

R3: P11 L05: "properties: a 1m diameter circle" - Consider reformulating

Response: Will reformulate. See response to R3: P11 L05-07, below.

Action: See response to R3: P11 L05-07, below.

R3: P11 L05-07: "which could be held completely flat, which could survive the harsh outdoor conditions of Arctic winter, which could be easily and repeatably rotated to the appropriate orientation, and which had sufficient optical polarization quality." – to -> which could be held completely flat and could maintain within required characteristics even in harsh outdoor conditions of Arctic winter. The depolarizing optic must be easily and repeatable rotated to the appropriate orientation and should have sufficient optical polarization quality.

Response: The authors agree that the wording can be improved.

Action: The sentences "The problem is that no feasible polarizing optic had the required properties: a 1m diameter circle, which could be held completely flat, which could survive the harsh outdoor conditions of Arctic

winter, which could be easily and repeatably rotated to the appropriate orientation, and which had sufficient optical polarization quality." will be changed to "The problem is that no feasible polarizing optic has the required properties. We require a 1 m diameter optic which can be held completely flat and which can survive the harsh outdoor conditions of Arctic winter. The polarizing optic must be able to be easily and repeatably rotated to the appropriate orientation, and should have sufficient optical polarization quality. "

R3: P11 L10: "held between the telescope's tertiary mirror and the focus stage worked better" - please indicate on Fig 2 (label 7?)

Response: This is location label 8 on Figure 1. The light enters the focus stage from the telescope, most recently having reflected from the tertiary mirror.

Action: Bracketed text added to read: "...telescope's tertiary mirror and the focus stage (location label 8 in Fig. 1) worked better..."

R3: P11 L12: "By sacrificing the inclusion of both" - what do you mean by sacrificing the inclusion? [Full sentence reads: "By sacrificing the inclusion of both the lidar's telescope and focus stage in the calibration, the rotating polarizer test becomes possible at CRL."]

Response: We would like to include all optics in this test. However, if we try to do that, the test is impossible for practical reasons at CRL, with our resources. We cannot make the test we would really like to. The compromise is to leave three things out of the test (the "both" in this sentence - which was a mistake, as there are 3 and not 2 items): 1. The roof window, 2. the lidar's telescope and 3. the lidar's focus stage. If we begin the test downstream of these optics, the test becomes possible. Thus, we're "sacrificing" some components of our perfect but impossible test in favour of a different test which is imperfect, still useful, and possible. The authors recognize that the wording is confusing for several reasons and will correct this.

Action: Remove the sentence "By sacrificing the inclusion of both the lidar's telescope and focus stage in the calibration, the rotating polarizer test becomes possible at CRL". Replace with the following sentence: " By removing the roof window, the lidar's telescope, and the focus stage from consideration in the calibration, the rotating polarizer test becomes possible at CRL.".

R3: P11 L15: "It" should be replaced by -> The cube beamsplitter

Response: That would indeed be more specific.

Action: Replace "It" by "The cube beamsplitter" in the sentence "It can be rotated precisely and is stably mounted on a kinematic rotation mount..." to make it read "The cube beamsplitter can be rotated precisely and is stably mounted on a kinematic rotation mount..."

R3: P11 L17: "test, there is no advantage to using lidar returns as the light source"- usually when using the lidar return as the light source we take into account the height dependency of the optics (Freudenthaler 2016): Light collected from different heights have different lightpaths and different incidence angles in the receiving modules. this must be taken into account since the lidar optics are not polarization independent - see fig 2 - count number is strongly different for the same initial light source. This issue should be reconsidered.

Response: Yes, this statement was an oversight in writing this manuscript. The authors are aware of such effects, and these become important in the follow-on paper (in which there is a very strong height-dependence in a calibration term needed for the alternate depolarization calculation method). In the case for the tests of Figure 2, the advantages of a lamp vastly outweigh the advantages of using the laser light. Namely, the test using the lamp is actually possible. While there are in principle advantages to including any height dependent effects as suggested by the reviewer, in practice the glassine waxed paper depolarizer attenuates the laser signal too much to be of use in the practical case for a laser of CRL's power. We will rewrite this small section, including some of the wording included by the reviewer.

Action: Remove the sentence "Because the telescope and focus stage are being omitted in the test, there is no

advantage to using lidar returns as the light source; a current-stabilized constant lamp source provides more signal with better control of the experimental setup." and replace it with the following several sentences, which read "The telescope and focus stage are being omitted in the test. If the lidar return were used as the light source, we could still take into account the height dependency of the optics. Light collected from different heights have different lightpaths and different incidence angles in the receiving modules (Freudenthaler 2016), and this may have effects on the calibration. While there are in principle advantages to including any height dependent effects, in practice the glassine waxed paper depolarizer attenuates the backscattered laser signal too much to be of use in the practical case for a laser of CRL's power. Thus, there is little advantage to using lidar returns as the light source; a current-stabilized constant lamp source provides more signal with better control of the experimental setup."

R3: P11 L25: "optics are contributing any non-simple-gain effects to the signals" - please reformulate. It is not clear.

Response: We will clarify this with some specifics.

Action: Change sentence "Omitting the first optics in the detector chain means that this test does not give us a whole-system understanding, although it does allow us to say with certainty whether the downstream optics are contributing any non-simple-gain effects to the signals." to read "Omitting the first optics in the detector chain means that this test does not give us a whole-system understanding, although it does allow us to say with certainty whether the downstream optics are contributing any effects other than relative gain (e.g. retardation, rotation, etc.) to the signals."

R3: P11 L26-27: ""If we consider the optics and detector starting after the focus stage, can we use the simplified Eq. (4) and Eq. (5) to find the calibration constant, and then to determine depolarization ratio and depolarization parameter" - the three signals calibration would include all optics in the receiving unit of the lidar instrument. This should be a viable option for this study.

Response: See response to item R3: P09 L10, and Section 9.1. Note also that the "third" signal (polarization-independent Rayleigh Elastic) shares only some, but not all, optics with the depolarization channels. Notably, the depolarization channels see transmission through the VLWP optic, while the polarization-independent channel sees a reflection off this optic. The depolarization channel then also includes a collimating lens and pellicle beamsplitter which are not included in the polarization-independent channel's light path. On the other hand, the polarization-independent channel experiences one extra partially reflective mirror just before its filters: This picks off part of the light to be used for the Rotational Raman channels downstream. Thus, a three signals calibration for CRL is not quite straightforward.

Action: See response to item R3:P09 L10.

R3: P11 L27 parameter?".

Response: We will add a period to the end of the sentence.

Action: Add a period to the end of this sentence, after the question mark and quotation mark: <<We pose the question, "If we consider the optics and detector starting after the focus stage, can we use the simplified Eq. (4) and Eq. (5) to find the calibration constant, and then to determine depolarization ratio and depolarization parameter?">> such that it reads "We pose the question, "If we consider the optics and detector starting after the focus stage, can we use the simplified Eq. (4) and Eq. (5) to find the calibration constant, and then to determine depolarization ratio and depolarization parameter?".>>

R3: P12 L07: "There are very similar equations for the case in which we use backscattered laser light rather than lamp light." - This is not fully correct if we consider the laser polarization purity, laser rotation, emission optics

Response: Laser rotation and emission optics rotation should not matter, because our "parallel" direction is not aligned with "real horizontal", but rather has been calibrated to be parallel to the plane of non-depolarized backscattered lidar light. Laser purity is another issue altogether, but is not one which we can measure to better

precision than we can make our depolarization ratio or depolarization parameter measurements, and our calibrations. We concede that the lamp light does not allow for altitude-dependent effects like we would be sensitive to with laser light, as per response to item R3: P12 L14a.

Action: None.

R3: P12 L11-12: "The absolute angles were determined in post-processing, such that the maximum in the parallel channel is 0." - This data combined with information on the actual position of the cube could be used to extract the laser rotation relative to the parallel detection. Was this study performed?

Response: Our initial transmitted light is taken to be that which exits the roof, so such a study would be interesting but not necessary for the present work. It would be highly useful, however, in the event that we can modify any optical elements in the system - having knowledge of our absolute polarization state in the lab reference frame would be very useful in that situation.

Action: None

R3: P12 L14a: 7.5m - if the light source is a lamp placed on the telescope frame, the range dependence should be inexistent. Under these assumptions, the altitude bin could be extended to several hundred meters or even km to improve the photocounts number. By using this method Fig 2b would have a much better fit in the n*pi regions (n=0,1,2).

Response: This is a very reasonable comment. This would likely prove to be a better approach. As the approach taken here was sufficient for the purposes of this study, we will not re-analyze these data at this time. However, in future re-analyses we will take this suggestion into account.

Action: None.

R3: P12 L14b: "There is approximately a 2 degree or 0.035 radian uncertainty" – Is this uncertainty taken into account in the study?

Response: This is not taken into account mathematically. The measurements were made at a spacing of 10 degrees between measurements. Thus the uncertainty in angle is far less than the measurement spacing. Further, we assign absolute angles in post-processing, so any uniform shift in the angles (e.g. overshooting the angle marks consistently in the same direction by the same amount) would be eliminated before the plot in Figure 2 is produced. In the event that the angles truly are uncertain by 2 degrees, an examination of Figure 2 shows why this will not matter particularly for the two diagnostic criteria that we sought with this test (namely, that the signals went to zero appropriately, and that they were symmetric about their maxima). A 2 degree change in angle would move any given measurement point by about the width of the point shown on the plot (1/5 of the spacing to the next point over). Any change of this small amount would be well within the rather large \sigma uncertainty from photon counting statistics. The fit line would remain well within the uncertainty envelope. We note that Freudenthaler 2016 dealt at length with the specific issue of uncertainty due to angular error in a calibration optic. This paper was not available at the time of the initial analyses of our measurements (which occurred in 2013), and our measurements do not have the precision required to warrant such a more involved treatment of this uncertainty at this time for CRL.

Action: None.

R3: P12 L14c: "angles" - what angles? The calibrator or polarotor - it should be the calibrator, right?

Response: Yes, these are the rotation angles of the added calibration cube polarizer (which is placed at location 7 of Fig. 1).

Action: Modify the following sentence: "There is approximately a 2 degree or 0.035 radian uncertainty in the angles when doing this calibration." such that it now reads "There is approximately a 2 degree or 0.035 radian uncertainty in the rotation angles of the calibration cube polarizer when doing this calibration.".

R3: P12 L15: "The overall signals in S parallel far exceed the overall signals S cross." - Why? What does this indicate?

Response: The reason for different count rates is primarily the Visible Long Wave Pass filter (see Section 7), which suppresses the perpendicular signal. The VLWP effectively acts as a partial polarizer in the system. This phenomenon is excluded from consideration in such references as Freudenthaler 2016 (see Page 4229, section 13, "Assumptions and constraints of the model", assumption 5, which states "optical elements of the lidar do not depolarize"), but is the most important depolarization calibration characteristic of the CRL. It is this huge difference in count rates which cause the large value for k. This sentence is here to draw the attention of the reader to the different scales on the Figure 2 plots. At first glance, the figures look quite similar, but there are far fewer photons to work with for calculations in the perpendicular channel.

Action: None.

R3: P12 L15: "each" –> each of the two channels

Response: The authors agree that this should be more specific.

Action: Change sentence "Note the different scales for each: The overall signals..." to read "Note the different scales for each panel in Fig. 2: The overall signals..."

R3: P12 L16-17: "by allowing them as free parameters in a fit to these signals," - Does this return a unique solution?

Response: It does not return a unique solution. Even the simplified 3-calibration-parameter method from Eqn. 27 does not return a unique solution. A least-squares-fit to the physically allowed fits to Eqn. 27 is more productive, and is explored in Chapter 8 of (McCullough 2015).

Action: Change sentence "We could attempt to estimate all 7 unknown terms in Eq. (19) and Eq. (20) by allowing them as free parameters in a fit to these signals, but for CRL, there is a better way: ..." to read "We could attempt to estimate all 7 unknown terms in Eq. (19) and Eq. (20) by allowing them as free parameters in a fit to these signals, but this does not return a unique solution. For CRL, there is a better way to proceed:..."

R3: P12 L20-22: please give more details on why is this. [Full sentence reads: The signal equations, Eq. (19) and Eq. (20), are simplified a great deal if M02 = 0 and M12 = 0."]

Response: This is not a "deep" comment motivated by physics - just an observation that the equations *would* in fact be much simpler if certain terms within them happened to be equal to zero. It is a trivial sentence on its own, but but it gives the reader an idea about our motivation for carrying out certain calibration tests. irement for (M02 = 0 and M12 = 0) to be the case in any given lidar, so we go on to specify the necessary conditions under which it *will* turn out to be the case in a lidar. Then we test for these conditions using CRL's data, and those conditions *do* turn out to be the case for our CRL lidar. Thus, the equations do become simpler. The reason this matters is that we cannot actually solve the initial formulation of the equations 19 and 20 uniquely. But with our tests proving that M02 = M2 = 0, we arrive at equation 27. This we can (and have done, but not shown here) fit allowing 3 free parameters. This is better. Still, we do not arrive at a unique solution. So we carry on with similar reasoning for further constraining of the problem: Noting that the equations *would* be simpler yet *if* it turned out that M10 =M01 and M00 =M11 for a particular lidar. We test the conditions for that to be true, and it turns out to be true for CRL. Thus, equations 32 and 33 are valid for CRL and may be used and solved.

Action: Add a sentence to Page 12 line 20, preceding "This is the case if there is...". It will now read: "The signal equations, Eq. (19) and Eq. (20), are simplified a great deal if M02 = 0 and M12 = 0. There is no requirement that the constants M02 or M12 have these values for given lidar, but a test can verify whether it is the case: M02 and M12 will both be equal to zero if there is..."

R1: Page 13 Line 0: Fig. 2 – Photocounts were defined on the previous page as "photons per time bin per

altitude bin", This should be mentioned in the caption, and the readers would like to know how many altitude bins were included in the data.

Response: Good point. We will add this.

Action: a) In the caption for Figure 2, insert after this sentence: "Polarized calibration measurements as a function of incident light polarization angle." a new sentence, which reads " Photocounts are given per 1 min, 7.5 m altitude bin." b) In the caption for Figure 2, insert after this sentence: "The colourbar indicates the natural logarithm of the number of data points at each location, which is the result of producing a histogram for each angle theta." the following new sentence, which reads: " Data points from from all 16000 altitude bins, for all individual minutes of measurement at the angle theta (typically 3 to 5 min), is included in the histograms".

R3: P13 L01: "If our measurements are symmetric, with ..." - A detailed explanation must be included. What are the considerations that are the base of this result? This explanation must be detailed in the manuscript.

Response: These are mathematical results, not physical ones. The result of M02 = M12 is obtained by setting equation 21 and equation 22 equal to one another, and solving. The test for whether it is *appropriate* for the CRL to set equations 21 and 22 equal is given in Table 2. There, we show that Signal_para_theta=pi/4 is equal by measurement to Signal_para_theta_3*pi/4. Thus, equation 21 = equation 22, and therefore M02 = M12. A full mathematical description is given in (McCullough 2015, PhD. Thesis), but the authors were advised to remove the bulk of the algebra from the paper to be submitted to AMT.

Action: None.

R3: P14 L04-05: please reformulate also including detailed explanations

Response: Same argument as item R3: P13 L01, but referring to the perpendicular measurements, and equations (23, 24) rather than (21, 22).

Action: None.

R3: P14 L10-11: "The mean signal values are ... perpendicular" - is this relevant?

Response: There were requests in previous drafts of this paper that we include the values in the text as well as in the table.

Action: None.

R3: P14 L20: "the parallel and perpendicular channel signals each go to zero" - Fig 2 b shows the number of photon counts as a function of incident light polarization angle. The low amount of photons used for each point makes it difficult to perform a fit on the data. The mean values around n*pi (n=0,1,2) are clearly forced to be zero. We can see six values around pi that are zero. To clearly say that "........perpendicular channel signal each go to zero" it is mandatory to have a higher amount of photons. This may also apply to Fig 2 a. This could be accomplished if the author increases the time and height bins

Response: The plotted points are not mean values. They are the peak count rate at an angle as determined by a peak-fit of a 50-bin histogram of all values at that angle. For high count rate angles, the peak of the fit is coincident with the mean value. For low count rate angles, this is not the case. The thick plotted lines are not fit lines. They are just a visual aid to the eye, tracing through the peak count rate for each channel. Taking a sum over more altitude bins would have been a better approach, as the reviewer mentioned in item R3: P12 L14a. We did some sensitivity tests by blocking the entrance aperture of the cube polarizer (the extra one added for calibrations) and measuring to determine our lower bound on photon counts, so we should in future be able to use this to help estimate a "background" measurement on top of which our signal (even if that signal is zero photons) will sit.

Action: None

R3: P15 L14-15: "For situations in which the true signal is zero, a mean of the measured signal will be reported as a larger value, thus not being indicative of the most probable photon counting result." - This is one reason why the number of photoncounts must be increased either by increasing the time interval or by increasing the altitude window to hundred of meters or even to km

Response: See response to item R3: P14 L20.

Action: Same as response to response to item R3: P14 L20.

R3: P15 L23: "by others in the community" to -> in the remote sensing community

Response: Agreed.

Action: Will change "by others in the community" to "by others in the remote sensing community".

R3: P15 L24-25: "This assumes that the telescope does not contribute to these quantities in a significant way." - is this assumption based on any measurement? We see that the telescope includes many 45 degree mirrors. This should have a significant influence.

Response: This is a significant drawback to the calibrations we were able to manage within the practical resources of the CRL lab. The authors are aware that Newtonian telescopes are notorious for contributing to the polarization of the beam transmitted through them. One check on this is the tests given in Section 7, particularly in Table 1. This tests only changes in polarization (not retardation, rotation, etc), but indicates that the whole assembly of the roof window plus the three telescope mirrors are a contributor, attenuating the perpendicular signal quite a bit (test 9). The Visible Long Wave Pass filter's effect is larger (test 5). Note that our telescope is Dall-Kirkham, and not Newtonian. Neely2013, which also uses a Dall-Kirkham telescope, also states "The receiver consists of a F/14.3 Dall–Kirkham telescope with a 508-cm focal length and 35.6-cm aperture (see Table 1). This telescope design uses symmetric low angles of incidence on the mirrors, which minimizes polarization effects caused by the system.", which is encouraging as ours is of similar design. Interestingly, the four focus stage mirrors in our lidar (also 45 degrees) contribute almost nothing to the suppression of either the parallel or perpendicular signal (test 8). This is expected because two of the focus stage mirrors are in one plane, and two in another plane, so the polarization effects should cancel out.

Actions: Modify the two sentences here: "This assumes that the telescope does not contribute to these quantities in a significant way. This result is reasonable, as the reflectivity of all telescope mirrors are high." such that they read "This assumes that the Dall-Kirkham telescope does not contribute to these quantities in a significant way. This result is reasonable, as the total reflectivity of all telescope mirrors is high for unpolarized light, thus limiting the amount by which light of either polarization plane can be reduced relative to the other. "

Add new sentences after: "The focus stage of the telescope contains four mirrors in two planes so that polarization induced by one mirror will be cancelled out by that induced in the next. Thus it is reasonable that the focus stage does not contribute a large amount to $d$ and $\delta$. Nevertheless, the exclusion of the telescope is a limitation of the calibration, and quantities calculated by this test will not be totally representative of the whole lidar system."

R3: P15 L25-26: "This result is reasonable, as the reflectivity of all telescope mirrors are high." - How does this statement exclude any depolarization effects that the optics may have on the collected light?

Response: If the reflectivity for unpolarisad light at the relevant incidence angle is 99%, for example, then the worst is 2% difference in the two directions. Otherwise you cannot get 99% from the sum.

Action: See action for item R3: P15 L24-25.

R3: P16 L08: "rather the of the polychromator" -> one extra "the"

Response: Typo.

Action: Remove the extra "the". Change "This value is not representative of the whole receiver; rather the of the polychromator only." to read "This value is not representative of the whole receiver; rather of the polychromator only.".

R3: P16 L08-09: "It includes no effects of the telescope or focus stage." - In the upper paragraph the author stated that: ".... This assumes that the telescope does not contribute to these quantities in a significant way. This result is reasonable, as the reflectivity of all telescope mirrors are high." This paragraph states that the results are not representative since the telescope is not included. These two statements contradict each other. Please reconsider the statements.

Response: We're assuming they have no rotation, retardation, etc, effect. We allow them a partial polarization effect. The calibration is limited in this way, unfortunately. We have modified the text for item R3: P15 L24-25 to account for the reviewer response here in item R3: P16 L08-09. We have also included text in the introduction listing other lidar groups who make their k calibrations without consideration of their telescopes.

Action: See response to item R3: P15 L24-25.

R3: P16 L10: "Because of these limitations ..... k, using an unpolarized light test" - unpolarized light was also used in the upper section. Please reformulate so that the reader clearly understand what the author mean by this statement.

Response: There are no unpolarized light tests preceding this. We write about the optic used to generate it, and to send unpolarized light through a polarizer for a polarized light test earlier. We also write about how we would need an unpolarized light test to find k. But this section on page 16 is the first actual test where unpolarized light goes right into the receiver without being sent through a polarization generating calibration optic also.

Action: None.

R3: P16 L12-13: "optics. Having measured the partial-polarizer-like form of the upstream optics Mueller Matrix using the polarized calibration test, we can proceed with confidence in the tests in the following section." – consider reformulating

Response: We will rephrase this.

Action: "Obtaining a partial-polarizer-like form of the upstream optics Mueller Matrix as a result in our polarized calibration test (Section 5) allows us to proceed to Section 6 with confidence that the tests we will use to determine k and M10/M00 for the whole receiver are applicable to the CRL."

R1: Page 16 line 18 should read …to go through …

Response: typo.

Action: Change sentence "The first method forces backscattered lidar light go through a depolarizing sheet of glassine waxed paper before being measured" to read "The first method forces backscattered lidar light to go through a depolarizing sheet of glassine waxed paper before being measured"

R3: P16 L20: "it enters the lab" – could be changed to: is collected by the instrument

Response: The requested change risks rendering the meaning of the sentence incorrect. We are concerned with unpolarized light hitting the very first optic in the receiver (i.e. the window). To say "collected by the instrument" is ambiguous in this context. While some readers would consider this to mean "hits the window which is the first optic in the instrument, and which collects the light", other readers may read this as "enters the PMT directly" or worse, "Produces a signal of X in the data file".  The wording leaves something to be desired, so we will fix this.

Action: Change the sentence "Either of these methods is preferable to using sky light alone without ensuring its total depolarization as it enters the lab." to read "Either of these methods is preferable to using sky light alone without ensuring its total depolarization as it enters the window above the telescope."

R3: P16 L22: "complete depolarization is" - what is complete depolarization? is this complete for cases when the depolarization ratio is 1? Is this condition satisfied in ice clouds?

Response: This comment is in reference to the calibrations done by some lidar groups which simply *assume* that all ice clouds produce d = 1, which is totally unpolarized light. Obviously this is not true in the atmosphere, which is the point of doing a glassine sheet test instead. Yes, "complete depolarization" is meant to indicate d = 1.

Action: Modify the sentence "Even in atmospheric conditions which are thought to be depolarizing (e.g. clouds in which multiple scattering is expected, or ice clouds for which complete depolarization is expected), complete depolarization at all altitudes for the duration of the measurement cannot be ensured." to read "Even in atmospheric conditions which are thought to be depolarizing (e.g. clouds in which multiple scattering is expected, or ice clouds for which complete depolarization of d = 1 is possible), complete depolarization at all altitudes for the duration of the measurement cannot be ensured."

Author comment: Page 17 L 11: Removing extraneous words.

Action: Remove "in any case" from the sentence "Note that these equations for M10/M00 and k work equally well for the case in which we use a lamp to illuminate the lidar as the several differences in the initial matrix equation cancel out in any case:..." such that it reads "Note that these equations for M10/M00 10 and k work equally well for the case in which we use a lamp to illuminate the lidar as the several differences in the initial matrix equation cancel out:".

Author comment: Page 17 L 13: Missed a pluralization in the following sentence: "The constant M10/M00 and k can be calculated from one another".

Action: Pluralize "constant" such that the sentence reads "The constants M10/M00 and k can be calculated from one another."

R2: P17L21: I understand that authors use the depolarization sheet to isolate the polarizing effect since *there is no alternative* but, in any case, I would appreciate the technical specifications of the depolarization sheet (glassine). From the phrase 'To keep the photon count rates as high as possible during the test, only a single layer of glassine was used, although using two sheets in series ensures more complete depolarization' (P18L26), I would say that depolarizer is not perfect (d≠1). Was the depolarization degree of the sheet measured? Any information in this regard would be great for the scientific community (accurate measurements of the polarizing characteristics of the *depolarization sheet* are not so common). Did authors consider use the equations to find the effect of an almost perfect *depolarization sheet?*

Response: The discovery that the glassine sheets were effective depolarizers was a good one! Yes, we did some measurements of its properties, and plan to carry out further tests in the future to be more precise. Page 10 line 6 lists the results of our tests. The testing was done with an LED light source, a 532 nm interference filter (to match the laser wavelength), one piece of sheet polarizer held static for generating the polarized light which enters the glassine, then the glassine itself, then another piece of linear sheet polarizer which was rotated as an analyzer after the light passes through the glassine. The intensity was measured as a function of the rotation angle of the 2nd polarizer by a photodiode with a voltmeter. The test was repeated with one sheet and two sheets of glassine. Full details are available in McCullough 2015, Chapter 4.6.2.1 "Calibration tests of the optical qualities of glassine waxed paper". We did not consider the equations in terms of a non-perfect depolarization sheet. With other uncertainties much larger than 1% (the residual polarization after one glassine sheet), it is not expected that defects in the depolarization capability of the glassine will dominate the error budget. Conversely, they are expected to be dominated by other sources of error.

Action: We will change a sentence, and add a sentence to page 10. Change the sentence "After one sheet the residual polarisation is less than 1% (Polarization = 0.009±0.006), and two sheets in series eliminates the

polarization completely, as tested by our group.", such that it now reads "As tested by our group, after one sheet of glassine the residual polarisation is less than 1% (Polarization = 0.009±0.006)." Then add a new sentence which reads "Full details of this characterization are available in McCullough 2015, Chapter 4.6.2.1 "Calibration tests of the optical qualities of glassine waxed paper"".

The full paragraph is now: "Glassine waxed paper depolarizer: Typically used to protect works of art, the depolarizing properties of Lineco Glassine 20 (Lineco Glassine Acid Free Tissue 16" × 20", 12 pack, product number 448-1626) were found to be highly satisfactory. As tested by our group, after one sheet of glassine the residual polarisation is less than 1 % (Polarization = 0.009 ± 0.006). Full details of this characterization are available in McCullough (2015), Chapter 4.6.2.1 "Calibration tests of the optical qualities of glassine waxed paper". The depolarizing properties were not affected by the product's exposure to damp, nor to wetting and subsequent drying out. This product was mounted in such a way as to be held relatively taut in a frame, or held in place by other mechanical means, depending on the specific calibration test. See McCullough (2015) for photographs of the arrangements used for the calibrations in this paper. Glassine is highly scattering material, so when the entire roof hatch window is covered with it, the received lidar signal is greatly reduced."

*R3: P17 L23: When the roof hatch open is open – please review*

Response: CRL has a 1 m cube metal box extending above the roof window as a sort of chimney. The top of this box is the "roof hatch", which acts as the lid of the box. A motor is used to open this lid when weather is appropriate for observations. When weather is bad, or the lidar is off, the roof hatch can be closed to protect the delicate roof window. This is not relevant to the contents of the paper, so the reference to the roof hatch will be removed from the manuscript.

Action: Remove beginning of the sentence "When the roof hatch open is open, the glassine sheet is exposed to wind..." to read "When the lidar is operating, the glassine sheet is exposed to wind..."

R3: P17 L23: "Using a flexible material like glassine was important in the Arctic winter." – is this sentence important?

Response: This sentence is important. A solid object cannot fit down the top of the roof hatch box in order to be placed over the roof window without great risk to the window itself, particularly if dropped knocked against the window. The object must also be able to be carried up a ladder, and through a constricted ladder cage, by a person, at -50 C, in the darkness, and in some amount of wind. This is not trivial. Having an optic which is not going to fly away and break is important. An optic which can be somewhat folded to be carried up is fantastic! It is an important practical consideration for CRL's calibrations, so we choose to leave this information in the manuscript.

Action: None.

R3: P18 L03-09: is this explanation really necessary? Could this be replaced with a comment on what were the requirements of the setup?

Response: We wish to keep this explanation here. In doing these tests ourselves, we found that many previous references were very vague as to exactly how they got depolarized light into their system, etc. We had to do quite a bit of testing and refining of laboratory methods in order to come up with something that would work. We choose to explain in detail the methodology used so that readers fully understand the experiments.

Action: None.

R3: P18 L14: "and has been indicated here in white." – Please remove

Response: We will reword this. Adding this information in the text was requested by several thesis reviewers, and the authors believe that the addition is a good one. We fix a typo in "ovarlap" as well.

Action: Change "...geometric ovarlap function of CRL, and has been indicated here in white." to read "geometric

overlap function of CRL, and has been indicated in Fig. 3 in white".

R3: P20 L01-07: this section could be reduced. Suggestion: "Different approaches showed that the best retrieval method for the assessment of k was to .....(2.)......

Response: As other reviewers did not request a reduction of this section, and because this section provides responses to questions asked by Reviewer 3 (e.g. items R3b, R3 P10 L24, R3 P12 L15, and R3 P15 L24-25), we choose to retain Section 7 basically in its present form.

Action: None

R3: P20 L17-18: "directly before the polarotor. It was moved sequentially upstream, placed between any two optics where there was room to safely insert it, up to and including right in front of the lamp, upstream of the focus stage" – suggest changing to -> "before the polarotor and then moved sequentially upstream ."

Response: We will modify the sentence, to satisfy both your suggestion and some information that we feel is necessary to retain. We feel that the added detail provides, without dwelling on it, the motivation for doing the tests only in the locations indicated by Table 1 and Figure 1. Otherwise, the obvious question would be "why didn't we test more locations". The reference to Figure 1 was requested in technical review.

Action: Change "This test started with the depolarizing sheet directly before the polarotor. It was moved sequentially upstream, placed between any two optics where there was room to safely insert it, up to and including right in front of the lamp, upstream of the focus stage (Fig. 1, marked as numbers from 1 through 8)." to read "This test started with the depolarizing sheet directly before the polarotor and then moved sequentially upstream, placed between any two optics where there was room to safely insert it (Fig. 1, marked as numbers from 1 through 8)."

R3: P20 L19: "Industrial kitchen grade waxed paper was used for this test" - since the author presented a better solution for this material - the test should at least include a comparison between the two and a explanation on why this material is still presented in the manuscript.

Response: We only found out about glassine much later, after we'd already completed the tests with waxed paper. The location of the CRL laboratory in the High Arctic makes it inaccessible to our lidar operators most of the year. We have limited opportunity to repeat tests once they have given us the required results to within acceptable uncertainties. The waxed paper test in Table 1 was carried out before all the other tests, using a borrowed 10 cm x 10 cm sheet of waxed paper from the Eureka Weather Station kitchens. Its good optical depolarizing quality is demonstrated with the results from Test 1 and Test 4. Following that trip to CRL, we tested numerous waxed paper brands at our university laboratory, including our original waxed paper sample. Other brands were found to have highly variable optical properties, and none as good as our original sample of waxed paper from Eureka. Parchment paper does not produce good results, either. However, Lineco Glassine was found to be fantastic: It comes in large sheets, and the optical properties are consistent from sheet to sheet, and package to package. Thus, our later tests for k and M10/M00 were made for the whole system using glassine. It was deemed not worth the time and risk to the instrument to repeat the tests from Table 1 using glassine. This test is time consuming to carry out, and has certain risk to the receiver's delicate optics. Also, the values in that table operate mainly as indicators of which optics are responsible for the depolarization in the receiver, but we do not use any of these values in the final calibration calculations of Section 6. A thorough comparison of the materials is beyond the scope of this paper, but we will emphasize here that with such variation between waxed paper brands, it is not the recommended material to use when we have a better-characterized, inexpensive, easily-identified, readily-available option with Lineco Glassine.

Action: Add the new sentence "Glassine was later found to be more consistently depolarizing from sheet to sheet than are general brands of waxed paper, but we elected not to re-do the test in Sect. 7.2, because the particular sheet of waxed paper used here was tested in our laboratory, and found to be sufficiently depolarizing." at the end of the paragraph.

R3: P21 L15: "Many lidar groups" - please provide examples

Response: A reasonable suggestion. The original sentence reads "Many lidar groups choose to use calibration lamps part way through their system, rather than using a lamp which scans or is projected over the whole entrance aperture at the first optic of the system. (One notable exception to this trend is the lidar group at Howard University led by Prof. Venable."

Action: Will include some references. The sentences will now read: "Many lidar groups (Sassen and Benson, 2001; Alvarez et al., 2006; Wang et al., 2009; Freudenthaler et al., 2009) choose to use calibration lamps, or depolarization optics, part way through their system. An alternative is to use a lamp which scans or is projected over the whole entrance aperture at the first optic of the system. Notably, the lidar group at Howard University uses a mapping lamp applied to a water vapour lidar (Venable et al., 2011)."

Author comment: Page 21 L 19:

Action: Reword the sentence "Tests become easier to do as one moves the optic downstream in the detector..." such that it now reads "Tests become easier farther downstream in the detector for several reasons..."

R3: P21 L28: "the more convenient calibration is insufficient." - What does this mean?

Response: For CRL these tests show that k changes by a factor of 3.4 between the entrance to the polychromator and the entrance to the entire system. Therefore calibrating from the the more convenient points is insufficient and the whole system from the entrance aperture must be involved

Action: None.

R3: P21 L30a: "remove or upgrade the" – suggest changing to -> "change the"

Response: Agreed. Will include suggestion from item R3: P21 L30 here as well.

Action: Change sentence "A second use for these test measurements is that they allow us to see which optics would be most advantageous to remove or upgrade the next time we change optics in the lidar." to read "A second use for these test measurements is that they allow us to see which optics would be most advantageous to change the next time we upgrade optics in the lidar."

R3: P21 L30b: "we change optics in" – suggest changing to -> We upgrade

Response: See response from item R3: P21 L30a, above.

Action: See response from item R3: P21 L30a, above.

R1: Page 22 Line 0: Fig. 5. The parameter (sigma delta) has not been defined.

Response: Yes. Will fix this. See also the expanded text indicated in item R3: P22 L0 which describes a little about how this value is calculated.

Action: Change Fig. 5 caption first sentence from "Atmospheric depolarization ratio measurements from 12 March 2013, with associated absolute uncertainties in units of depolarization ratio." to read "Atmospheric depolarization ratio measurements (\delta) from 12 March 2013, with associated absolute uncertainties (\sigma\delta) in units of depolarization ratio."

R3: P22 L0: Figure 5: "25%" – how did you estimate this value?

Response:  The whole paragraph currently reads "Using the best determination of the calibration constant, k = 21.0±0.2, (Sect. 6.3), we can determine \delta and d using Eq. (4) for a day's measurements to show the performance of CRL. Here we use measurements obtained on 12 March 2013, which we chose because two distinct cloud morphologies are present, as are a variety of signal levels in both depolarization channels, and

because some particular places in the plot require special interpretation, which is discussed below. Figures 5 and 6 show the depolarization ratio, depolarization parameter, and the uncertainties and relative errors for each. Many data points have uncertainties on the order of 10% and smaller."

Parts requiring attention are:

a) We missed writing Eq. (5) in the first sentence. We must include a reference to Eq. (5) because that is the equation for d.

Action: Change the sentence "Using the best determination of the calibration constant, k = 21.0±0.2, (Sect. 6.3), we can determine \delta and d using Eq. (4) for a day's measurements to show the performance of CRL." such that it now reads  "Using the best determination of the calibration constant, k = 21.0±0.2, (Sect. 6.3), we can determine \delta and d using Eq. (4) and Eq. (5) for a day's measurements to show the performance of CRL."

b) We need to explain how absolute uncertainty is determined.

Response: An uncertainty budget is worked out in McCullough 2015, Chapter 5 (pp. 66 - 134). It is not shown in this version of the paper. The errors were propagated using standard uncertainty propagation rules through the algebraic functions shown in this document, assuming uncorrelated errors. We must add some of this information back in to answer questions from the reviewers.

Action: After the sentence "Many data points have uncertainties on the order of 10% and smaller.", add a new subsection: 8.1 "Uncertainty propagation", which includes the following text:

The uncertainty and errors were propagated using standard uncertainty propagation rules through low-level data processing equations, and then through the algebraic functions shown in this document, assuming uncorrelated errors (for a full account see McCullough (2015)). The Licel system uses simultaneous photon counting and analogue detection for the PMTs, and thus two signals are merged together for each "measurement channel". In all cases, the data is read into software and overflow flagged bits are removed. Uncertainties are initially propagated separately for photon counting and analogue signals. Then they are combined.

For photon counting data, most of the uncertainty in the photon counting channel is derived from shot noise. This uncertainty is then propagated through the equations of the low level data processing in the following order: 1. Begin with photon shot noise. This is applicable for both raw counts and raw background counts. This is a statistical uncertainty. 2. Propagate this uncertainty through the dead time equation, in which the dead time uncertainty is a systematic uncertainty (cannot improve by coadding or longer integration). 3. Propagate the uncertainty through the coadding equations. This generally reduces the overall size of the relative uncertainty (see Sect. 8.1) in the measurement. 4. Determine the uncertainty in the background level, which was determined from the dead time corrected background values at high altitudes. This is a statistical uncertainty, because it incudes the shot noise uncertainty. 5. Add the dead-time corrected, coadded, shot uncertainty and the dead-time corrected, coadded, background uncertainty in quadrature to get a total photon counting uncertainty for each data point.

For analogue data, the procedure is different: 1. Begin with raw analogue digital signals. 2. Account for the analogue- to-digital converter uncertainty. 3. Include the analogue shot noise uncertainty (a statistical uncertainty). 4. Account for the uncertainty in turning analogue count rates into range-scaled values. 5. Remove dark count profiles and account for this uncertainty. 6. Determine the uncertainty in the sky background constant determined from background values at high altitudes. This is a statistical uncertainty, because it incudes the shot noise uncertainty. Then account for it during background subtraction. 7. Account for the uncertainty involved in converting analogue range-scaled voltage signals to equivalent photon count rates (uncertainty depends on precision when calculating the "gluing coefficients" or "merging coefficients" based on fits to lidar data).

Finally, the two profiles, photon counting and analogue, are combined into a single resulting photocount profile for each measurement channel, for each (coadded) time bin. At each point, the raw photon counting rate determines which contributor's count value will be included: photon counting or analogue. The uncertainty at each point in the combined profile is then simply the uncertainty of the contributing data point. These combined

profiles can then be used for higher analysis into useful data products such as depolarization ratio and depolarization parameter. The uncertainties in these profiles are the uncertainties on the values $S_{\parallel}$ and $S_{\perp}$ which appear in Eq. (2) and Eq. (3) of this paper. We then propagate these uncertainties through the equations Eq. (2) and Eq. (3), again assuming uncorrelated errors, to arrive at an overall estimate for the absolute uncertainty in $\delta$ and d.

The uncertainties given as "absolute uncertainty" are the 1-sigma combined effects of systematic and statistical uncertainties and errors, and are expressed as, for example, d = 0.200 ± 0.003, in which the ± uncertainty is given in the same units as d, and would directly give the size of an error bar on a 2-D plot. Some confusion may arise when comparing values of $\delta$ and d from this document to those given elsewhere. Because depolarization parameter can exist from values of 0 to 1, some authors express "d = 0.2" as "d = 20%", for example. This is never done in the present document, precisely to avoid ambiguity with the relative errors discussed next. All values of $\delta$ and d are given as decimal, non-percent numbers, as are absolute uncertainty values. In other words, expressed with absolute uncertainty, we have $x \pm \sigma_x$ . Relative uncertainty aims to describe the uncertainty in relation to the measured size of the value of d, and is expressed as a percent: d = 0.200 ± (0.003/0.2) $*$ 100%, which is d = 0.2 ± 1.5%, for the same example. In other words, expressed as relative uncertainty, we have $x \pm (\sigma_x/x)$.

c) We need to better explain what we mean by absolute vs. relative uncertainty, so that we can explain how the 25% relative uncertainty is arrived at.

Response: Absolute uncertainty is, for example: d = 0.200 +/- 0.003, in which that +/- uncertainty is given in the same units as d. (Literally, draw error bars of 0.003 size above and below the value of 0.2). Relative uncertainty is, for the same example: d = 0.200 +/- (0.003/0.2)*100% --> d = 0.2 +/- 1.5 %.

Action: Following the new text from item (b), above, include the following new subsection: 8.2 "Relative uncertainty in \delta and d". Include the following new text in this section:

The absolute uncertainty described in Section 8.1 is always expressed in units of \delta or d, as in d = 0.200 +/- 0.003. In other words, expressed with absolute uncertainty, we have x +/- sigma_x. Relative uncertainty aims to describe the uncertainty in relation to the measured size of the value of d, and is expressed as a percent: d = 0.200 +/- (0.003/0.2)*100% --> d = 0.2 +/- 1.5 %, for the same example. In other words, expressed as relative uncertainty, we have x +/- (\sigma_x)/x.

d) Now that we have added two subsections, we must give the remainder of Section 8 its own subsection title.

Response: We create a new subtitile.

Action: Create the new subsection 8.3, entitled "Interpretation of sample atmospheric measurements". Text beginning with the sentence "Just below 2km altitude, a region of high depolarization is evident with low uncertainty. This implies that this region of the cloud is icy rather than made of liquid droplets." and continuing to the end of Section 8 will now exist in this new subsection.

R3: P22 L02: "on 12 March 2013," - this date is prior to the calibration date. It must be stated that the calibration factors are constant and can be used for measurements collected before the actual calibration date.

Response: Oversight. We will address this.

Action: Add a sentence following the sentence ending in '"...plot require special interpretation, which is discussed below.". The new sentence will read "The calibration constant k has been measured to be stable on the scale of several years' time for CRL. Therefore the time between the k calibration example shown in this paper (1 April 2013) and the date of the measurements given in Section 8 (12 March 2013) is of no consequence. "

R3: P22 L03: "because two distinct cloud morphologies are present" - A simpler example should be used at the beginning since the aim is to demonstrate the performance of the instrument.

Response: It is simple to show an example with a single type of cloud, but this would not effectively demonstrate the performance of the instrument. To perform well, CRL must be able to handle a wide range of depolarizing atmospheric conditions, several of which are demonstrated in the example of 12 March 2013. None of the morphologies are coincident in time in this example, so the results from each time period during that day (e.g. before 11 UTC versus after 11 UTC) will not be conflated with results from the other, and we feel that the example day we chose is sufficiently clear without being a trivial example.

Action: None

R3: P22 L10a: "regions, but this does not tell the whole story" - is this part necessary?

Response: This comment exists because of our experience seeing how depolarization parameter measurements are interpreted by the wider atmospheric community. That is, people tend *not* to distinguish between a particular data point having some uncertainty while still being a proper measurement indicative of the state of the atmosphere at that location at that time, and a data point which has a well-constrained uncertainty and yet which does not in any way reflect the actual state of the atmosphere. Even if the uncertainties quoted were zero, some more sophisticated interpretation using the context of the measurement is required in order that the values be used. For example, in regions far within a very thick cloud, the basic Rayleigh single scattering lidar assumptions are not appropriate - but we can still calculate a depolarization value. It just won't necessarily mean anything.

Action: We have reworded the sentences "Is this because the cloud has suddenly turned into liquid droplets? Perhaps, but there are a few other factors to consider. First, the uncertainty is higher in these regions, but this does not tell the whole story." such that it now reads "This could be because the cloud has suddenly turned into liquid droplets, but there are other factors to consider: First, the uncertainty is higher in these regions, but this does not tell the whole story."

R3: P22 L10b: "This calculated uncertainty expresses only the uncertainty in the calculated result from Eq. (4)" – consider reformulating

Response: See response to item R3: P22 L10a. Perhaps we are not making this point quite clearly. We will consider rewording part of this paragraph.

Action: None. See response to item R3: P22 L10a.

R3: P22 L14: "valid as a proxy for particle phase – despite our (possibly precise) ability to calculate it." - consider excluding this section

Response: See response to item R3: P22 L10a.

Action: None. See response to item R3: P22 L10a.

R3: P22 L15a: "is decreasingly trustworthy high in the cloud" – consider reformulating

Response: See response to item R3: P22 L10a.

Action: None. See response to item R3: P22 L10a.

R3: P22 L15b: Figure 6: could this figure be included in fig 5 with two color scales for the two parameters?

Response: This would needlessly complicate the plot with colour scales which are difficult to interpret. If one of d or \delta is on a linear scale, the other will not be. This can be seen by examination of the Eqn 3.  Further, this paper is expected to be of use to a variety of readers: Those who (optimally, in our opinion) examine depolarization in terms of d, and those who (perhaps for familiarity or historical reasons either in interpretation or in their comparison codes) work exclusively in terms of \delta. We wish to make this paper as accessible as possible to each of these large groups of potential readers, producing plots for each group in a manner that they are used to seeing.

Action: None.

R2: P22L15-P23L0: I agree with the authors that the effect of the multiple scattering on depolarization measurements has to be evaluated but is out of the scope of this work. Nevertheless, some references on this regard will be appreciated by the readers.

Response: We can accomodate this request.

Action: After the sentence "In future, it would be well to quantify the added uncertainty due to the likelihood of multiple scatters.", remove the following sentence which reads "As this is beyond the scope of this paper, care must be taken when interpreting the depolarization values, even those with low measurement uncertainty.". In its place, add the following several sentences: "The overall effect of multiple scattering is an increase in depolarization, but detailed effects are expected to differ for liquid and ice particle cases. According to laboratory experiments and theoretical calculations, multiple scattering from liquid droplets is likely to induce depolarization of no more than 3 to 4 % for a lidar with a field of view of ~1 mrad, with depolarization depending also on particle number density \cite{Liou1972} and \cite{LiouLahore1974}. Atmospheric measurements of water clouds support these estimates, and demonstrate a linear increase in multiple-scattering-induced depolarization with increased receiver acceptance angle \cite{Sassen Petrilla 1986}. More recent monte-carlo simulations of water clouds find that a third-order polynomial describes the relation between depolarization ratio and multiple-scattering fraction \cite{Hu2006}. Ray tracing code ice particles also indicates that multiple scattering leads to an increase in the depolarization ratio, however the magnitude of the increase showed a strong dependence on cloud optical thickness and particle shape \cite{NoelChepfer2002}. A full integration of these effects is beyond the scope of the present paper. Therefore care must be taken when interpreting the depolarization values presented herein, even those with low measurement uncertainty."

References to be added:
K. Liou and H. Lahore, "Laser sensing of cloud composition: a backscattered depolarization technique," Journal of Applied Meteorology, vol. 13, pp. 257–263, March 1974.
K. N. Liou, "On depolarization of visible light from water clouds for a monostatic lidar," in Journal of Atmospheric Science, vol. 29, pp. 1000–1003, 1972.
V. Noel, H. Chepfer, G. Ledanois, A. Delaval, and P. H. Flamant, "Classification of particle effective shape ratios in cirrus clouds based on the lidar depolarization ratio," Applied Optics, vol. 41, no. 21, pp. 4245–4257, 2002.
Lidar depolarization from multiple scattering in marine stratus clouds Kenneth Sassen and Richard L. Petrilla APPLIED OPTICS / Vol. 25, No. 9 / 1 May 1986
pp 1450 - 1459

R3: P23 L01-03: "A further possible contributing factor is that the two channels may have differing amounts of extinction if the transmission function of the atmosphere is polarization-dependent." – It is not clear what the author meant by this statement. More explanations are required.

Response: The lidar equation contains two transmission terms indicating how "transparent" the atmosphere is to light, between the lidar and the scattering altitude. One term is for the emitted laser light (on the way "up") and the other is for the backscattered light (on the way "back down" through the atmosphere). These are proportional to exp(- optical depth), and are described variously as in terms of optical depth and extinction (e.g. Gimmestad 2008) or transmittance (Sassen in Weitkamp 2005). In our situation, we look only at 532 nm for the emitted and returned beams, in a Rayleigh scattering process. Thus there is no wavelength-dependent difference in the transmittance terms as there would be for Raman scattering. However, there could still be a polarization-dependent difference if the atmospheric extinction is not independent of polarization state of the light passing through. If we have polarized light on the way up, and any amount of unpolarized light on the way down, and the transmission functions differ for each case, then the ratio of these functions will not quite cancel out in our equations. Sassen 2005 indicates that certain anisotropic targets such as uniformly oriented ice particles can induce such a difference for radar measurements, but that this has not been studied at length for lidar. Others (Gimmestad, etc) state that atmospheric extinction is most often independent of polarization state, and therefore this is expected to be a small effect for lidar. Similar arguments are made in Hayman and Thayer 2012, citing Kaul 1998. This consideration is beyond the scope of this paper, but because we have not controlled for this

factor, we mention it for completeness.

Action: None.

R3: P23 L11 – P24 L03: "Compare the .... as well" - please rephrase. This section is meaningful to the study but care must be given in phrasing the message. [Full section is: "Compare the data coverage of the upper left panel (panel (a)) in either Fig. 6 or Fig. 5 with the coverage of its corresponding upper right panel (panel (b)). Far more low-depolarization values are kept by cutting off using the absolute uncertainty rather than the relative uncertainty, without losing any interpretation confidence. Extra information is available at 3km just after 05:00 UTC: There are distinct regions of increased depolarization parameter which are retained when cutting depolarization parameter on absolute uncertainty rather than on relative uncertainty. Although these are above thick cloud, and so multiple scattering may influence the interpretation of the specific values of d, the relative values can still be instructive, so it is useful to retain these. Features such as liquid layers within otherwise frozen clouds, or frozen parts within liquid clouds, would be detectable in similar situations, and regions of aerosols within clear air as well." ]

Response: See the changes made in response to item R3: P22 L0. Further details regarding absolute and relative uncertainties are provided there. The authors are unclear regarding which part of the phrasing the reviewer finds objectionable. We hope that in adding the details for R3: P22 L0, the paragraph here, two pages later, will be more meaningful.

Action: None at this location.

Authors comment: Page 24 line 4: There is a typo in the word "microphyisical"

Action: Change the sentence "Depending on the application, it may be of use to some lidar users to know simply that the cloud is inhomogeneous in cloud particle phase, or that an aerosol layer is present, without the specific microphyisical details which are usually available with a well-constrained absolute depolarization parameter value." to read Change the sentence "Depending on the application, it may be of use to some lidar users to know simply that the cloud is inhomogeneous in cloud particle phase, or that an aerosol layer is present, without the specific microphysical details which are usually available with a well-constrained absolute depolarization parameter value."

R1: Page 24 line 6 – change photons to photocounts.

Response: We will change this.

Action: Change the sentence "Further coadding to lower resolution would help with the data coverage by increasing the number of perpendicular photons per bin" to read "Further coadding to lower resolution would help with the data coverage by increasing the number of perpendicular photocounts per bin".

R3: P24 L07-10: "However, this can only be carried out to a certain point, after which the low resolution depolarization measurements will be misleading, as any instances of thin liquid layers (low d and delta) residing within an ice cloud (high d and delta) would, at low resolution, show a smooth region with intermediate values of d and delta which are not actually present anywhere within the binned region. " - Please divide this section into several sentences. It is relatively hard to follow the message.

Response: We will change this.

Action: Replace the sentence which reads "However, this can only be carried out to a certain point, after which the low resolution depolarization measurements will be misleading, as any instances of thin liquid layers (low d and delta) residing within an ice cloud (high d and delta) would, at low resolution, show a smooth region with intermediate values of d and delta which are not actually present anywhere within the binned region. " with the following several sentences: "However, this can only be carried out to a certain point, after which the low resolution depolarization measurements will be misleading. For example, any instances of thin liquid layers (low

d and delta) residing within an ice cloud (high d and delta) would, at low resolution, show a smooth region with intermediate values of d and delta. As such intermediate values of d and delta are not actually present anywhere within the binned region, it would be incorrect to use these values for interpretation of the cloud itself. ".

Authors comment: Page 24 line 13: The paragraph has been expanded such that it now expresses the goals for future work, giving better context to sections 9.1 and 9.2.

Action: That paragraph now reads: "CRL can now make depolarization parameter measurements at a precision of < 10 % relative uncertainty in clouds at a resolution of 5 min × 37.5 m (Sect. 8), despite the less than optimum optical configuration of CRL. The major difficulty for CRL is in receiving sufficient perpendicular signal at the depolarization PMT, as indicated by the very large calibration value of k = 21 found in the system, and by the low photocount rates in the perpendicular channel during atmospheric measurements. For any unpolarized light which scatters back to the roof window, the portion which would be allowed through the perpendicular analyzer near the PMT is preferentially suppressed by lidar receiver optics in comparison to the portion of unpolarized light which would be allowed through the parallel analyzer. There are several possibilities for improvement of the depolarization measurements: Changes to the depolarization parameter calculation method, and changes to lidar hardware. Suggestions for each of these are provided in Sect. 9.1 and Sect. 9.2, respectively."

R1: Page 24 line 15 – delete the words perpendicular-polarized.

Response: We agree the phrasing is awkward. We will rephrase to retain the intended meaning. It is not the total number of photons which are insufficient - only the photons which are allowed through the perpendicular channel are too few. This is because they are preferentially suppressed by the receiver optics as compared to photons which are allowed through the parallel analyzer. The perpendicular signal is therefore too low to measure well. The parallel channel has plenty of photons, and therefore plenty of signal. We will modify so that we are not referring to particular photons having a particular polarization, to be more in line with the notation of Gimmestad 2008.

Action: Change the sentence "The major difficulty for CRL is in receiving sufficient perpendicular-polarized photons at the depolarization PMT, as indicated by the very large calibration value of k = 21 found in the system." and add an extra sentence, to read all together "The major difficulty for CRL is in receiving sufficient perpendicular signal at the depolarization PMT, as indicated by the very large calibration value of k = 21 found in the system, and by the low photocount rates in the perpendicular channel during atmospheric measurements. For any unpolarized light which scatters back to the roof window, the portion which would be allowed through the perpendicular analyzer near the PMT is preferentially suppressed by lidar receiver optics in comparison to the portion of unpolarized light which would be allowed through the parallel analyzer.".

R3: P24 L16-17: "There are several possibilities for improvement of the depolarization measurements: Changes to the depolarization parameter calculation method, and changes to lidar hardware." – Please rephrase

Response: Rather than rephrasing, we will add a sentence to indicate that there are more details coming in the next sections of the paper.

Action: Following the sentence "There are several possibilities for improvement of the depolarization measurements: Changes to the depolarization parameter calculation method, and changes to lidar hardware.", add the following sentence: "Suggestions for each of these are provided in Section 9.1 and 9.2, respectively."

R3: P24 L22-27: Please rephrase. Too vague

Response: We will delete the sentence.

Action: Delete the sentence "This calibration profile feeds into an alternate expression for d which depends only on the high-count-rate parallel and unpolarized channel measurement.", but keep the reference to McCullough 2017.

R3: P24 L29 – P25 L02: "Hardware....channels." – Please rephrase

Response: We will rephrase.

Action: Change the paragraph which reads: "Hardware upgrades are the other option for improving the CRL depolarization measurements. They are detailed here because any or all of these would improve both calculation methods for d by increasing signals in the perpendicular channel, and because they are of use in any lidar which does not have access to an unpolarized channel at the same wavelength as its depolarization channels. In the context of the calibrations in this paper, hardware improvements seek to reduce the value of k by increasing the number of perpendicular photons which reach the depolarization PMT." to read:"Hardware upgrades are the other option for improving the CRL depolarization measurements. Any or all of these would improve both calculation methods for d by increasing signals in the perpendicular channel. In the context of the calibrations in this paper, hardware improvements seek to reduce the value of k by increasing the number of perpendicular photons which reach the depolarization PMT. These hardware improvements are also relevant for any other lidar which does not have access to a polarization-independent channel at the same wavelength as its depolarization channels, and thus which cannot take advantage of the new calculation method from Section 9.1."

R3: P25 L03-19: is there really a necessity for this paragraph?

Response: It was requested in previous drafts that we add these specific recommendations.

Action: None.

R1: Page 25 line 7 – replace perpendicular with unpolarized.

Response: This would render the statement incorrect. We don't care about the component of unpolarized light which the parallel channel can measure as these signals are already high. We will reword this, and fix the typo which is already in the sentence, as well.

Action: Change the sentence "Third, replacing the VLWP filter with one which is less polarizing, or less polarizing in the perpendicular-suppressing direction would reject as few as possible the perpendicular photons which enter the telescope." such that it reads "Third, replacing the VLWP filter with one which is less polarizing, or less polarizing in the perpendicular-suppressing direction would reject as few as possible of the photons which enter the telescope and which are eventually allowed through the perpendicular polarization analyzer of the polarotor."

R1: Page 25 line 12 "… in practice this means a lower gain …" (missing word).

Response: Typo. We also change the words "two signals" to "intensities" in the next sentence to make it accurate.

Action: Change the sentence "In practice means a lower gain setting is required to optimize the parallel channel to avoid PMT saturation." to read "In practice this means a lower gain setting is required to optimize the parallel channel to avoid PMT saturation.". Also change the sentence "The high dynamic range of the combined analogue and photon counting of the Licel recorders helps somewhat, but having two signals of more comparable levels would be a better choice." such that it now reads "The high dynamic range of the combined analogue and photon counting of the Licel recorders helps somewhat, but having intensities of more comparable levels to begin with would be a better choice."

R3: P25 L20: "Throughout this work, k values far from unity have been presented as being undesirable. There" – Is this the transmitted message? As a reader, I do not understand this message from the manuscript.

Response: We agree that this is not the overall message of the paper, but the manuscript does make clear that a large k is indicative of problems. For example, Page 6 lines 15 - 18 state "In the idealized case where the optics do not contribute to the polarization, k would be unity. However, characterizing measurements suggest a value closer to k = 21 for CRL (Sect. 6.3), indicating that optics upstream of the polarotor are significantly polarizing. As this is the case, it seemed sensible to investigate potential additional optical contributions.", which means that our very large value of k was unexpected and not desirable. This is implied elsewhere as well, as we discuss the

implications of suppressing the perpendicular signals to such a degree that we have difficulty measuring them. The point of the discussion in this part of Section 9.2 is that while we expected k = 1, and were unhappy with k >> 1, someone might actually be at an advantage in a system in which k < 1 (which is not usually discussed in the literature, but is true).

Action: None.

R1: Page 25 lines 22-24 – this was cited under General Comments. Re-write in light of the key facts.

Response: This part will be rewritten.

Action: Change the passage "For d = 1, half of the backscattered light reaching the roof window is parallel, and the other half perpendicular. For d = 0, all of the light is parallel, and none is perpendicular. Therefore, the maximum perpendicular signal ever possible to be backscattered to the lidar's roof window and through the receiver is only half the maximum parallel signal ever possible. With a whole-system calibration value k = 1, the maximum measured perpendicular signal will also be half of the maximum measured parallel signal." such that it reads "For d = 1, half of the backscattered intensity reaching the roof window would be admitted to an ideal (k = 1) parallel channel, and the other half of the intensity would be admitted to the perpendicular channel. For d = 0, all of the intensity is parallel, and none is perpendicular. Therefore, the maximum perpendicular intensity ever possible to be backscattered to the lidar's roof window and through the receiver is only half the maximum parallel signal ever possible. With a whole-system calibration value k = 1, the maximum measured perpendicular signal will also be half of the maximum measured parallel signal."

In addition, at line Page 3 line 24, the following sentence has been removed: "Only photons of the appropriate polarization orientation enter the measurement profiles for each of the parallel and perpendicular photocount measurement channels."

R2: P25L23: 'half of the backscattered light reaching the roof window is parallel…' Parallel to what? This way to understand the depolarization is dangerous. The polarization state of the photons is not binary (parallel/perpendicular). The parallel and perpendicular signal *with respect to the polarizing component of the emitted laser beam* is the way we measured the received light. Please, revise the whole manuscript.

Response: See response to item R1: Page 25 lines 22-24, for this particular passage in the paper. We will revise the manuscript to include descriptions of polarization according to Gimmestad 2008 and conventional notation. We recognize that the state of the photons is not binary, and should not have expressed this passage in that manner. For collections of photons, we can consider a vector decomposition of any planes of polarization at intermediate angles between "parallel" and "perpendicular" into components which map onto those two planes. "Parallel" is described elsewhere in the manuscript to be specifically the angle with respect to the polarization state of the emitted laser beam.

Action: Same as response to item R1: Page 25 lines 22-24. See also responses to items R1: Page 2 lines 17-21, R1: Page 3 lines 5-6, R2: P5L15, R1c, R1d, and similar.

R3: P25 L31-34a: is this section necessary for the manuscript?

Response: This explains the reasons that we didn't just fix the problems "the easy way" already, given that we know what to do in terms of hardware to improve the depolarization system.

Action: None

R3: P25 L31-34b: Conclusions - detailed information on the calibration method and its particularities must be presented.

Response: Agreed.

Action: We have rearranged the conclusions section, and have expanded it with new text to improve the paper in

this regard. The new text provides a list of each calibration test with its results and implications. The new text for Section 10, Conclusions, is included after the existing P26 L04, replacing lines P26 L05 - L08:

A single Mueller matrix suffices to describe the collective effects of all receiver optical elements between the telescope's focus stage and the Polarotor polarizing beamsplitter. The following matrix elements, and ratios of matrix elements, were measured using a polarized lamp test: M02 = 0; M12 = 0; M11 = M00; M10 = M10; M10/M00 = 0.77±0.18. Because M11 = M00 and M10 = M01, we conclude that the optics represented by the "upstream optics" matrix act as a partial polarizer (without rotation, retardation, or other optical effects) on the calculation of d and δ. Therefore, the traditional calculation equations for d (Eq. (2)) and δ (Eq. (3)), which use the single calibration constant k, are valid for the CRL.

Using an unpolarized lamp light test from the same location, the results are M10/M00 = 0.719 ± 0.001, which is equal within uncertainty to the polarized-light calibration value, and k = 6.12 ± 0.02.

An unpolarized light test using laser returns from the sky, depolarized by a Glassine sheet above the receiver roof window, includes the optical contributions of the roof window, telescope, and focus stage, in addition to the upstream optics matrix. The results of this test are M10/M00 = 0.910 ± 0.002 and k = 21.0 ± 0.2. These values are representative of the entire lidar receiver, so are applied to CRL's routine atmospheric measurements of depolarization.

Detailed optic-by-optic measurements of M10/M00 and k using unpolarized lamp light reveal that the Visible Long Wave Pass filter is the most highly polarizing optic, increasing k by a factor of 5.6. The next most polarizing optic is the combination of the telescope and roof window, which increase k by a factor of 3.1.

R3: P25 L31-34c: Conclusions on what is the most polarization sensitive optics in the lidar receiving unit must also be included.

Response: Agreed.

Action: We have added the sentences "Detailed optic-by-optic measurements of M10/M00 and k using unpolarized lamp light reveal that the Visible Long Wave Pass filter is the most highly polarizing optic, increasing k by a factor of 5.6. The next most polarizing optic is the combination of the telescope and roof window, which increase k by a factor of 3.1.", included in the Conclusions section.

R3: P26 L11: "night" to "period"

Response: Agreed.

Action: Change the sentence beginning "For the test night, a thick partially frozen cloud was present...", change the sentence to read "For the test period, a thick partially frozen cloud was present..."

Also add the text "Typical depolarization parameter absolute uncertainties are on the order of ±0.05 (≤ 10% relative uncertainty) within clouds at time and altitude resolutions of 5 min and 37.5 m respectively, with higher precision and higher resolution possible in select cases, and uncertainty somewhat larger (±0.1) at the same altitude outside of clouds." for more detail.

R3: P26 L12-13: "and the reduced reliability of the depolarization measurements farther into the thick cloud are evident as multiple scattering becomes important." – Please rephrase

Response: We will modify this to clarify.

Action: Change the sentence "For the test night, a thick partially frozen cloud was present before 10:00 UTC, and the reduced reliability of the depolarization measurements farther into the thick cloud are evident as multiple scattering becomes important." so that the following sentences are inserted instead: "For the test night, a thick partially frozen cloud was present before 10:00 UTC. As the beam penetrates farther into the thick cloud, the likelihood of multiple scattering increases, as does the extinction of the lidar beam. Therefore the depolarization

measurements have reduced reliability at higher altitudes in this cloud."

R3: P26 L13: "above it was present" – please rephrase

Response: We will modify this to clarify.

Action: Change the sentence "A non-frozen cloud with regions of higher depolarization material above it was present after 15:00 UTC." to read "After 15:00 UTC there is a smaller cloud with regions of higher and lower depolarization."

R3: P26 L14: "sensitive to the low" – suggest changing to -> "sensitive also in low"

Response: We do not feel that this change is necessary

Action: None.

R3: P26 L15: "functioning, well-characterized depolarization" – ambiguous
[Full passage reads: "This work has resulted in a functioning, well-characterized depolarization measurement system at CRL. Since this work, a new calculation technique based on similar calibration principles and Mueller Matrix algebra has been developed for CRL. This takes advantage of CRL's Visible Rayleigh Elastic unpolarized channel and produces depolarization parameter measurements to similar precision, but at 5 to 10 times higher resolution, and with better depolarization coverage of time and space. These improvements will be detailed in an upcoming publication."]

Response: We will remove the word "functioning" from this sentence.

Action: In the sentence "This work has resulted in a functioning, well-characterized depolarization measurement system at CRL. ", remove the word "functioning" such that the sentence now reads "This work has resulted in a well-characterized depolarization measurement system at CRL. "

We have also removed the rest of the existing paragraph following that sentence (see item R3: P26 L15-19, below) and have replaced it with "Using a similar Mueller algebra exercise, these calibration methods may be applied to other lidars to elucidate the properties of the optics. This work shows that it is possible to add depolarization capability to lidars which were not originally designed specfically for polarization measurements."

R3: P26 L15-19: consider excluding from the manuscript

Response: The new paper follows on directly from this one. We have mentioned it earlier in the text, so will remove this from the conclusions.

Action: Remove entire passage: "Since this work, a new calculation technique based on similar calibration principles and Mueller Matrix algebra has been developed for CRL. This takes advantage of CRL's Visible Rayleigh Elastic unpolarized channel and produces depolarization parameter measurements to similar precision, but at 5 to 10 times higher resolution, and with better depolarization coverage of time and space. These improvements will be detailed in an upcoming publication." The final paragraph has some text added which reads "This work has resulted in a well-characterized depolarization measurement system at CRL. Using a similar Mueller algebra exercise, these calibration methods may be applied to other lidars to elucidate the properties of the optics. This work shows that it is possible to add depolarization capability to lidars which were not originally designed specfically for polarization measurements."

**Depolarization calibration and measurements using the CANDAC Rayleigh-Mie-Raman Lidar at Eureka, Canada**

Emily M. McCullough[1,2a,*], Robert J. Sica[1], James R. Drummond[2], Graeme Nott[2,4a], Christopher Perro[2], Colin P. Thackray[2], Jason Hopper[2], Jonathan Doyle[2], Thomas J. Duck[2], and Kaley A. Walker[3]

[1]Department of Physics and Astronomy, The University of Western Ontario, 1151 Richmond St., London, ON, N6A 3K7
[2]Department of Physics and Atmospheric Science, Dalhousie University, 6310 Coburg Rd., PO Box 15000, Halifax, NS, B3H 4R2
[3]Department of Physics, University of Toronto, 60 St. George St., Toronto, Ontario, M5S 1A7
[4]Facility for Airborne Atmospheric Measurements, Building 146, Cranfield University, Cranfield, MK43 0AL, UK
[a]Indicates present affiliation

*Correspondence to:* Emily M. McCullough (e.mccullough@dal.ca)

**Abstract.**

The Canadian Network for the Detection of Atmospheric Change (CANDAC) Rayleigh–Mie–Raman Lidar (CRL) at Eureka, Nunavut, has measured tropospheric clouds, aerosols, and water vapour since 2007. In remote and meteorologically significant locations, such as the Canadian High Arctic, the ability to add new measurement capability to an existing well-tested facility is extremely valuable. In 2010, linear depolarization 532 nm measurement hardware was installed in the lidar's receiver. To  minimize disruption in the existing lidar channels and to preserve their existing characterization so far as is possible,  the depolarization hardware was placed near the end of the receiver cascade. The upstream optics already in place were not optimized for preserving the polarization of received light. Calibrations and Mueller  Matrix calculations are used to determine and mitigate the contribution of these upstream optics on the depolarization measurements. The results show that with appropriate calibration, indications of cloud particle phase (ice vs. water) through the use of the depolarization parameter are now possible to  a precision of $\pm 0.05$ absolute uncertainty ($\leq 10\%$ relative uncertainty) within clouds at time and altitude resolutions of 5 min  and 37.5 m respectively, with higher precision and higher resolution possible in select cases. The uncertainty is somewhat larger outside of clouds at the same altitude, typically with absolute uncertainty $\leq 0.1$. Monitoring changes in Arctic cloud composition, including particle phase, is essential for  an improved understanding of the changing climate locally and globally.

**1 Introduction**

Clouds influence Earth's radiation budget, and thus its weather and climate. Clouds reflect sunlight (cooling), and trap heat from the ground (warming). The combined effect of these competing influences is poorly understood, especially in the Arctic, because it depends significantly on the structure and microphysical properties of the clouds, and the environment in which the clouds exist (Curry et al., 1996). Ice clouds radiate differently than water clouds (Sun and Shine, 1994). Tropospheric clouds

occur frequently in the Arctic, with liquid content found at all times of year, often within mixed-phase clouds (Intieri et al., 2002; Shupe, 2011).  In order to develop models with improved fidelity of the cloud phase, more observational measurement datasets are required (Shupe, 2011), with phase transitions being of particular interest (Kalesse et al., 2016).  Measurements of

5  cloud particle phase (ice vs. water) are therefore necessary in order to more fully understand the radiation balance of the Arctic atmosphere. Liquid droplets can exist well below 0° C, so cloud temperature is not sufficient to determine the phase of cloud particles  (e.g. Shupe et al., 2008; Boer et al., 2009; Curry et al., 1996; Sassen, 1991; Westbrook and Illingworth, 2013). Lidar depolarization measurements, which discern cloudy regions containing spherical particles (i.e. liquid droplets) from those containing nonspheres (i.e. ice particles), are one method by which cloud particle phase may be examined (Schotland et al.,

10  1971; Sassen, 2005; Bourdages et al., 2009).

The Polar Environment Atmospheric Research Laboratory (PEARL) is located in Eureka, Nunavut (80° N, 86° W) in Canada's High Arctic. PEARL has more than 25 instruments dedicated to the in situ and remote sensing study of atmospheric phenomena at a latitude where few measurements are typically available. With climate changes amplified at such latitudes (Serreze and Barry, 2011), PEARL's measurements  are a valuable contribution to global atmospheric and environmental

15  science.

The Canadian Network for the Detection of Atmospheric Change (CANDAC) Rayleigh–Mie–Raman Lidar (CRL), was installed at PEARL in 2007. It has since made measurements of visible and UV particulate backscatter coefficient, aerosol extinction, water vapour mixing ratio, and other quantities using its 355 nm and 532 nm lasers and comprehensive detection package (Doyle et al., 2011; Nott et al., 2012). Adding 532 nm linear depolarization capabilities to this instrument is an

20  economical way to add additional capacity to study Arctic clouds, in concert with other instruments at PEARL such as the Millimetre Cloud Radar (Moran et al., 1998), the E-AERI interferometer (Mariani et al., 2012), and the Starphotometer (Baibakov et al., 2015). To preserve continuity in the long-term data sets from other CRL channels, no existing optics were altered or removed during the installation of the depolarization channels. Figure 1 is a diagram of the CRL's receiver, showing the seven original measurement channels, and indicating the locations of the new pellicle beamsplitter, Polarotor rotating Glan-Thomson

25  prism, interference filter, focusing lens, and photomultiplier tube of the 532.1 nm depolarization channel. CRL uses a single PMT to measure light of two polarization planes on alternate laser shots, with a laser repetition rate of 10 Hz. This is similar to Platt (1977), which operated with a laser repetition rate of 1 to 5 Hz. CRL's higher repetition rate means that the assumption of simultaneous measurements in both polarization planes is reasonable. As the original lidar optics were not chosen for their polarization properties, the optical design of the CRL has made the calibration of the depolarization measurements challenging.

**1.1 Depolarization lidar theory**

[Figure]

**Figure 1.** Diagram of CRL's receiver system, showing all 7 existing measurement channels plus the newly installed depolarization hardware. The new hardware consists of the pellicle beamsplitter, the Polarotor, and the interference filter, focusing lens, and PMT, to the right of the polarotor. Numbers correspond to calibration test numbers from Table 1, and indicate the test locations of the depolarizing sheet used in those tests. "7" also marks the location of the calibration cube polarizer added to the system for the calibrations in Sect. 5.1, and "8" is also the location of the lamp and depolarization sheet during the polarized light tests in that section. During regular lidar sky measurements, neither the depolarizing sheet nor the calibration cube polarizer remain in the optical path. This figure is based on Fig. 2 of Nott et al. (2012).

With the new depolarization capabilities, we aim specifically to investigate "the atmospheric phenomena which change the polarization state of the light received by a lidar relative to the state of the  transmitted  light" (Gimmestad, 2008). Depending on the optical qualities of the particles, a population of randomly oriented identical particles in the atmosphere should either a) not change the polarization state of the light (i.e. all light from that population will be returned polarized parallel with respect to the state of the transmitted light), or b) should cause the light to become completely unpolarized on its return. There may be more than one population of particles present in any given scattering volume. The calculation to determine the change in polarization requires a ratio of the intensity of light which is returned unpolarized to the total intensity of light which is returned in any and all polarization states (Flynn et al., 2008; Gimmestad, 2008). Expressed in this manner, the quantity of interest is $d$, the depolarization parameter: the portion of the total light intensity $I$ which has become

depolarized through scattering. Similar descriptions, called depolarization factor, are given as early as van de Hulst (1957). The depolarization parameter is defined as:

$$d = I_{\text{unpol.}}/(I_{\text{pol.}} + I_{\text{unpol.}}). \tag{1}$$

In the event that the atmosphere does not depolarize the beam, there will be no intensity returned with polarization
5  different than the transmitted light, and therefore $d = 0$. In the case of complete depolarization, $d = 1$.

Because lidars measure signals from photomultiplier tubes (PMTs), and not the backscattered light intensity directly, the equation for $d$ and must be reformulated in terms of lidar observables. Gimmestad (2008) demonstrates this development using Mueller Matrix algebra, with normalized matrices. Two quantities are measured. The first is the signal in a channel which uses a polarization analyzer to admit light polarized parallel to the polarization plane of the transmitted laser beam
10  (the "parallel" channel), and signal in a channel which uses a polarization analyzer to admit light polarized perpendicularly to this plane (the "perpendicular" channel). In this way, in the absence of any complicating factors, for linearly polarized transmitted light, the parallel channel will be sensitive to half of the backscattered light which has been unpolarized during scattering ($\frac{1}{2}I_{\text{unpol}}$) and all light which remains polarized during scattering ($I_{\text{pol}}$). The second
15 signal is that in the perpendicular channel, which will be sensitive only to half of the unpolarized light ($\frac{1}{2}I_{\text{unpol}}$), and none of the light which remains polarized when backscattered. In Gimmestad's paper, the signals in the receivers, $S$, are individually "assumed to be calibrated", but no further details about these calibrations are provided. Presumably, this assumption considers the combined effects of all optics upstream of the PMT and the gain of the PMT, acting together as a constant attenuation factor for each individual channel. If the factors differ between channels, the overall effect in the system as a whole is that of partial
20 polarizer.

Under these conditions, the equation for depolarization parameter is given as:

$$d = \frac{2kS_\perp}{S_\parallel + kS_\perp} = \frac{2k\frac{S_\perp}{S_\parallel}}{1 + k\frac{S_\perp}{S_\parallel}} = \frac{2}{\frac{1}{k}\frac{S_\parallel}{S_\perp} + 1}, \tag{2}$$

in which: $S_\perp$ is the signal measured by the perpendicular channel, $S_\parallel$ is the signal measured by the parallel channel, and $k = \frac{G_\parallel}{G_\perp}$ is the depolarization calibration constant, in which $G_\parallel$ is the gain (or attenuation) of the parallel channel, and $G_\perp$ is
25 the gain (or attenuation) of the perpendicular channel. The third form for $d$  Eq. (2) is easier to handle experimentally as each measurement appears only once and thus uncertainties may be considered uncorrelated.

 Historically, "depolarization" has also referred to $\delta$, the depolarization ratio. This quantity is proportional to the ratio of the perpendicular signal $S_\perp$ to the parallel channel $S_\parallel$ (e.g. Hohn, 1969; Schotland et al., 1971; Liou and Schotla. The depolarization parameter $d$ is directly relatable to the expression for depolarization ratio, $\delta$, through the same signal
30 measurements and the same calibration constant:

$$\delta = k\frac{S_\perp}{S_\parallel}. \tag{3}$$

The conversion between the quantities $d$ and $\delta$ is:

$$d = 2\delta/(1+\delta). \tag{4}$$

A variety of expressions for "depolarization" are described in Cairo et al. (1999). The $\delta$ described in the current paper is most closely related in meaning to the Pal and Carswell (1973) "volume linear depolarization ratio" cited therein, but it is not strictly equivalent because no claims are made here about the connection between $\delta$ and backscatter coefficient. Rather, $\delta$ is defined here only as a function of measured quantities. Gimmestad (2008) provides motivation for moving away from all $\delta$ descriptions, pointing out that $d$  is consistent with the rest of optical physics and scattering theory. Expressing depolarization as $d$ has since been adopted in such publications as Hayman and Thayer (2009, 2012) and Neely III et al. (2013). Results in the present paper will be provided in terms of both $d$ and $\delta$ so that readers working under either paradigm can readily make use of the figures and calculations.

The expressions for $d$ and $\delta$ of Eq. (2) and Eq. (3) are all referred to in this paper as "traditional" in the sense that in each equation, a single $k$ value determines the calibration.

**1.2 Literature review of depolarization calibrations**

The calibration constant $k$ can be determined by introducing unpolarized light into the detector (i.e. setting $d = \delta = 1$) or, equivalently, light polarized at $\pm 45°$ with respect to each of the planes of polarization for the detectors (also sets $d = \delta = 1$) and measuring the signals in each channel. $k$ is then the ratio of the two signals. The location of calibration optics or lamps within the lidar determines how much of the system will be characterized through the calibration. The most strict meaning of $k$ is the ratio of gains of the detectors, if the polarization state of the calibration light is defined directly before the polarizing beamsplitter. A wider interpretation for the meaning of $k$ can include relative gains resulting from other receiver optics if the polarization state is defined earlier in the receiver, and can include relative gains of transmitter optics if the polarization state for the test is set within the transmitter. The laser is assumed to be completely linearly polarized. The orientation of the parallel and perpendicular  polarization analyzers must be correctly set at $90°$ to each other, and oriented correctly with respect to the usual polarized returns from the transmitted laser beam, before the calibrations for $k$ commence. Using $k$ as the only calibration factor ignores the possibility of any retarding and rotating effects which may exist in the transmitter and receiver, in all of the optical components. This is more likely to be appropriate for  lidars which have few receiver optics before the polarization analyzer (e.g. Wang et al., 2009), and is less likely to be appropriate for lidars which have many receiver optics which are not optimized for polarization measurements, such as the CRL.

Some groups begin the calibration for $k$ with lidar returns from an atmospheric region which is assumed not to depolarize the light (or to depolarize only a known minimal amount as a result of molecular scattering) for the duration of the calibration. Then a half waveplate is included in the receiver to control the orientation of the polarized backscattered light as it enters the detectors, aligning it at $\pm 45°$ with respect to both polarization analyzers (Spinhirne et al., 1982; McGill et al., 2002; Alvarez et al., 2006; D This type of calibration is not sensitive to polarization effects in the transmitter optics, nor to any optics upstream of the

 half wave plate. This typically means omitting at least the telescope, and sometimes more optics. A notable exception is Neely III et al. (2013), which has a waveplate optic in the roof window.

In an alternate version of the half waveplate calibration, this waveplate may be placed in the transmitter to control the orientation of the plane of polarization of the laser light transmitted to the sky (Liu and Wang, 2013; Neely III et al., 2013; Freudenthaler, 2( Eloranta and Piironen (1994) uses a pockels cell in their laser to the same effect. Locating the calibration optic in the transmitter is a method which includes as many optics as possible in the calibration.

All calibrations using polarized light must be concerned with obtaining the correct orientation of the waveplate (or other relevant optic). Spinhirne et al. (1982) and some lidar examples in Freudenthaler et al. (2009) arrange the waveplate as well as possible such that the output is at $+45°$ and  /or $-45°$, and report the calibration measurements only from those specific angles. Other groups show improved results for $k$ by calculating its value at a variety of waveplate rotation angles, then using a fit to determine the optimal rotation angle from which to use the calibration values (Alvarez et al., 2006; Hayman and Thayer, 2009; Snels e Of those in the latter case, methods using both $+45°$ and $-45°$ measurements together (calibrations $90°$ apart from one another), can have error terms which compensate well for one another in the event that the waveplate is misaligned by the same amount in each case (Freudenthaler et al., 2009, some systems in Freudenthaler, 2016).

Sassen and Benson (2001) use a different method for simulating a $d = \delta = 1$ situation. They introduce unpolarized lamp light to their detector from the point of focus of the telescope. This calibration method is not sensitive to any polarization effects in the transmitter optics or the telescope, but the polarization state of the calibration light is well-known, and there are no calibration optic rotation angles to introduce errors.

CRL calibrations for $k$ use a sheet of depolarizing material in the receiver. This can be placed at a variety of locations within the receiver. The results of these tests provide the motivation for this paper. When using a depolarizing sheet directly upstream of the polarotor (see Fig. 1, location 1 and 4; same location as half wave plate used in e.g. Alvarez et al. (2006); Wang et al. (2009); Freudent we find that $k = 1$. This is exactly as expected. Effectively, this is an estimate for the strict version of $k$: the ratio of PMT gains – and CRL uses the same PMT for each depolarization channel. Repeating the calibration measurements with a depolarizing sheet at the entrance to the receiver roof window suggest a value closer to $k = 21.0 \pm 0.2$ for the whole CRL receiver (Sect. 6.3), indicating that optics upstream of the polarotor are significantly polarizing. Clearly a more thorough instrument depolarization characterization is required for the CRL. If our optics are so highly polarizing, they may carry other optical consequences as well, which Eq. (2) and Eq. (3) are insufficient to describe.

Various approaches are available in the literature to account for non-ideal depolarization lidar hardware, each with their own assumptions. Some calibrations are tests with a temporarily-installed optic. These allow for calibration factors to be determined, which will then be applied to regular measurements made without the temporary calibration optic in place. Other

calibrations consist of adjusting compensation optics (typically by adjusting their rotation angle) which will remain in the lidar during regular measurements. Some of these are the same optics used for the calculations of $k$. At present, the lidar of Neely III et al. (2013) seems most capable of a whole-system calibration. Their calibration optics exist in multiple places within the transmitter, and multiple places within the receiver. The laser light can be rotated directly as it exits the laser, again as it exits the laboratory, again as it enters the telescope, and altered yet again as the light enters the PMTs. Further, the lidar was designed to make depolarization measurements, and optics could be selected and oriented with this in mind, as indicated in Hayman and Thayer (2012). Their liquid crystal variable retarder has some effects related to laboratory temperature which must be considered, but the authors have accounted for these. No other lidar of which we are aware has all of these capabilities. The more common calibrations each assess only some of the possible complications. Two are discussed briefly here.

[remaining 58,210 characters of this post omitted]

---

## Referee Report (RR1)

General Comments

The paper presents an experimental method to assess the performance of the depolarization channels newly installed on the CRL lidar instrument at Eureka, Nunavut. It gives a nice overview of the upgrades performed on the CRL and of the efforts required to fully characterize the newly installed depolarization channels. This paper would be helpful for lidar groups that are working on adding depolarization channels to existing lidar instruments since part of the techniques have a general applicability.

The updated version of the paper covers many new key points required for a hardware orientated paper. Also, the new structure of the paper is much easier to follow, giving a clearer overview of the intended aim. Still, I have a feeling that the concluding remarks could be improved. Since the paper is focused on the experimental perspective, more weight should be given to the upgrades and techniques used in the study. The current conclusions could be considered as part of the discussions since values for specific parameters (like Mueller parameters, k) are provided in this section and discussions on these values are also given. I would suggest shifting part of the conclusions to the discussions sections and give a more general touch to the conclusions.

As an example: the conclusions should also cover a description on how was the depolarization channel calibrated, what were the methods used to calibrate – with focus on the new particularities used for the CRL, strengths and weaknesses of the methods used to calibrate the depolarization, problems and limitations encountered during the study and the methods used to overcome them, what are the optical components that mainly influence the depolarization (partially covered but more information could be given covering all the optical components), what is the accuracy of your measurements (already covered but more weight should be given to this), the performance of the lidar depolarization channels after this study (how did the study improved the collected depolarization products).

It would be nice to see this paper published since it makes use of original methods to calibrate the depolarization channels – methods that would also be helpful for other lidar groups.